# WHEN AND WHERE TO RESET MATTERS FOR LONG-TERM TEST-TIME ADAPTATION

**Taejun Lim**[1]    **Joongwon Hwang**[2]    **Kibok Lee**[1]
[1]Yonsei University    [2]ETRI
[1]{taejun.lim, kibok}@yonsei.ac.kr   [2]jwhwang@etri.re.kr

## ABSTRACT

When continual test-time adaptation (TTA) persists over the long term, errors accumulate in the model and further cause it to predict only a few classes for all inputs, a phenomenon known as *model collapse*. Recent studies have explored reset strategies that completely erase these accumulated errors. However, their periodic resets lead to suboptimal adaptation, as they occur independently of the actual risk of collapse. Moreover, their full resets cause catastrophic loss of knowledge acquired over time, even though such knowledge could be beneficial in the future. To this end, we propose (1) an **A**daptive and **S**elective **R**eset (**ASR**) scheme that dynamically determines when and where to reset, (2) an importance-aware regularizer to recover essential knowledge lost due to reset, and (3) an on-the-fly adaptation adjustment scheme to enhance adaptability under challenging domain shifts. Extensive experiments across long-term TTA benchmarks demonstrate the effectiveness of our approach, particularly under challenging conditions. Our code is available at https://github.com/YonseiML/asr.

## 1 INTRODUCTION

Test-time adaptation (TTA) (Liang et al., 2020; Sun et al., 2020; Wang et al., 2021) aims to address the growing challenge of distribution shifts in real-world applications by enabling model adaptation at test time. Recently, TTA research has expanded to continual scenarios (Wang et al., 2022; Döbler et al., 2023), allowing models to adapt to a non-stationary stream of domains, where updates proceed continuously while errors accumulate over time. However, when domain shifts persist over the long term, these errors further result in *model collapse* (Niu et al., 2023; Shumailov et al., 2024), in which models converge to generate incorrect predictions concentrated on only a few classes for all inputs. To address this, recent studies have explored methods seeking to preserve knowledge from the source domain when adapting to target domains (Wang et al., 2022; Marsden et al., 2024; Press et al., 2023). A straightforward yet effective method involves periodically resetting model parameters to those of the source model (Press et al., 2023), which erases accumulated updates and errors, thereby rescuing the model from irreversible collapse. However, such a mechanism forces resets to depend on a single pre-defined reset interval across all situations, leading to too frequent or infrequent resets. Moreover, this completely erases the knowledge acquired during adaptation, thereby disrupting forward knowledge transfer within the continuously adapting model (Díaz-Rodríguez et al., 2018).

To this end, we propose an **A**daptive and **S**elective **R**eset (**ASR**) scheme that dynamically determines when and where to reset based on the concentration of predicted classes to estimate the risk of model collapse. We trigger a reset once the risk is deemed significant, and adjust its scope according to the severity of the risk. Several studies (Bai et al., 2021; Yang et al., 2024) showed that corruption from label noise begins at the end of the network. Since this corruption results in collapse, we prioritize layers closer to the output for reset. Fig. 1 illustrates how our ASR scheme differs from the aforesaid naive reset approach. In addition, we introduce an importance-aware regularizer to recover essential knowledge lost due to reset. We estimate parameter importance through newly formulated Fisher information. Based on this, parameters crucial for previous tasks are aligned with their accumulated state, which incorporates all prior target knowledge. Finally, we propose adjusting model adaptation on the fly based on domain discrepancy. We define prediction inconsistency to quantify this discrepancy, and then use it to update model hyperparameters via reparameterization, improving our adaptability under challenging domain shifts. Our contributions are as follows:

- We propose an *Adaptive and Selective Reset (ASR)* scheme that dynamically determines when and where to reset, effectively preventing model collapse while mitigating knowledge lost due to reset.

Figure 1: Illustrative comparison between a naive reset approach (RDumb; Press et al. (2023)) and our Adaptive and Selective Reset (ASR) based on the same model (ETA; Niu et al. (2022)). RDumb fully resets parameters at fixed intervals (e.g., every 1000 steps), whereas ASR dynamically decides when and where to reset, achieving more stable (i.e., less abrupt performance drop at each reset) and higher performance. Dotted vertical lines indicate when resets occur.

- Beyond the reset, we introduce an *importance-aware regularizer* to recover parameters that are inevitably reset but deemed crucial to prior tasks, and *on-the-fly adaptation adjustment* that updates model hyperparameters according to domain discrepancy to enhance adaptability.

- Extensive experimental results across various long-term TTA benchmarks demonstrate the effectiveness of our method. Remarkably, it yields a substantial 44.12% improvement over the state of the art on the challenging CCC-Hard (Press et al., 2023).

## 2 RELATED WORK

**Test-time adaptation.** TTA enables a model to adapt to unknown target environments without any target assumptions. Since true labels are unavailable at test time, early works have explored effective unsupervised adaptation (Kundu et al., 2020; Li et al., 2020; Liang et al., 2020). Initial TTA research proposed to adjust batch normalization statistics (Schneider et al., 2020; Mirza et al., 2022), which evolved toward integrating self-training approaches (Zhang et al., 2022; Goyal et al., 2022), such as entropy minimization, improving predictive confidence on target data (Wang et al., 2021), which has been developed to prevent wrong confidence intensification (Zhang et al., 2025a; Han et al., 2025).

**Continual test-time adaptation.** Self-training methods face a critical challenge in a non-stationary domain stream, where their performance gradually deteriorates over time with noisy pseudo-labeling repeated (Wang et al., 2022; Niu et al., 2023). It accumulates errors, enhancing predictive confidence in incorrect predictions, eventually leading them to converge to suboptimal solutions, a phenomenon known as model collapse (Niu et al., 2023; Shumailov et al., 2024). Several studies (Niu et al., 2023; Hoang et al., 2024) empirically demonstrated that once collapsed, a model assigns all inputs to a few dominant classes. CoTTA (Wang et al., 2022) addresses this issue by stabilizing its self-training scheme using augmentation-averaged pseudo-labels and preventing source knowledge forgetting via stochastic parameter restoration. To better handle error accumulation, recent research has explored reliable adaptation methods, such as adaptive learning rates (Park et al., 2024; Maharana et al., 2025) and adaptive loss functions (Liu et al., 2024a).

**Long-term test-time adaptation.** While effective at preventing collapse in standard continual TTA, existing methods encounter difficulties in more realistic settings such as gradual (Döbler et al., 2023) or smooth (Press et al., 2023) long-term domain shifts. To address this, ROID (Marsden et al., 2024) proposes a weight ensembling method that updates the current model by combining with a weighted pre-trained model. CMF (Lee & Chang, 2024) further extends this by updating the pre-trained model with the current model, inspired by the Kalman filter (Särkkä & Svensson, 2023). More aggressive methods have also been proposed, including resetting all parameters to their pre-trained state at every fixed step (Press et al., 2023). Others trigger resets only when extremely high predictive confidence (Niu et al., 2023) or severe distribution shifts from the source domain (Wang et al., 2024) are detected. Another line of research has developed regularization to restrict the deviation between the pre-trained and current parameters, such as weighting regularization via Fisher information (Niu et al., 2022) or adjusting the regularization coefficient based on parameter divergence from the pretrained state (Hoang et al., 2024). This coefficient can also be dynamically assigned for every single layer based on its depth (Yang et al., 2024) or its sensitivity to distribution shifts (Choi et al., 2022). Our work aligns with the emerging trend of *long-term TTA*, which considers more realistic settings with long-term domain shifts. It addresses the limitations of conventional reset methods that either reset too frequently or too infrequently and fully discard the knowledge acquired during adaptation. Specifically, our approach dynamically determines when and where to reset while recovering essential knowledge lost by reset.

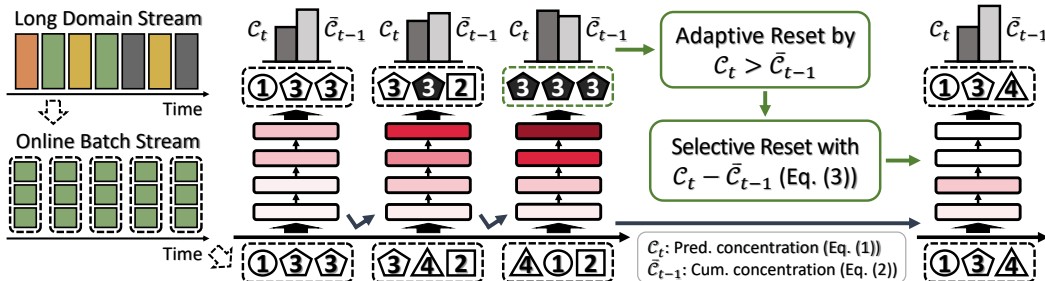

Figure 2: Overview of our Adaptive and Selective Reset (ASR) scheme, which compares prediction concentration $\mathcal{C}_t$ with its cumulative counterpart $\bar{\mathcal{C}}_{t-1}$ for each test batch from a long domain stream, triggers a reset when $\mathcal{C}_t > \bar{\mathcal{C}}_{t-1}$, indicating that the model is corrupted severely enough to collapse, and determines layers to reset based on $\mathcal{C}_t - \bar{\mathcal{C}}_{t-1}$, which reflects how severely the model is corrupted. On the upper side, icons inside dashed boxes, labeled with numbers, denote class labels. White icons represent correct predictions, while black icons represent incorrect predictions.

## 3 METHOD

### 3.1 PROBLEM DEFINITION

Given a pre-trained source model $f_{\theta_0}$, our goal is to improve its performance at test time over a long sequence of test domains without access to source data. A handful of test samples arrive in sequence and are then inaccessible once processed via the model. At step $t$, the current model $f_{\theta_{t-1}}$ is given a test sample $x_t^i$ and generates a prediction $\hat{y}_t^i = \sigma(f_{\theta_{t-1}}(x_t^i))$, where $f_*$ yields logit outputs and $\sigma$ is the softmax function. The model is evaluated using its predictions $\hat{y}_t$, and is then adapted as $\theta_{t-1} \rightarrow \theta_t$ using unsupervised objective functions. Moreover, we aim for stable adaptation without performance degradation under collapse-prone scenarios such as perpetually changing or cyclically recurring domain streams. To this end, we address the limitations of existing reset approaches, such as suboptimal reset timing and catastrophic loss of knowledge, through the following three main components: (1) Adaptive and Selective Reset (ASR; illustrated in Fig. 2), (2) importance-aware knowledge recovery, and (3) on-the-fly adaptation adjustment.

### 3.2 MOTIVATION

We observed that RDumb's periodic reset (Press et al., 2023) is only suitable for standard TTA benchmarks where domain shifts occur at regular intervals. However, real-world settings do not follow the fixed shift schedule, and their shift timing can vary significantly. RDumb can reset either too early or too late, misaligned with the actual risk of collapse, leading to suboptimal or unstable adaptation. Fig. 1 shows that RDumb suffers from a substantial performance drop after each reset. This is primarily due to the full reset, which discards all knowledge acquired so far, causing a significant delay to recover the pre-reset performance. These observations motivate our reset scheme, which resets only when a model is at the risk of collapse and reduces knowledge lost by reset. We support our motivation by quantifying the performance drop and the recovery delay in Appendix A.1.

### 3.3 ADAPTIVE AND SELECTIVE RESET

**When to reset.** We introduce an adaptive reset scheme that triggers a reset only when a high risk of collapse is detected. To achieve this, we define prediction concentration $\mathcal{C}_t$, leveraging the notion that entropy reflects the uniformity of a distribution, where `Softmax(Mean(Logits))` serves as the underlying measure, as follows:

$$\mathcal{C}_t = \sum_{c=1}^{C} \hat{p}_{t_c} \log(\hat{p}_{t_c}) \quad \text{where} \quad \hat{p}_t = \sigma \left( \frac{1}{|\mathcal{B}_t|} \sum_{i=1}^{|\mathcal{B}_t|} f_{\theta_{t-1}}(x_t^i) \right). \tag{1}$$

$C$ is the total number of classes, and $\hat{p}_{t_c}$ indicates the probability of the $c$-th class in $\hat{p}_t$, obtained by applying the softmax function $\sigma$ to the average logits of the batch $\mathcal{B}_t$ at time step $t$. Note that $\mathcal{C}_t$ quantifies prediction concentration; hence, a large $\mathcal{C}_t$ indicates low prediction diversity and a high risk of collapse. Although we can measure the concentration of predicted classes, it remains unclear *when it is high enough to suggest that the model is on the verge of collapse*. We argue that when the concentration $\mathcal{C}_t$ deviates from its long-term normal behavior, it can be regarded as an indication that collapse is likely to emerge, and define cumulative concentration $\bar{\mathcal{C}}_t$, computed via exponential

moving average (EMA), as follows:

$$\bar{\mathcal{C}}_t = \mu_{\mathcal{C}} \cdot \bar{\mathcal{C}}_{t-1} + (1 - \mu_{\mathcal{C}}) \cdot \mathcal{C}_t, \tag{2}$$

where $\mu_{\mathcal{C}}$ is the momentum coefficient, and $\bar{\mathcal{C}}_0$ is initialized as $-\log(\alpha_0 \cdot C)$ using a pre-defined $\alpha_0$. We compare $\mathcal{C}_t$ with its cumulative counterpart $\bar{\mathcal{C}}_{t-1}$ to determine whether to trigger a reset at each step $t$. $\bar{\mathcal{C}}_{t-1}$ is reinitialized as $-\log(\alpha_0 \cdot C)$ if the model is reset; otherwise it is updated via Eq. (2). We choose $\alpha_0$ such that the initial cumulative value is sufficiently larger than $\mathcal{C}_t$, preventing premature resets in early stages and providing sufficient warm-up time for $\bar{\mathcal{C}}_{t-1}$ to form a reliable estimate of the long-term normal behavior of $\mathcal{C}_t$ (see the top right of Fig. 2). We trigger a reset immediately when $\mathcal{C}_t > \bar{\mathcal{C}}_{t-1}$ is detected to prevent the accumulation of corrupted Fisher information (Sec. 3.4).

To demonstrate that $\mathcal{C}_t$ is effective in detecting a high collapse risk, we evaluate its correlation with accuracy on the CCC-Hard dataset with the ETA model, as shown in Fig. 3, where lower accuracy indicates a higher risk of collapse. We find a strong Pearson correlation of 0.88, confirming the reliability of $\mathcal{C}_t$. Detailed settings and additional analysis are provided in Appendix A.2.

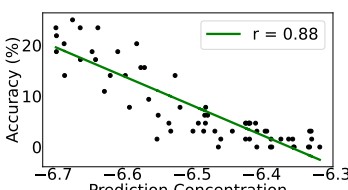

Figure 3: Corr. of $\mathcal{C}_t$ and Acc.

**Where to reset.** A critical drawback of reset is catastrophic loss of knowledge accumulated over time. To alleviate this, we take advantage of the hierarchical nature of deep neural networks. In early collapse, layers closer to the input tend to be more robust to corruption than those closer to the output, since the corruption from label noise begins at the end of the network (Bai et al., 2021; Yang et al., 2024). Inspired by this, we introduce a selective reset scheme that determines which layers to reset, prioritizing those closer to the output. As corruption intensifies during collapse, the reset should be applied to more layers in proportion to the estimated collapse severity. We quantify this severity by measuring how far our concentration metric $\mathcal{C}_t$ deviates from its long-term normal behavior, i.e., $\mathcal{C}_t - \bar{\mathcal{C}}_{t-1}$. We define a reset proportion $r_t$ that controls how many layers are reset, as follows:

$$r_t = r_0 + \lambda_r \cdot (\mathcal{C}_t - \bar{\mathcal{C}}_{t-1}). \tag{3}$$

$r_0$ and $\lambda_r$ are pre-defined as the minimum reset proportion and the reset proportion scaling factor. $r_t$ is always greater than $r_0$ since reset is triggered only when $\mathcal{C}_t > \bar{\mathcal{C}}_{t-1}$, and $r_t$ is subject to an upper bound of 1.0 (i.e., 100%). The layers to be reset are determined starting from the output, such that the last $r_t$ proportion of layers are reset, while the remaining $1 - r_t$ are preserved[1].

## 3.4 IMPORTANCE-AWARE KNOWLEDGE RECOVERY

Although we mitigate catastrophic loss of knowledge due to reset, some highly important knowledge is still inevitably erased. To further address this issue, we introduce an importance-aware regularizer designed to recover essential knowledge lost. At every iteration, we accumulate learnable parameters and their importance matrices computed via Fisher information (Kirkpatrick et al., 2017; Zenke et al., 2017; Schwarz et al., 2018). We then apply the regularizer to strongly guide parameters deemed significant in previous tasks toward alignment with the accumulated ones, as follows:

$$\mathcal{L}(\mathcal{B}_t; \theta_{t-1}) = \mathcal{L}_u(\mathcal{B}_t; \theta_{t-1}) + \lambda_{\mathcal{F}} \sum_{i=1}^{|\theta_{t-1}|} \bar{\mathcal{F}}^i \left( \theta_{t-1}^i - \bar{\theta}^i \right)^2, \tag{4}$$

where $\mathcal{L}$ and $\mathcal{L}_u$ denote total and unsupervised losses, $\bar{\mathcal{F}}^i$ and $\bar{\theta}^i$ are the $i$-th accumulated Fisher matrix and accumulated parameter, $\theta_{t-1}^i$ is the $i$-th learnable parameter of $\theta_{t-1}$, and $\lambda_{\mathcal{F}}$ is the regularization coefficient.

In the accumulation phase, the following dilemma arises: *While parameters increasingly align with the current domain over time, their proximity to reset makes them more vulnerable to corruption.* By *proximity to reset*, we mean that over a prolonged period of adaptation, errors accumulate substantially, compromising model integrity and increasing the necessity of a reset. We further provide empirical evidence to support this in Appendix A.3. EMA is a commonly used accumulation technique, but it is not an ideal choice here, as it inherently prioritizes recent information. To address this, we introduce a hybrid accumulation scheme that combines cumulative moving average (CMA) with EMA. At each iteration $t$, CMA accumulates learnable parameters and their Fisher matrices equally. EMA then aggregates the CMA-accumulated values at every reset-triggered point, after which CMA values are re-initialized to zero (see Algorithm 1). The EMA-accumulated parameters and Fisher matrices correspond to $\bar{\theta}$ and $\bar{\mathcal{F}}$ in Eq. (4). More details on this scheme are provided in

---

[1]For example, with 15 layers and $r_t = 0.5$, the 8 deepest layers are reset (rounded to the nearest integer).

Appendix A.4, and its computational efficiency is analyzed in Appendix A.5. Moreover, we prove that the proposed regularizer effectively recovers essential knowledge by providing both theoretical and empirical evidence in Appendix A.6.

### 3.5 ON-THE-FLY ADAPTATION ADJUSTMENT

While we assume domain-evolving settings, we do not explicitly model how domain evolution unfolds when designing our method. Under challenging domain evolution, memorization of previously seen target domains becomes important for effective adaptation. However, our adaptability may struggle to keep pace, since reset discards what the model has learned so far during adaptation. In this case, strong guidance from the Fisher regularizer (Sec. 3.4) becomes necessary to fully exploit essential knowledge of prior target domains. Moreover, such a challenging case amplifies the source–target discrepancy, thereby worsening label noise. Pseudo-labels tend to degenerate into nearly random assignments (Semenova et al., 2023), leading to unreliable estimation of $\mathcal{C}_t$ in Eq. (1). To address this, we propose adjusting model adaptation on the fly based on domain discrepancy. We define prediction inconsistency $\phi_t$ to quantify domain discrepancy, as follows:

$$\phi_t = \frac{1}{|\mathcal{B}_t|} \sum_{i=1}^{|\mathcal{B}_t|} \mathbb{I}\left(\arg\max(\breve{y}_t^i) \neq \arg\max(\hat{y}_t^i)\right), \tag{5}$$

where $\mathbb{I}$ denotes the indicator function, and $\breve{y}_t^i$ and $\hat{y}_t^i$ are the softmax outputs of the source model $f_{\theta_0}$ and the current model $f_{\theta_{t-1}}$ for sample $i$ at step $t$, respectively. A larger $\phi_t$ (closer to 1) indicates greater domain discrepancy. Based on this, we adjust adaptation on the fly by reparameterizing the regularization coefficient $\lambda_{\mathcal{F}}$ in Eq. (4) and the momentum coefficient $\mu_{\mathcal{C}}$ in Eq. (2) as follows:

$$\lambda_{\mathcal{F}} = \lambda_0 \cdot \phi_t^2, \tag{6}$$

$$\mu_{\mathcal{C}} = \mu_0 \cdot \phi_t + 1 - \mu_0, \tag{7}$$

where $\lambda_0$ and $\mu_0$ are pre-defined. As $\phi_t$ increases, $\lambda_{\mathcal{F}}$ increases within $[0, \lambda_0]$ for stronger regularization in Eq. (4), and $\mu_{\mathcal{C}}$ increases linearly within $[1 - \mu_0, 1]$ to reduce the update effect of $\bar{\mathcal{C}}_{t-1}$ in Eq. (2). If $\lambda_0 = 0$, no knowledge recovery occurs; if $\mu_0 = 0$, no update of $\bar{\mathcal{C}}_{t-1}$ occurs.

## 4 EXPERIMENTS

### 4.1 SETUP

**Datasets.** As discussed in Press et al. (2023), standard TTA benchmarks are inappropriate for evaluating the stability of continual TTA methods under long-term adaptation, where models are highly prone to collapse. To address this, we leverage two recently introduced benchmarks (1, 2) specifically designed for collapse and adapt two widely used TTA benchmarks (3, 4) to better reflect long-term collapse-prone scenarios. We conduct experiments on the following four benchmarks: 1) **Continually Changing Corruptions (CCC)** (Press et al., 2023) is a benchmark systematically processed from ImageNet-C (Hendrycks & Dietterich, 2019). This assumes smooth domain shifts over the long term, where one gradually fades while another emerges, with the two overlapping. It is also divided into three adaptation difficulty levels (Easy / Medium / Hard), each incorporating three corruption orderings and three corruption evolving speeds, resulting in nine variations per level. 2) **Concatenated ImageNet-C (CIN-C)** is an extended version of ImageNet-C, containing 50K images per corruption ($10\times$ more than the original), where 15 corruption types under the most severe condition (level 5) are in sequence. It is often used by several works (Wang et al., 2022; Niu et al., 2022; Gong et al., 2022; Brahma & Rai, 2023) to demonstrate their stability, while comparing with Tent (Wang et al., 2021) that fails to avoid collapse. Moreover, 3) **ImageNet-C (IN-C)** and 4) **ImageNet-D109 (IN-D109)** (Peng et al., 2019), both widely used TTA benchmarks, are adapted to reflect model collapse, following prior works (Press et al., 2023; Hoang et al., 2024). IN-C cyclically repeats the sequence of corruptions 20 times, including only four types on which the source model reaches less than 10% accuracy. IN-D109 is also processed in the same way as IN-C, but selects four domain types based on less than 50% accuracy. Refer to Appendix B for further dataset details.

**Baselines.** We compare our approach with state-of-the-art continual TTA approaches. We categorize them into two groups based on whether they include any modules to prevent collapse. The first group without any safeguard against collapse includes ETA (Niu et al., 2022), RoTTA (Yuan et al., 2023), ViDA (Liu et al., 2024b), C-MAE (Liu et al., 2024a), PALM (Maharana et al., 2025), COME (Zhang et al., 2025a) and REM (Han et al., 2025). The second group involving collapse-prevention modules includes EATA (Niu et al., 2022), CoTTA (Wang et al., 2022), RDumb (Press et al., 2023), SAR (Niu et al., 2023), ROID (Marsden et al., 2024), CMF (Lee & Chang, 2024), PeTTA (Hoang et al., 2024)

and ReservoirTTA (Vray et al., 2025). Among all these baselines, we particularly compare with add-on methods including RDumb, COME and ReservoirTTA for a reliable evaluation of our reset scheme. Specifically, we apply these add-on methods to the commonly applicable baselines (i.e., ETA, EATA and ROID) and compare them with our approach in Table 1.

**Implementation details.** We implement all baselines in a unified repository (Marsden et al., 2024) using `PyTorch` (Paszke et al., 2019) and obtain all reported results via code replication to ensure a fair and consistent comparison. Experiments are conducted on ResNet-50 (He et al., 2016), provided by either `torchvision` or `RobustBench` (Croce et al., 2021), as well as ViT-B-16 (Dosovitskiy et al., 2021) to evaluate the generalization of our approach. As for our implementation, we basically follow our TTA baselines. Specifically, $L_u$ in Eq. (4) is formulated based on what they define as the final loss. We determine hyperparameters using only 5% of a single holdout split in CCC-Hard and use the determined ones across all datasets. We assess robustness to hyperparameter changes across all CCC levels in Appendix A.7. For the CCC-based analysis, we consistently use a single split with speed 2000 and seed 44. More details on our implementation are available in Appendix A.8.

## 4.2 MAIN RESULTS

**a) CCC.** We evaluate our reset scheme by comparing it with add-on methods that share the same TTA baselines, using the CCC benchmark in Table 1. Our approach achieves the best average performance across all baselines, with particularly large improvements on CCC-Hard. In contrast, COME performs poorly due to the absence of an anti-collapse mechanism. While the TTA baselines in Table 1 are built with entropy minimization (EM), we further report the results using objectives other than EM to validate the general utility of our method in Table 2. As any TTA model trains itself using pseudo-labels derived from its own noisy predictions, it eventually collapses (Niu et al., 2023), thus making our reset effective regardless of the underlying objective.

| Method | Easy | Medium | Hard | Mean |
|---|---|---|---|---|
| ETA (ICML'22) | 43.24±1.0 | 19.03±6.9 | 0.32±0.1 | 20.86 |
| + RDumb (NeurIPS'23) | 49.47±0.8 | 39.42±1.5 | 9.77±1.8 | 32.89 |
| + COME (ICLR'25) | 6.34±4.3 | 0.56±0.1 | 0.14±0.0 | 2.35 |
| + ReservoirTTA (NeurIPS'25) | 48.86±0.6 | 42.03±0.9 | 1.15±0.8 | 30.68 |
| + ASR (Ours) | 51.20±0.8 | 41.88±1.6 | 17.10±2.1 | **36.73** |
| EATA (ICML'22) | 49.52±0.9 | 39.19±1.7 | 0.82±0.4 | 29.84 |
| + RDumb (NeurIPS'23) | 49.95±0.8 | 39.93±1.4 | 11.09±1.8 | 33.66 |
| + COME (ICLR'25) | 46.67±3.3 | 36.63±1.6 | 0.80±0.4 | 28.03 |
| + ReservoirTTA (NeurIPS'25) | 51.51±0.4 | 42.03±0.9 | 7.58±3.8 | 33.71 |
| + ASR (Ours) | 50.40±0.9 | 40.78±1.7 | 16.22±1.4 | **35.80** |
| ROID (WACV'24) | 49.88±0.8 | 40.47±1.4 | 12.48±2.6 | 34.28 |
| + RDumb (NeurIPS'23) | 49.69±0.8 | 40.05±1.4 | 15.41±1.5 | 35.05 |
| + COME (ICLR'25) | 0.12±0.0 | 0.13±0.0 | 0.12±0.0 | 0.12 |
| + ReservoirTTA (NeurIPS'25) | 50.86±0.3 | 40.89±0.8 | 13.71±1.0 | 35.15 |
| + ASR (Ours) | 51.41±0.8 | 42.80±1.5 | 22.21±1.2 | **38.81** |

Table 1: Evaluation of our reset via a comparison with add-on methods across the commonly applicable TTA baselines using accuracy (%) on the CCC benchmark.

Nevertheless, as shown by ROID, self-entropy-based objectives that strongly sharpen confidence tend to be more effective because they significantly influence class concentration. Furthermore, CCC reveals the limitations of continual TTA methods and the effectiveness of our method under long-term collapse-prone settings in Table 4. Most baselines fail or collapse on CCC-Hard, whereas RDumb prevents collapse but degrades ROID on other levels. In contrast, our approach achieves consistent improvements and yields the largest gains on CCC-Hard, surpassing the second best by 44.12% (i.e., 15.41 → 22.21).

| Method | Easy | Medium | Hard | Mean |
|---|---|---|---|---|
| ROID (WACV'24) | 49.88±0.8 | 40.47±1.4 | 12.48±2.6 | 34.28 |
| + RDumb (NeurIPS'23) | 49.69±0.8 | 40.05±1.4 | 15.41±1.5 | 35.05 |
| + ASR (Ours) | 51.41±0.8 | 42.80±1.5 | 22.21±1.2 | **38.81** |
| CMF (ICLR'24) | 49.31±0.9 | 40.61±1.6 | 0.89±0.6 | 30.27 |
| + RDumb (NeurIPS'23) | 49.00±0.8 | 39.88±1.5 | 15.04±1.9 | 34.64 |
| + ASR (Ours) | 51.04±0.9 | 42.03±1.6 | 21.04±1.1 | **38.04** |
| PeTTA (NeurIPS'24) | 36.89±2.2 | 22.64±2.8 | 6.00±0.8 | 21.84 |
| + RDumb (NeurIPS'23) | 36.81±2.2 | 22.39±2.6 | 5.81±0.6 | 21.67 |
| + ASR (Ours) | 39.36±2.3 | 23.51±2.5 | 7.27±0.6 | **23.38** |

Table 2: Evaluation of our general utility under various objectives including EM and non-EM.

| Method | Easy | Medium | Hard | Mean |
|---|---|---|---|---|
| Source | 54.92±0.2 | 41.74±0.6 | 14.83±0.6 | 37.16 |
| CMF (ICLR'24) | 61.52±0.7 | 51.50±6.3 | 1.79±1.7 | 38.27 |
| C-MAE (CVPR'24) | 51.15±2.3 | 43.48±3.9 | 26.92±2.3 | 40.52 |
| REM (ICML'25) | 66.16±0.3 | 57.99±0.9 | 10.97±9.9 | 45.04 |
| ETA (ICML'22) | 45.07±10.4 | 33.71±4.6 | 1.22±0.5 | 26.67 |
| + RDumb | 59.99±0.6 | 50.50±1.4 | 23.27±1.1 | 44.58 |
| + ASR (Ours) | 60.58±0.7 | 51.63±1.6 | 24.45±0.9 | 45.55 |
| ROID (WACV'24) | 60.85±0.7 | 52.19±1.3 | 14.30±8.2 | 42.45 |
| + RDumb (NeurIPS'23) | 60.60±0.7 | 51.68±1.3 | 25.72±1.4 | 46.00 |
| + ASR (Ours) | 61.48±0.7 | 53.55±1.3 | 28.09±0.6 | **47.71** |

Table 3: Acc. (%) comparison on ViT.

We further evaluate the generalization of our method using ViT-B-16 in Table 3. We compare only to baselines that report ViT performance in their original papers. While CMF (Lee & Chang, 2024) and REM (Han et al., 2025) achieve strong results on CCC-Easy/-Medium, they eventually suffer from collapse on CCC-Hard. However, C-MAE (Liu et al., 2024a) validates its efficacy on CCC-

| Method | CCC | | | CIN-C | | IN-C | | IN-D109 | |
|---|---|---|---|---|---|---|---|---|---|
| | Easy | Medium | Hard | i.i.d. | non-i.i.d. | Visit 1 / 20 | Mean | Visit 1 / 20 | Mean |
| Source | 33.89±0.2 | 16.87±0.2 | 1.27±0.0 | 18.01±0.0 | 18.01±0.0 | 3.08 / 3.08 | 3.08±0.0 | 32.52 / 32.52 | 32.52±0.0 |
| ETA (ICML'22) | 43.24±1.0 | 19.03±6.9 | 0.32±0.1 | 43.61±0.4 | 43.63±0.4 | 30.64 / 35.80 | 35.88±1.2 | 41.24 / 34.21 | 37.22±2.1 |
| RoTTA (CVPR'23) | 2.28±0.6 | 1.76±0.6 | 0.69±0.2 | 29.05±2.0 | 29.71±1.7 | 12.45 / 12.96 | 17.60±2.8 | 39.89 / 34.34 | 40.61±3.1 |
| ViDA (ICLR'24) | 12.68±0.8 | 5.75±0.5 | 0.42±0.0 | 17.76±0.1 | 17.76±0.1 | 3.09 / 2.84 | 2.99±0.1 | 0.01 / 0.01 | 0.01±0.0 |
| PALM (AAAI'25) | 1.56±0.2 | 0.74±0.3 | 0.13±0.0 | 12.69±6.3 | 12.08±6.1 | 24.66 / 30.98 | 30.70±1.4 | 13.86 / 1.42 | 2.06±2.7 |
| EATA (ICML'22) | 49.52±0.9 | 39.19±1.7 | 0.82±0.4 | 47.81±0.2 | 47.54±0.2 | 31.31 / 36.35 | 36.32±1.2 | 41.62 / 41.32 | 41.61±0.3 |
| + COME (ICLR'25) | 46.67±3.3 | 36.63±1.6 | 0.80±0.4 | 44.14±0.3 | 44.09±0.3 | 30.20 / 32.06 | 33.02±1.1 | 42.94 / 44.91 | 45.11±0.6 |
| CoTTA (CVPR'22) | 17.50±1.0 | 9.83±0.9 | 1.52±0.5 | 35.51±2.6 | 35.29±2.4 | 18.78 / 37.22 | 34.39±4.8 | 41.76 / 40.55 | 43.91±2.1 |
| SAR (ICLR'23) | 37.94±1.2 | 22.25±1.9 | 2.03±0.5 | 40.35±1.8 | 40.07±0.6 | 24.38 / 34.93 | 34.09±2.4 | 40.86 / 33.11 | 39.09±3.4 |
| CMF (ICLR'24) | 49.31±0.9 | 40.61±1.6 | 0.89±0.6 | 48.61±0.1 | 48.28±0.2 | 35.07 / 39.40 | 39.35±1.0 | 44.69 / 45.46 | 45.25±0.3 |
| PeTTA (NeurIPS'24) | 36.89±2.2 | 22.64±2.8 | 6.00±0.8 | 31.55±0.1 | 31.61±0.1 | 11.91 / 12.40 | 12.65±0.3 | 39.56 / 42.69 | 42.76±0.8 |
| ROID (WACV'24) | 49.88±0.8 | 40.47±1.4 | 12.48±2.6 | 48.58±0.1 | 48.25±0.1 | 35.32 / 38.02 | 37.96±0.6 | 46.02 / 46.17 | 46.16±0.1 |
| + RDumb (NeurIPS'23) | 49.69±0.8 | 40.05±1.4 | 15.41±1.5 | 48.00±0.1 | 47.67±0.1 | 35.60 / 35.75 | 37.18±1.2 | 46.07 / 45.62 | 45.99±0.2 |
| + ReservoirTTA (NeurIPS'25) | 50.86±0.3 | 40.89±0.8 | 13.71±1.0 | 47.34±0.1 | 46.96±0.1 | 31.54 / 37.09 | 36.84±1.2 | 44.71 / 45.76 | 45.61±0.3 |
| + ASR (Ours) | 51.41±0.8 | 42.80±1.5 | 22.21±1.2 | 49.50±0.2 | 49.14±0.2 | 35.66 / 42.96 | 41.56±1.7 | 46.13 / 46.32 | 46.49±0.1 |

Table 4: Comparison with state-of-the-art continual TTA approaches across four datasets using accuracy (%). Results for each level of CCC (Easy / Medium / Hard) are averaged over nine variations, considering three different corruption orderings and three corruption evolving speeds. CIN-C results are averaged over ten runs. In the *non-i.i.d.* setting, we use a Dirichlet parameter $\delta = 0.1$, following prior works (Gong et al., 2022; Yuan et al., 2023). For IN-C and IN-D109, we report averages at the first and last (20th) visits, as well as overall averages across all visits. Gray denotes model collapse, indicating performance worse than that of the source model (Press et al., 2023).

Hard but does not generalize well to the other levels. In contrast, our approach not only maintains strong performance on CCC-Hard but also achieves the best average performance.

**b) CIN-C.** Table 4 presents the results on CIN-C using the average accuracy over ten permutations, where corruption types are randomly shuffled. Baselines that achieve stable adaptation on CCC also perform well on CIN-C. In particular, weight ensembling approaches (Marsden et al., 2024; Lee & Chang, 2024) demonstrate their effectiveness by achieving the top ranks among baselines. However, our method still yields the best performance on CIN-C, even though it is less prone to collapse. Most existing works assume label-i.i.d. testing environments, but this assumption does not always hold in real-world applications. Recently, attention has been increasingly given to non-i.i.d. settings where labels are temporally correlated. Following Gong et al. (2022); Yuan et al. (2023), we use a Dirichlet parameter $\delta = 0.1$ to control the test class distribution. Table 4 shows that our approach achieves the best adaptation performance, indicating that our reset scheme can properly determine the timing and scope of reset even under non-i.i.d. settings. Further non-i.i.d. results are shown in Fig. 7.

**c) IN-C.** We report the average accuracy over corruptions at the first and last (20th) visits, as well as the overall average across all visits for IN-C in Table 4. Most baselines succeed in avoiding collapse and achieve substantial improvements over the source model. Using IN-C, our approach proves that it is effective not only in preventing collapse but also in enhancing model adaptability by getting the best performance consistently across the first, last and overall visits.

**d) IN-D109.** The results for IN-D109 are reported in the same manner as IN-C, as shown in Table 4. Several baselines experience performance degradation when comparing Visit 1 and 20. This implies early collapse that might be due to the reduced number of classes. IN-D109 has only 109 classes, roughly $10\times$ fewer than other datasets. Consequently, a skewed prediction distribution is more easily observed than in other datasets. However, our method still yields stable and superior performance.

## 4.3 ABLATION STUDY

We ablate each component from our approach to validate its individual effectiveness. Table 5 shows that dynamically determining when and where to reset is the most significant factor, as reflected in the top two ablated results. First, we replace our adaptive reset with a fixed-interval reset using $T = 20000$, where $\mu_0$ is omitted since $\mathcal{C}_t$ in Eq. (1) is no longer computed. Second, we ablate our selective reset by applying full-parameter reset with $r_t = 1.0$ in

| $\mathcal{C}_t$ (1) | $r_t$ (3) | $\bar{\mathcal{F}}$ (4) | $\lambda_0$ (6) | $\mu_0$ (7) | Easy | Medium | Hard | Mean |
|---|---|---|---|---|---|---|---|---|
| ✗ | ✗ | ✗ | ✗ | ✗ | 49.74 | 40.19 | 11.81 | 33.91 |
| ✗ | ✓ | ✓ | ✓ | ✗ | 49.83 | 40.58 | 17.16 | 35.86 |
| ✗ | ✓ | ✗ | ✓ | ✓ | 49.83 | 40.35 | 15.99 | 35.39 |
| ✓ | ✓ | ✗ | ✓ | ✓ | 51.04 | 42.19 | 20.18 | 37.80 |
| ✓ | ✓ | ✓ | ✗ | ✓ | 51.07 | 42.33 | 20.27 | 37.89 |
| ✓ | ✓ | ✓ | ✓ | ✗ | 50.82 | 41.86 | 20.70 | 37.79 |
| ✓ | ✓ | ✓ | ✓ | ✓ | 51.19 | 42.42 | 21.36 | 38.32 |

Table 5: Effect of components in ASR on ROID .

Eq. (3). The remaining components in Eqs. (4), (6), and (7) individually have limited impact but yield meaningful gains when combined. For ablated variants that remove the adaptive coefficients, we fix the corresponding hyperparameters to constant values. When $\lambda_0$ is removed, $\lambda_{\mathcal{F}}$ is set to 5.0, and when $\mu_0$ is removed, $\mu_{\mathcal{C}}$ is set to 0.995. More ablation study results are provided in Appendix C.

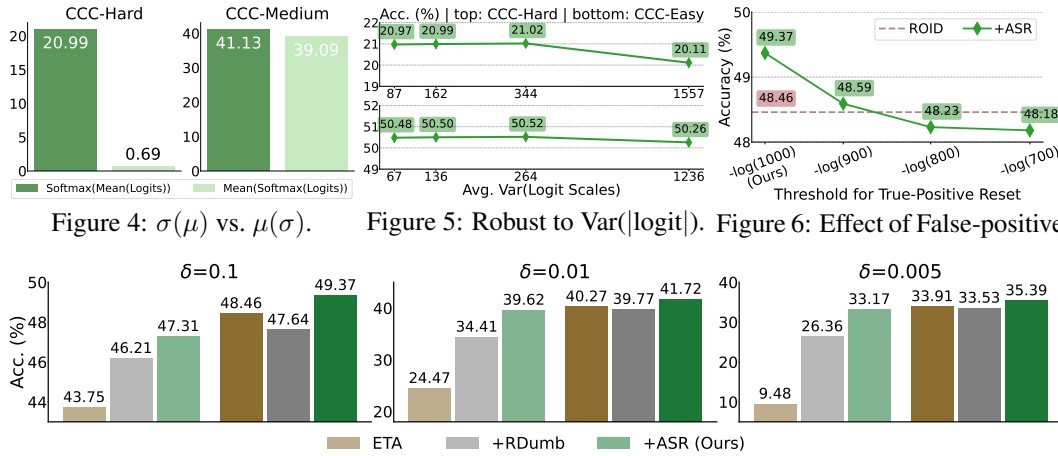

Figure 4: $\sigma(\mu)$ vs. $\mu(\sigma)$.    Figure 5: Robust to Var($|$logit$|$).   Figure 6: Effect of False-positive

Figure 7: Comparison of ETA / ROID and their variants with RDumb and ASR over different Dirichlet parameters $\delta$ on *non-i.i.d.* CIN-C. The lower the $\delta$, the more imbalanced the label distribution.

## 4.4 ANALYSIS

**a) Empirical analysis on `Softmax(Mean(Logits))`.** Model collapse arises from errors accumulated over a long period of time, which severely degrades model performance. A key characteristic of collapse is that the model predicts only a few classes for all inputs. However, detecting collapse is non-trivial. This is because true labels are inaccessible at test time, making it impossible to determine whether the model produces skewed predictions due to collapse or whether the data itself contains skewed classes. As discussed in Sec. 4.4(b), we find that accurately detecting true collapse is not strictly necessary. Even when collapse is falsely detected, triggering a reset based on prediction bias still facilitates effective model adaptation. Accordingly, we focus on analyzing how to effectively measure bias in the predictive distribution. Although `Mean(Softmax(Logits))` would be one of the most straightforward metrics for measuring prediction bias, we instead employ `Softmax(Mean(Logits))`. By being more sensitive to large-magnitude logits, this approach ensures that the dominance of high-confidence predictions (Wei et al., 2022)—the primary drivers of model updates and collapse—is preserved and not diluted by lower-confidence predictions. This provides a clearer signal of prediction bias, enabling more reliable detection of model collapse. Fig. 4 shows that `Softmax(Mean(Logits))` maintains stable adaptation even under highly collapse-prone conditions, whereas `Mean(Softmax(Logits))` fails, achieving only 0.69% accuracy on CCC-Hard. Moreover, we empirically show that large variance in logit magnitudes across samples within a batch does not pose a significant issue for `Softmax(Mean(Logits))`. This is important because it demonstrates that `Softmax(Mean(Logits))` is not dominated by a few samples with extremely large logits. We artificially manipulate logit-magnitude variance through a per-sample transformation and evaluate the robustness of `Softmax(Mean(Logits))` under increasingly extreme variance, as shown in Fig. 5. Even when the variance increases by more than 15×, accuracy remains nearly unchanged (<1%p on CCC-Hard and <0.3%p on CCC-Easy). For this experiment, we use a single split (transition speed 1000; random seed 43) with ROID (Marsden et al., 2024).

**b) Risk of false-positive reset.** One may question *"whether our approach still works well under label imbalance, where predictions are normally highly skewed."* Imbalanced priors, however, do not disrupt our method. As model predictions become more skewed, $\bar{\mathcal{C}}_{t-1}$—the moving average of past prediction bias—increases accordingly, providing a baseline. A high risk of collapse is then detected only when $\mathcal{C}_t$ is significantly higher than this baseline. Thus, $\bar{\mathcal{C}}_{t-1}$ serves as a reliable reference, making our reset mechanism robust against label imbalance, as empirically shown in Fig. 7 under various imbalanced settings. A related question is *"whether our approach remains effective when $\mathcal{C}_t$ temporarily spikes due to extremely label-imbalanced inputs under i.i.d. class priors."* We argue that a reset is generally desirable whenever $\mathcal{C}_t$—the current prediction bias—becomes high, since any high $\mathcal{C}_t$ produces biased update signals that can lead the model to converge to suboptimal solutions. To test this, we conduct a controlled experiment using a single batch with uniformly distributed labels, ensuring that any triggered reset is a *true-positive*. For each reset, $\mathcal{C}_t$ is computed for the batch as in Eq. (1) and compared against a pre-defined threshold, initialized as described below Eq. (2). Raising the threshold increases true-positive resets while reducing false-positive resets

| CCC-Easy | Original | Gain (%) | Modified | Gain (%) |
|---|---|---|---|---|
| ETA | 43.46 | - | 43.17 | - |
| + RDumb | 49.53 | +13.9% | 47.36 | +9.7% |
| + ASR (Ours) | 51.27 | +17.9% | 51.15 | +18.4% |
| ROID | 49.95 | - | 49.54 | - |
| + RDumb | 49.76 | -0.3% | 49.33 | -0.4% |
| + ASR (Ours) | 51.47 | +3.0% | 51.46 | +3.8% |

Table 6: Acc. (%) of original and modified CCC-Easy using seed 43. Gains (%) are relative to each corresponding baseline (e.g., ETA or ROID).

| CCC-Hard | Original | Gain (%) | Modified | Gain (%) |
|---|---|---|---|---|
| ETA | 0.41 | - | 1.83 | - |
| + RDumb | 9.46 | +2207% | 11.88 | +549% |
| + ASR (Ours) | 15.95 | +3790% | 17.61 | +862% |
| ROID | 9.63 | - | 16.51 | - |
| + RDumb | 14.03 | +45.6% | 15.99 | -3.1% |
| + ASR (Ours) | 21.22 | +120% | 21.56 | +30.5% |

Table 7: Acc. (%) of original and modified CCC-Hard using seed 43. Gains (%) are relative to each corresponding baseline (e.g., ETA or ROID).

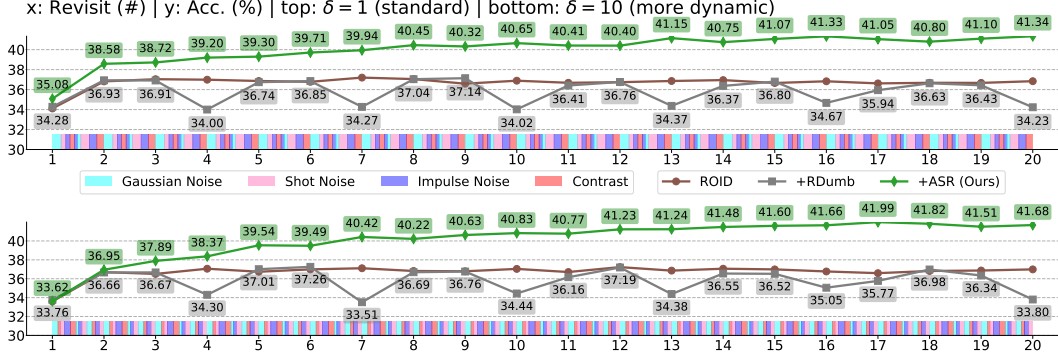

Figure 8: Performance comparison over revisits on IN-C with CDC settings. Domains dynamically change over time, controlled by $\delta$, and are color-coded to visualize their dynamics.

caused by temporarily high $\mathcal{C}_t$ in a correctly adapting model. Fig. 6 shows that encouraging false-positive resets by lowering the threshold improves performance, thereby confirming that interrupting biased updates—even when the model appears to adapt correctly—helps maintain robust long-term adaptation. For this experiment, we use a single split of CIN-C, corresponding to the first split in Table 32, with ROID as our baseline.

**c) Dynamically changing corruptions: A variant of CCC.** As discussed in Sec. 3.2, real-world domain shifts rarely follow a fixed schedule. Nonetheless, we adopt fixed domain-shift intervals in our benchmarks to maintain consistency with previous studies and enable fair comparisons. To further evaluate robustness under more realistic conditions, we introduce a modified CCC variant, *dynamically changing corruptions*, where the duration of each corruption is randomly sampled from {1000, 2000, 5000} batches. Unlike the original CCC, where each corruption persists for a fixed length, our variant introduces a stochastic transition schedule that more closely reflects the unpredictability of real-world data streams. Tables 6–7 present a comparison between the original CCC and our modified variant. In CCC-Easy, the performance gains observed in the original CCC are consistently reproduced in our modified variant across all methods. However, in CCC-Hard, RDumb shows degraded performance on ROID under our modified variant. This suggests that a fixed reset schedule can hinder adaptability when corruption transitions are both challenging and unpredictable. In contrast, our adaptive reset schedule consistently preserves performance gains across both benchmarks, demonstrating its robustness to such temporal unpredictability.

**d) CDC setting for dynamic domain-shift schedule.** We demonstrate the robustness of our approach under dynamic domain shifts by applying the Continual Dynamic Change (CDC; Zhang et al. (2025b)) protocol to IN-C. This protocol explicitly introduces *rapid domain switching* and *stochastic domain durations*, controlled via the Dirichlet parameter $\delta$. We evaluate our approach under both a standard setting ($\delta = 1.0$) and a more highly dynamic setting ($\delta = 10.0$) to further emphasize our robustness. As shown in Fig. 8, RDumb at $\delta = 1.0$ exhibits recurrent drops in accuracy; for instance, its accuracy falls from 36.91% to 34.00% at the fourth transition. In contrast, ASR maintains robust performance and steadily improves over time, rising from 35.08% to 41.34% across 20 transitions. A similar trend is observed at $\delta = 10.0$, where RDumb undergoes repeated performance degradations, whereas ASR remains resilient and gradually improves, reaching 41.68% by the 20th transition. Full evaluation results under these CDC settings are provided in Appendix D.1.

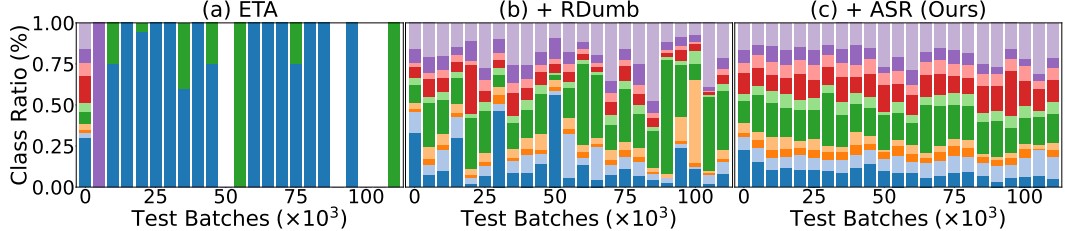

Figure 9: Histogram of predictions on CCC-Hard for randomly selected ten fixed class labels, comparing ETA, RDumb, and ASR to evaluate robustness against model collapse. Results are measured every $10^3$ batches, with class labels color-coded consistently.

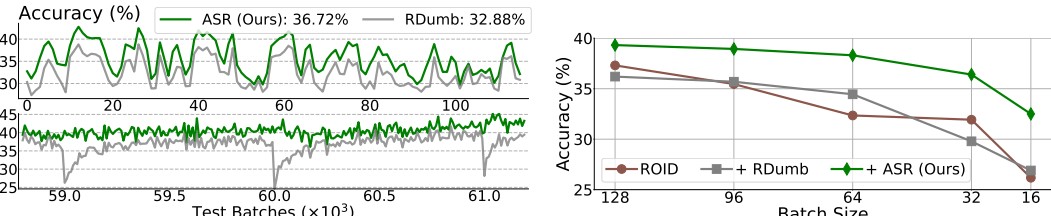

Figure 10: Comparison of ASR and RDumb on ETA. Accuracy (%) is shown from a global perspective (top) up to 110K batches, and at a finer temporal scale (bottom) from 59K to 61K batches.

Figure 11: Average accuracy (%) of ROID, RDumb, and ASR across all CCC levels under varying batch sizes from 128 down to 16.

**e) Collapse analysis.** We track predictions for ten fixed classes over time to analyze model collapse, as shown in Fig. 9. Experiments are conducted on CCC-Hard under the standard assumption of a *uniform* class distribution. We employ ETA (Niu et al., 2022) as a baseline due to its high susceptibility to collapse, ensuring a clear comparative analysis. Fig. 9 reveals that while ETA initially predicts a diverse set of classes, its prediction diversity abruptly diminishes, eventually failing to assign any of the ten monitored class labels. While RDumb (Press et al., 2023) helps mitigate such failure, its class distribution remains unstable and biased to some extent. In contrast, our method demonstrates superior resilience against collapse, consistently maintaining a uniform class distribution.

**f) Stability over time.** Beyond overall quantitative results, we examine whether our approach maintains stability over time, as it is vital for reliable real-world deployment. Fig. 10 illustrates accuracy trends for RDumb and ASR on the ETA model[2]. We compute the average accuracy at each step over $10^3$ batches across all CCC levels. ASR consistently outperforms RDumb from a global perspective (top), and its stability is evidenced by its significantly smaller performance fluctuations compared to RDumb when viewed at a finer temporal scale (bottom).

**g) Robustness to batch size.** We evaluate the robustness of our approach with respect to batch size, as shown in Fig. 11. We report the average accuracy across all CCC levels while varying the batch size from 128 down to 16. Although performance decreases with smaller batch sizes as expected, our method degrades more gracefully than ROID and RDumb. It consistently outperforms others across all sizes and shows the potential to scale more effectively beyond a batch size of 128. While RDumb excels primarily at its default size (64), our method maintains stable, improved performance regardless of the size. For single-sample inputs, this challenge can be addressed by accumulating samples over time and adapting once a sufficient count is reached, following Gong et al. (2023); Niu et al. (2024). Results for extremely small batch sizes (below 16) are provided in Appendix D.2.

## 5 CONCLUSION

In this paper, we mitigate model collapse in long-term TTA via Adaptive and Selective Reset (ASR), combined with importance-aware knowledge recovery and on-the-fly adaptation adjustment. Experimental results demonstrate the effectiveness of our proposed method across long-term TTA benchmarks, particularly in challenging settings. Specifically, our method outperforms the state-of-the-art by 44.12% on CCC-Hard. We hope that our work motivates further exploration into advanced reset algorithms for robust and stable adaptation over the long term, while preventing model collapse.

---

[2]Two methods are identical at $t = 0$, but the initial point in Fig. 10 (top) averages $t \in [0, 999]$.

## 6 ACKNOWLEDGMENTS

This work was partially supported by the National Research Foundation of Korea (NRF) grant funded by the Ministry of Science and ICT (MSIT) of the Korean government (RS2024-00341749), and Institute of Information & Communications Technology Planning & Evaluation (IITP) grant funded by MSIT (RS-2022-II220124, RS-2023-00259934, RS-2025-02283048).

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

# A ADDITIONAL DETAILS ON ASR

## A.1 LIMITATIONS OF FULL-PARAMETER RESET

**a) Performance drop.** To discuss the limitations of full-parameter reset, we first quantify RDumb's performance degradation after reset under the same setup as in Fig. 1. For each reset, we compute the average accuracy over 10 batches immediately before and after the reset. We then take the difference between these two averages and report the mean difference across all resets. RDumb has detrimental resets, which are around 40% of the total resets. While beneficial resets yield an average improvement of +4.32%p, detrimental resets cause an average degradation of -11.55%p, which is 2.67× larger and moreover exceeds RDumb's overall average accuracy of 9.77%. In contrast, while our approach has detrimental resets, which are around 38% of the total, their average performance degradation is only -2.50%p, which is 4.62× less than that of RDumb.

**b) Recovery delay.** When performance drops after reset, we count how many batches are required to reach the maximum accuracy that has been achieved within the 10 batches before the drop. When the model fails to reach it until the next reset, we set the recovery delay as the reset-to-reset interval (e.g., 1000). RDumb requires an average of 214 batches, indicating that recovery occupies 20% of the total adaptation steps, as a reset occurs every 1000 batches. RDumb requires a substantial amount of time to recover performance drops after reset, underscoring the inefficiency of full-parameter reset. When performance drops, our method needs an average of 32 batches, which is 6.68× fewer than RDumb.

## A.2 MORE DETAILS ON PREDICTION CONCENTRATION

To compute the correlation shown in Fig. 3, we employ ETA as a TTA baseline and CCC-Hard as a benchmark, as they exhibit clear collapse behavior and are thus suitable to illustrate the relationship between collapse and prediction concentration $\mathcal{C}_t$. Fig. 3 doesn't encode temporal information; thus, points on the right do not necessarily represent later time steps.

One may question *whether the pattern in Fig. 3 could be an artifact of the logit averaging in Eq. (1)*. To address this, we recompute $\mathcal{C}_t$ after eliminating the largest-magnitude logit in each batch, as well as after eliminating the top 10% of logits by magnitude. We then report the Pearson correlations in Table 8. Although these correlations are lower than the original value of 0.88 (see Fig. 3), they are still substantially strong. This indicates that the influence of extremely large-magnitude logits is limited, and that the pattern in Fig. 3 cannot be explained solely as an artifact of logit averaging.

| Excluded logits | Pearson correlation |
| --- | --- |
| None | 0.88 |
| Top-1 | 0.85 |
| Top-10% | 0.77 |

Table 8: Correlation between $\mathcal{C}_t$ and accuracy after eliminating large-magnitude logits.

Moreover, as a model approaches collapse, its predictions increasingly assign large logit values to a small set of dominant classes, leading to an increase in logit magnitude. Finally, this explains that the pattern in Fig. 3 is attributed to overall logits rather than only a few extreme outliers.

## A.3 RISK OF PROXIMITY TO RESET

Being close to a reset can compromise parameter integrity, which negatively impacts adaptation. We empirically illustrate this phenomenon by delaying the reset point, allowing corrupted parameters to accumulate in $\bar{\theta}$ (Eq. 4). Under recurring settings (IN-C), we observe harmful effects when corrupted information is reused. In the standard protocol, a reset is triggered when $\mathcal{C}_t > \bar{\mathcal{C}}_{t-1}$. For our variants, a reset is delayed until $\mathcal{C}_t - \bar{\mathcal{C}}_{t-1} > \varepsilon$, retaining adapting parameters beyond the standard reset point. Table 9 shows that delaying reset results in substantial performance degradation, even below ETA, confirming that parameters are vulnerable to corruption near the reset point and that such corruption severely impairs adaptation.

## A.4 MORE DETAILS ON KNOWLEDGE ACCUMULATION

We accomplish knowledge recovery by regularizing model parameters using their accumulated values and their estimated importance, as described in Sec. 3.4. However, special attention is required during the accumulation phase, as a trade-off arises: improving representation for the current domain increases vulnerability to corruption due to the accumulation of errors over time.

| Revisit # | $\varepsilon$ | 1 | 5 | 10 | 15 | 20 | Mean |
|---|---|---|---|---|---|---|---|
| ETA | - | 30.64 | 36.76 | 36.16 | 35.96 | 35.80 | 35.88 |
| + w/ delay | 0.001 | 28.42 | 33.39 | 37.80 | 38.82 | 39.02 | 36.25 |
| + w/ delay | 0.01 | 27.94 | 28.06 | 27.94 | 28.30 | 28.12 | 28.29 |
| + ASR (Ours) | - | 28.68 | 33.00 | 38.60 | 38.97 | 39.10 | 36.90 |
| ROID | - | 35.32 | 38.00 | 38.08 | 38.15 | 38.02 | 37.96 |
| + w/ delay | 0.001 | 35.08 | 40.01 | 41.42 | 41.96 | 41.54 | 40.85 |
| + w/ delay | 0.01 | 35.60 | 37.78 | 38.28 | 38.61 | 38.62 | 38.07 |
| + ASR (Ours) | - | 35.66 | 41.03 | 42.20 | 42.06 | 42.96 | 41.56 |

Table 9: Accuracy (%) under the standard and delaying reset settings on IN-C. Here, $\varepsilon$ denotes the threshold for delaying a reset; a larger $\varepsilon$ value encourages stronger reset delaying.

To tackle this problem, we propose a hybrid accumulation scheme that combines cumulative moving average (CMA) and exponential moving average (EMA). At every step, we use CMA to accumulate parameters $\theta_{t-1}^i$ and squared loss gradients with respect to each parameter, $\left(\nabla_{\theta_{t-1}^i} \mathcal{L}(\mathcal{B}_t; \theta_{t-1})\right)^2$, which correspond to the diagonal of the Fisher information matrix, as follows:

$$\tilde{\mathcal{F}}_t^i = \frac{(t - 1 - t_{\text{latest}}^*) \cdot \tilde{\mathcal{F}}_{t-1}^i + \left(\nabla_{\theta_{t-1}^i} \mathcal{L}(\mathcal{B}_t; \theta_{t-1})\right)^2}{t - t_{\text{latest}}^*}, \tag{8}$$

$$\tilde{\theta}_t^i = \frac{(t - 1 - t_{\text{latest}}^*) \cdot \tilde{\theta}_{t-1}^i + \theta_{t-1}^i}{t - t_{\text{latest}}^*}, \tag{9}$$

where $t_{\text{latest}}^*$ represents the latest reset point prior to $t$, and $\tilde{\mathcal{F}}_t^i$ and $\tilde{\theta}_t^i$ denote the CMA-accumulated Fisher information matrix and parameter for $\theta^i$, respectively, both initialized to zero values at $t = 0$. We then aggregate these CMA values at every reset using EMA, as follows:

$$\bar{\mathcal{F}}^i \leftarrow \mu_{\mathcal{F}} \cdot \bar{\mathcal{F}}^i + (1 - \mu_{\mathcal{F}}) \cdot \tilde{\mathcal{F}}_t^i, \tag{10}$$

$$\bar{\theta}^i \leftarrow \mu_{\theta} \cdot \bar{\theta}^i + (1 - \mu_{\theta}) \cdot \tilde{\theta}_t^i, \tag{11}$$

where $\mu_{\mathcal{F}}$ and $\mu_{\theta}$ are the momentum coefficients, both pre-defined as 0.9, and $\bar{\mathcal{F}}^i$ and $\bar{\theta}^i$ denote the EMA-accumulated Fisher information matrix and parameter for $\theta^i$, respectively, both initialized to zero. After the EMA updates, $\tilde{\mathcal{F}}_t^i$ and $\tilde{\theta}_t^i$ are re-initialized to zero.

## A.5 COMPUTATIONAL EFFICIENCY

We compare TTA baselines, ASR, and our ablations in terms of the number of trainable/total parameters, computation time (seconds per batch), and the average accuracy (%) across all CCC levels as shown in Table 10. Parameter restoration methods (i.e., ROID, RDumb, and ASR) require additional memory to store the initial model state, and the overhead for our extra parameters (primarily Fisher information) is negligible compared to the model size of 25.5M.

Each size of $\bar{\theta}$ and $\tilde{\theta}$ is $|\theta|$. Each size of $\bar{\mathcal{F}}$ and $\tilde{\mathcal{F}}$ is also $|\theta|$, as they retain only the diagonal elements, following standard practice in EWC (Kirkpatrick et al., 2017). As a result, each of these components occupies only 0.025M parameters (0.098% of the total). ASR computes Fisher information once per batch (size 64), increasing less than 0.001 seconds per batch. Overall, the computation and memory overhead of our additional parameters is minimal, making our method highly efficient in practice.

| Method | # Trainable | # Param | Time | Acc. |
|---|---|---|---|---|
| ETA | 53.1K | 25.5M | .083 | 21.72 |
| ROID | 53.1K | 51.1M | .125 | 33.91 |
| + RDumb | 53.1K | 51.1M | .125 | 35.39 |
| + ASR (Ours) | 53.1K | 51.2M | .200 | 38.32 |
| + w/o recovery (Sec. 3.4) | 53.1K | 51.1M | .200 | 37.80 |
| + w/o on-the-fly (Sec. 3.5) | 53.1K | 51.2M | .181 | 37.89 |

Table 10: Computational efficiency on CCC. # Trainable indicates the number of learnable parameters; # Param the total number of parameters; Time the seconds per batch (64 samples); and Acc. the average accuracy (%) across all CCC levels.

A.6   EFFECT OF KNOWLEDGE RECOVERY

Our proposed regularizer recovers essential knowledge that would otherwise be lost due to reset. We can explain this effect using the following two mechanisms from a theoretical perspective. First, we accumulate adapted parameters through a combination of CMA and EMA. It allows us to preserve historical adaptation information in a way analogous to Polyak averaging (Polyak & Juditsky, 1992), which provides a reliable information archive. Second, our Fisher-based regularization term follows the principle of Elastic Weight Consolidation (Kirkpatrick et al., 2017), giving stronger penalties to parameters that are more important in prior domains. Together, these mechanisms encourage crucial parameters to stay close to their pre-reset states, thereby effectively recovering essential knowledge. The knowledge recovery module introduced in Sec. 3.4 aims to effectively restore information that would otherwise be lost due to the reset. We evaluate its effectiveness under the same setup as Table 33 by measuring how much knowledge from prior domains is recovered. We define knowledge recovery as the average of per-domain gaps between the current performance and the best previously achieved performance. Positive values indicate successful recovery, while negative values indicate forgetting. As shown in Table 11, our approach consistently recovers substantial knowledge without forgetting. At the 10th revisit, ETA+ASR achieves +0.58 compared to -1.94 (w/o recovery), while ROID+ASR achieves +0.16 compared to -0.52 (w/o recovery).

Here, *knowledge* refers to information encoded in model parameters during adaptation, corresponding to $\bar{\theta}$ in Eq. (4). The essential knowledge is identifiable through Fisher information that highlights parameters that are highly informative about prior domains. Because directly quantifying knowledge is inherently challenging, we use task performance as a proxy for its assessment.

| Revisit # | 1 | ... | 10 | 15 | 20 | Mean |
|---|---|---|---|---|---|---|
| ETA + ASR (Ours) | 0.00 | ... | +0.58 | +0.08 | +0.02 | +0.24 |
| + w/o knowledge recovery | 0.00 | ... | -1.94 | -1.16 | -0.76 | -0.56 |
| ROID + ASR (Ours) | 0.00 | ... | +0.16 | +0.24 | +0.01 | +0.12 |
| + w/o knowledge recovery | 0.00 | ... | -0.52 | -0.42 | -0.10 | -0.14 |

Table 11: Knowledge recovery measured across multiple revisits on IN-C.

In addition, we evaluate the effectiveness of our knowledge recovery using accuracy as the evaluation metric. We assume domain-recurring settings by using IN-C as a benchmark to test whether models effectively preserve what they have learned. We compare our method with our variant that eliminates the knowledge recovery module (Sec. 3.4). As reported in Table 12, our variant without the recovery module exhibits a gradual accuracy decline across revisits, while our method maintains the superior performance. These results demonstrate that the recovery module effectively mitigates forgetting of knowledge acquired from prior domains.

| Revisit # | 1 | ... | 10 | 15 | 20 | Mean |
|---|---|---|---|---|---|---|
| ETA | 30.64 | ... | 36.16 | 35.96 | 35.80 | 35.88 |
| + ASR (Ours) | 28.68 | ... | 38.60 | 38.97 | 39.10 | 36.90 |
| + w/o knowledge recovery | 28.64 | ... | 37.45 | 36.49 | 36.34 | 36.56 |
| ROID | 35.32 | ... | 38.08 | 38.15 | 38.02 | 37.96 |
| + ASR (Ours) | 35.66 | ... | 42.20 | 42.06 | 42.96 | 41.56 |
| + w/o knowledge recovery | 35.35 | ... | 41.64 | 41.64 | 41.19 | 40.96 |

Table 12: Accuracy (%) comparison across multiple revisits on IN-C.

The benefit of knowledge recovery may appear negligible as experiments in Table 12 are conducted under easy-to-adapt settings. However, it does not imply that knowledge recovery is also ineffective in challenging adaptation scenarios. Although Table 12 proves that the knowledge recovery module functions as intended, IN-C is not a suitable benchmark for measuring the performance contribution. As discussed in Sec. 3.5, under challenging conditions, the regularization coefficient $\lambda_{\mathcal{F}}$ increases to encourage reuse of prior-domain information, thereby amplifying the impact of knowledge recovery. We thus consider CCC-Hard to better illustrate the contribution of the knowledge recovery module. Across several splits (e.g., 4, 7, and 8), eliminating this module substantially degrades performance,

whereas the full model consistently maintains higher accuracy. These results confirm that the module makes a clear performance contribution under challenging adaptation settings.

| Split # | 1 | 2 | 3 | 4 | 5 | 6 | 7 | 8 | 9 |
|---|---|---|---|---|---|---|---|---|---|
| ROID | 12.64 | 15.79 | 13.28 | 9.63 | 12.65 | 11.81 | 10.66 | 8.72 | 17.12 |
| + ASR (Ours) | 20.99 | 22.51 | 20.40 | 21.22 | 22.93 | 21.36 | 22.37 | 23.84 | 24.25 |
| + w/o recovery | 20.95 | 22.50 | 20.35 | 18.28 | 22.69 | 20.18 | 9.70 | 15.67 | 23.57 |

Table 13: Accuracy (%) comparison across 9 splits in CCC-Hard.

## A.7 HYPERPARAMETER SENSITIVITY

Since the validation set is unavailable in TTA, optimal hyperparameter tuning is inherently challenging. Instead, we tune hyperparameters using only 5% of a single holdout set in CCC-Hard (transition speed 2000; random seed 44). We further validate that our approach is insensitive to hyperparameter changes, as illustrated in Fig. 12. Specifically, we evaluate performance across all CCC levels while slightly perturbing the tuned values; the default settings are described in Table 14. The performance fluctuations observed in Fig. 12 are negligible. Finally, the capability to use a set of hyperparameters across all benchmarks underscores the practicality of our approach.

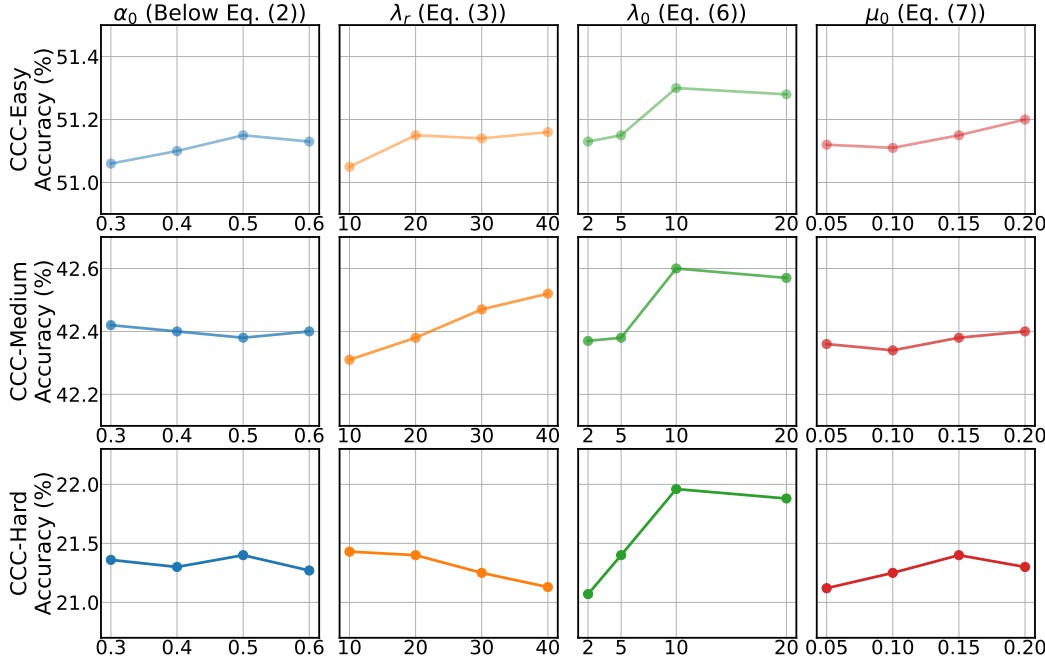

Figure 12: Hyperparameter sensitivity analysis showing robustness to variations around the default.

## A.8 IMPLEMENTATION DETAILS

The detailed hyperparameter settings are summarized in Table 14.

## A.9 ALGORITHM

The complete workflow of ASR is summarized in Algorithm 1.

## A.10 ASR UNDER ABRUPT DOMAIN CHANGES

Gan et al. (2023) observe that prediction confidence changes rapidly following domain shifts. Similarly, we find that prediction concentration exhibits abrupt dynamics with domain changes, as illustrated in Fig. 13. Since ASR depends on prediction concentration, it is essential to consider whether such abrupt behavior negatively impacts it. Abrupt declines in prediction concentration may appear to indicate random predictions. However, it is not severe enough to cause truly random behavior. In

| Hyperparameter | Reference | ResNet-50 | ViT-B-16 |
|---|---|---|---|
| $\alpha_0$: Initialization factor for $\bar{\mathcal{C}}_{t-1}$ | Below Eq. (2) in Sec. 3.3 | 0.5 | $5.0 \times 10^{-4}$ |
| $\mu_{\mathcal{C}}$: EMA update momentum for $\bar{\mathcal{C}}_{t-1}$ | Eq. (2) in Sec. 3.3 | 0.995 | 0.995 |
| $r_0$: Minimum reset proportion | Eq. (3) in Sec. 3.3 | 0.5 | 0.5 |
| $\lambda_r$: Reset proportion scaling factor | Eq. (3) in Sec. 3.3 | 20.0 | 0.1 |
| $\lambda_{\mathcal{F}}$: Fisher regularization coefficient | Eq. (4) in Sec. 3.4 | 5.0 | 5.0 |
| $\lambda_0$: Initialization factor for $\lambda_{\mathcal{F}}$ | Eq. (6) in Sec. 3.5 | 5.0 | 5.0 |
| $\mu_0$: Initialization factor for $\mu_{\mathcal{C}}$ | Eq. (7) in Sec. 3.5 | 0.15 | $1.0 \times 10^{-3}$ |

Table 14: Hyperparameter settings for ResNet-50 and ViT-B-16 used across all benchmarks.

---

**Algorithm 1:** Adaptive and Selective Reset (ASR)

---

**Input:** Test batches $\{\mathcal{B}_t\}_{t=1}^{T}$, adapting model $f_{\theta_*}$, source model $f_{\theta_0}$, cumulative concentration initialization factor $\alpha_0$, regularization coefficient initialization factor $\lambda_0$, EMA update momentum initialization factor $\mu_0$, minimum reset proportion $r_0$, reset proportion scaling factor $\lambda_r$, and EMA update momentum parameters $\{\mu_{\mathcal{F}}, \mu_{\theta}\}$.

Initialize $\bar{\mathcal{C}}_0 \leftarrow -\log(\alpha_0 \cdot C)$, $\tilde{\mathcal{F}}_0 \leftarrow 0$, and $\tilde{\theta}_0 \leftarrow 0$;

**for** $t \in \{1, \dots, T\}$ **do**

    `// 1) Model Adaptation`

    Generate logits $z_t = f_{\theta_{t-1}}(\mathcal{B}_t)$;

    Compute loss $\mathcal{L}(\mathcal{B}_t; \theta_{t-1})$ in Eq. (4);

    `// CMA-based Knowledge Accumulation`

    Update $\tilde{\mathcal{F}}_t$ and $\tilde{\theta}_t$ via Eq. (8) and Eq. (9);

    Update $\theta_t \leftarrow \underset{\theta_{t-1}}{\text{Optim}}\, \mathcal{L}(\mathcal{B}_t; \theta_{t-1})$;

    `// 2) On-the-fly Adaptation Adjustment`

    Compute prediction inconsistency $\phi_t = \frac{1}{|\mathcal{B}_t|} \sum_{i=1}^{|\mathcal{B}_t|} \mathbb{I}\left( \arg\max(\breve{y}_t^i) \neq \arg\max(\hat{y}_t^i) \right)$

        where $\breve{y}_t^i = \sigma(f_{\theta_0}(x_t^i))$ and $\hat{y}_t^i = \sigma(z_t^i)$;

    Adjust regularization coefficient $\lambda_{\mathcal{F}} = \lambda_0 \cdot \phi_t^2$

        and momentum coefficient $\mu_{\mathcal{C}} = \mu_0 \cdot \phi_t + 1 - \mu_0$;

    `// 3) Adaptive and Selective Reset`

    Compute prediction concentration $\mathcal{C}_t = \sum_{c=1}^{C} \hat{p}_{t_c} \log(\hat{p}_{t_c})$ where $\hat{p}_t = \sigma\left( \frac{1}{|\mathcal{B}_t|} \sum_{i=1}^{|\mathcal{B}_t|} z_t^i \right)$;

    **if** $\mathcal{C}_t - \bar{\mathcal{C}}_{t-1} \leq 0$ **then**

        Update $\bar{\mathcal{C}}_t \leftarrow \mu_{\mathcal{C}} \cdot \bar{\mathcal{C}}_{t-1} + (1 - \mu_{\mathcal{C}}) \cdot \mathcal{C}_t$;

    **end**

    **else**

        Compute selective reset factor $r_t = r_0 + \lambda_r \cdot (\mathcal{C}_t - \bar{\mathcal{C}}_{t-1})$ where $r_t \in [r_0, 1]$;

        Reset only the last $r_t$ proportion of total layers;

        Initialize $\bar{\mathcal{C}}_t \leftarrow -\log(\alpha_0 \cdot C)$;

        `// EMA-based Knowledge Accumulation`

        Update $\bar{\mathcal{F}} \leftarrow \text{EMA}(\bar{\mathcal{F}}, \tilde{\mathcal{F}}_t, \mu_{\mathcal{F}})$ in Eq. (10);

        Update $\bar{\theta} \leftarrow \text{EMA}(\bar{\theta}, \tilde{\theta}_t, \mu_{\theta})$ in Eq. (11);

        Initialize $\tilde{\mathcal{F}}_t \leftarrow 0$ and $\tilde{\theta}_t \leftarrow 0$;

    **end**

**end**

---

contrast, abrupt rises in prediction concentration often lead to $\mathcal{C}_t > \bar{\mathcal{C}}_{t-1}$, unintentionally triggering resets. Zhang et al. (2025b) note that negative knowledge transfer may occur with domain shifts and should be mitigated. Finally, such unintended resets can act as a safeguard against negative transfer. Overall, abrupt dynamics in prediction concentration with domain shifts do not pose a risk to ASR.

## B MORE DETAILS ON DATASETS

In this paper, we evaluate the stable adaptability of continual TTA methods under four long-term adaptation benchmarks, which are designed to induce model collapse.

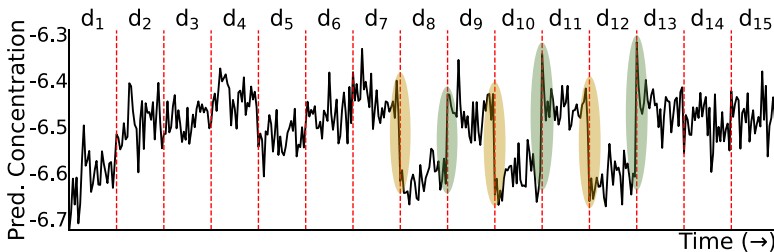

Figure 13: Prediction concentration in Eq. (1) over time under 15 corruption types in CIN-C. Dashed vertical lines (red) indicate corruption boundaries. Colored ellipses highlight abrupt dynamics associated with domain shifts ( yellow : abrupt decline; green : abrupt rise).

**1) Continually Changing Corruptions (CCC)** was introduced by RDumb (Press et al., 2023) and is constructed from the ImageNet-C dataset. CCC converts abrupt corruption transitions of ImageNet-C into smooth transitions by splitting integer corruption levels into continuous values from 0 to 5 in steps of 0.25. During the smooth corruption transition, one fades out (e.g., $0.5 \rightarrow 0.25$) and another fades in (e.g., $0.5 \rightarrow 0.75$) with the two overlapping. How these two corruptions move is determined by the source model's accuracy (0%–Hard / 20%–Medium / 40%–Easy). The "0%–Hard" setting results in the strictest corruption transitions. Moreover, each mode consists of different corruption types, as summarized in Table 15. Transition speed is defined as the number of images per step of the transition path and has three settings (1000 / 2000 / 5000). The ordering of corruptions in the path has three variants, which are determined by random seeds (43 / 44 / 45). CCC has a total of 27 combinations of 3 paths $\times$ 3 speeds $\times$ 3 orderings, and each combination contains approximately 7.5M images. Lastly, CCC has several factors contributing to model collapse, including prolonged time steps (Wang et al., 2022), similar corruption difficulty (Press et al., 2023) and corruption repetition (Hoang et al., 2024).

| Level | Corruption Types |
|---|---|
| Easy | Gaussian_noise, Shot_noise, Impulse_noise, Contrast |
| Medium | Gaussian_noise, Shot_noise, Impulse_noise, Defocus_blur, Glass_blur, Motion_blur, Zoom_blur, Snow, Frost, Fog, Contrast, Elastic, Pixelate |
| Hard | Gaussian_noise, Shot_noise, Impulse_noise, Defocus_blur, Glass_blur, Motion_blur, Zoom_blur, Snow, Frost, Fog, Contrast, Elastic, Pixelate, JPEG |

Table 15: Corruption types in a transition path for each adaptation difficulty.

**2) Concatenated ImageNet-C (CIN-C)** consists of image samples of the ImageNet-C validation set including 15 corruption types—*Gaussian noise*, *Shot noise*, *Impulse noise*, *Defocus blur*, *Glass blur*, *Motion blur*, *Zoom blur*, *Snow*, *Frost*, *Fog*, *Contrast*, *Brightness*, *Elastic*, *Pixelate*, and *JPEG*—at the highest severity level (level 5). CIN-C has 50K images for each corruption type, $10\times$ larger than the original ImageNet-C.

**3) ImageNet-C (IN-C)** is processed to evaluate stability against model collapse. This includes only four corruption types at level 5: *Gaussian noise*, *Shot noise*, *Impulse noise*, and *Contrast*. These are selected such that the source model achieves less than 10% accuracy, ensuring consistent adaptation difficulty across corruptions. Each corruption has 5K images, and the four corruptions are repeated 20 times, resulting in a total of 400K images, which represents corruption repetition, one of the collapse contributors. The corruption ordering is fixed as *Gaussian noise $\rightarrow$ Shot noise $\rightarrow$ Impulse noise $\rightarrow$ Contrast*.

**4) ImageNet-D109 (IN-D109)** is processed to evaluate stability against model collapse. It includes only four domains—*Clipart*, *Infograph*, *Painting*, and *Sketch*—selected from the six available domains such that the source model achieves less than 50% accuracy, ensuring consistent adaptation difficulty across domains. The domain sequence follows the fixed ordering *Clipart $\rightarrow$ Infograph $\rightarrow$ Painting $\rightarrow$ Sketch* and is repeated 20 times. Finally, IN-D109 includes only the classes shared with the DomainNet dataset, resulting in a total of 109 classes.

## C    ADDITIONAL ABLATION STUDIES

### C.1    EFFECT OF ADAPTIVE RESET

We evaluate the effectiveness of our adaptive reset scheme by comparing it with variants using fixed reset intervals. Table 16 shows that our adaptive reset effectively identifies when the model is on the verge of collapse and determines appropriate reset timing, leading to improved performance.

| Reset interval | Easy | Medium | Hard | Mean |
|---|---|---|---|---|
| **Fixed** | | | | |
| $T = 1000$ | 15.96 | 5.91 | 1.10 | 7.66 |
| $T = 10000$ | 49.88 | 39.69 | 15.75 | 35.11 |
| $T = 20000$ | 49.83 | 40.58 | 17.16 | 35.86 |
| $T = 50000$ | 49.90 | 40.16 | 14.26 | 34.77 |
| **Dynamic** | | | | |
| ASR (Ours) | **51.19** | **42.42** | **21.36** | **38.32** |

Table 16: Comparison with our variants using fixed reset intervals $T$ in terms of accuracy (%).

### C.2    EFFECT OF SELECTIVE RESET

Table 17 evaluates the effectiveness of our selective reset by comparing it with fixed-proportion reset variants. Among the variants, resetting the latter half of layers yields the best performance. Our reset also consistently initializes the reset proportion at 50% (i.e., $r_0 = 0.5$). Consequently, our selective reset is effective and resetting at least 50% of layers is important to mitigate accumulated errors.

| Reset target | Easy | Medium | Hard | Mean |
|---|---|---|---|---|
| **Fixed** | | | | |
| 20% | 49.27 | 40.01 | 16.83 | 35.37 |
| 50% | 50.91 | 42.07 | 20.80 | 37.93 |
| 80% | 50.72 | 41.40 | 19.68 | 37.27 |
| 100% | 49.83 | 40.35 | 15.99 | 35.39 |
| **Dynamic** | | | | |
| ASR (Ours) | **51.19** | **42.42** | **21.36** | **38.32** |

Table 17: Comparison with our variants resetting a fixed % of layers closer to the output.

### C.3    FAIR COMPARISON FOR RESET

We compare our reset to existing reset schemes proposed by SAR (Niu et al., 2023), RDumb (Press et al., 2023), and DA-TTA (Wang et al., 2024) across all CCC levels using ROID as a baseline, as shown in Table 18. Existing schemes either reset all parameters periodically (RDumb) or reset based on extremely high prediction confidence (SAR), or reset based on significant distribution shifts from the source domain (DA-TTA). For a fair comparison, our method ignores other components that are directly unrelated to reset; otherwise, it achieves 51.19 / 42.42 / 21.36% on CCC-Easy / -Medium / -Hard, respectively. Except for SAR, all existing schemes improve performance on CCC-Hard but degrade on easier levels. However, ASR consistently outperforms all other schemes, surpassing the second-best result by +2.80%p on average.

| Method | Easy | Medium | Hard | Mean |
|---|---|---|---|---|
| ROID | 49.74 | 40.19 | 11.81 | 33.91 |
| + SAR | 49.73 | 40.06 | 5.29 | 31.69 |
| + RDumb | 49.56 | 39.77 | 14.06 | 34.46 |
| + DA-TTA | 45.98 | 35.76 | 15.53 | 32.42 |
| + ASR (Ours) | **50.70** | **41.72** | **19.36** | **37.26** |

Table 18: Accuracy (%) comparison across various reset schemes on CCC.

## C.4 EFFECT OF HYBRID KNOWLEDGE ACCUMULATION

In our hybrid knowledge accumulation scheme, EMA is applied on top of CMA. CMA emphasizes local past information, mitigating the influence of recent parameters near collapse, whereas EMA globally weights more recent information, better reflecting distribution shifts. Table 19 compares our hybrid scheme with a CMA-only baseline, reporting accuracy (%) across all CCC levels.

| Method | Easy | Medium | Hard | Mean |
|---|---|---|---|---|
| CMA-only | 50.03 | 41.13 | 18.42 | 36.53 |
| Hybrid (Ours) | **51.19** | **42.42** | **21.36** | **38.32** |

Table 19: Hybrid knowledge accumulation (Ours) vs. CMA-only baseline.

## C.5 OPTIMALITY OF REPARAMETERIZATION

We assess the optimality of our reparameterization scheme in Eqs. (6)–(7). To design our reparameterization, we use only 5% of a holdout set from CCC-Hard (transition speed 2000; random seed 44) and determine the expression that balances simplicity and performance. As shown in Tables 20–21, we compare our expression with alternative expressions across all CCC levels. Across Tables 20–21, two candidate expressions exhibit comparable performance. Finally, we select the one that performs better on CCC-Hard, as it more effectively lowers the risk of poor adaptation in real-world scenarios.

| $\lambda_{\mathcal{F}}$ | Range | Easy | Medium | Hard | Mean |
|---|---|---|---|---|---|
| $\lambda_0$ | $\{\lambda_0\}$ | 51.07 | 42.33 | 20.27 | 37.89 |
| $\phi_t$ | $[0,1]$ | 51.09 | 42.26 | 20.56 | 37.97 |
| $\lambda_0 \cdot (1-\phi_t)^2$ | $[\lambda_0, 0]$ | 51.15 | 42.34 | 20.42 | 37.97 |
| $\lambda_0 \cdot \phi_t$ | $[0, \lambda_0]$ | 51.16 | 42.40 | 21.27 | 38.28 |
| $\lambda_0 \cdot \phi_t^2$ | $[0, \lambda_0]$ | **51.19** | **42.42** | **21.36** | **38.32** |

Table 20: Comparison with different expressions for $\lambda_{\mathcal{F}}$ across all CCC levels using accuracy (%).

| $\mu_{\mathcal{C}}$ | Range | Easy | Medium | Hard | Mean |
|---|---|---|---|---|---|
| $1-\mu_0$ | $\{1-\mu_0\}$ | 50.82 | 41.86 | 20.70 | 37.79 |
| $\phi_t$ | $[0,1]$ | 51.48 | 42.42 | 0.31 | 31.40 |
| $1-\mu_0 \cdot \phi_t$ | $[1, 1-\mu_0]$ | 51.17 | 42.47 | 4.07 | 32.57 |
| $1-\mu_0 \cdot (1-\phi_t^2)$ | $[1-\mu_0, 1]$ | 51.14 | 42.40 | 21.11 | 38.22 |
| $1-\mu_0 \cdot (1-\phi_t)$ | $[1-\mu_0, 1]$ | 51.19 | 42.42 | 21.36 | **38.32** |

Table 21: Comparison with different expressions for $\mu_{\mathcal{C}}$ across all CCC levels using accuracy (%).

# D ADDITIONAL RESULTS

## D.1 RESULTS UNDER CDC SETTINGS

In Tables 22–23, we present the full results under CDC settings, which extend those shown in Fig. 8.

| $\delta=1.0$ | Recurring visit 1 | 2 | 3 | 4 | 5 | 6 | 7 | 8 | 9 | 10 | 11 | 12 | 13 | 14 | 15 | 16 | 17 | 18 | 19 | 20 | Mean |
|---|---|---|---|---|---|---|---|---|---|---|---|---|---|---|---|---|---|---|---|---|---|
| ETA | 30.62 | 35.77 | 36.19 | 36.21 | 36.14 | 35.99 | 35.84 | 35.76 | 35.73 | 35.63 | 35.42 | 35.43 | 35.31 | 35.33 | 35.30 | 35.22 | 35.25 | 35.20 | 35.17 | 35.08 | 35.33 |
| + RDumb | 30.62 | 35.73 | 36.20 | 30.88 | 35.24 | 36.24 | 30.83 | 34.62 | 36.11 | 32.01 | 35.14 | 36.39 | 33.46 | 34.56 | 36.67 | 34.33 | 32.97 | 36.23 | 36.16 | 31.11 | 34.28 |
| + ASR (Ours) | 28.48 | 30.25 | 33.48 | 33.03 | 32.42 | 35.28 | 35.33 | 35.27 | 35.72 | 36.19 | 36.85 | 37.43 | 38.49 | 38.34 | 38.65 | 38.41 | 38.94 | 38.92 | 38.96 | 38.98 | 35.97 |
| ROID | 34.11 | 36.82 | 37.04 | 36.99 | 36.86 | 36.78 | 37.20 | 37.05 | 36.59 | 36.89 | 36.68 | 36.74 | 36.86 | 36.95 | 36.66 | 36.83 | 36.62 | 36.68 | 36.67 | 36.84 | 36.69 |
| + RDumb | 34.28 | 36.93 | 36.91 | 34.00 | 36.74 | 36.85 | 34.27 | 37.04 | 37.14 | 34.02 | 36.41 | 36.76 | 34.37 | 36.37 | 36.80 | 34.67 | 35.94 | 36.63 | 36.43 | 34.23 | 35.84 |
| + ASR (Ours) | 35.08 | 38.58 | 38.72 | 39.20 | 39.30 | 39.71 | 39.94 | 40.45 | 40.32 | 40.65 | 40.41 | 40.40 | 41.15 | 40.75 | 41.07 | 41.33 | 41.05 | 40.80 | 41.10 | 41.34 | 40.07 |

Table 22: Results on IN-C with CDC for $\delta = 1.0$ across revisit steps.

| | Recurring visit | | | | | | | | | | | | | | | | | | | | →| |
|---|---|---|---|---|---|---|---|---|---|---|---|---|---|---|---|---|---|---|---|---|---|
| $\delta = 10.0$ | 1 | 2 | 3 | 4 | 5 | 6 | 7 | 8 | 9 | 10 | 11 | 12 | 13 | 14 | 15 | 16 | 17 | 18 | 19 | 20 | Mean |
| ETA | 29.79 | 34.80 | 35.72 | 35.68 | 35.72 | 35.56 | 35.33 | 35.47 | 35.31 | 35.23 | 35.23 | 35.09 | 35.02 | 35.01 | 34.88 | 34.78 | 34.83 | 34.75 | 34.70 | 34.62 | 34.88 |
| + RDumb | 29.53 | 35.62 | 36.09 | 31.11 | 35.54 | 36.10 | 30.86 | 35.04 | 36.16 | 32.19 | 34.86 | 36.30 | 32.64 | 34.03 | 36.51 | 34.39 | 33.35 | 36.66 | 36.86 | 29.56 | 34.17 |
| + ASR (Ours) | 29.40 | 35.06 | 35.55 | 36.01 | 36.63 | 36.71 | 36.92 | 37.25 | 37.31 | 37.30 | 37.41 | 37.53 | 37.79 | 37.79 | 37.92 | 37.96 | 38.08 | 38.14 | 38.19 | 38.11 | 36.85 |
| ROID | 33.60 | 36.70 | 36.51 | 37.06 | 36.74 | 36.99 | 37.11 | 36.82 | 36.79 | 37.04 | 36.71 | 37.23 | 36.86 | 37.06 | 36.99 | 36.77 | 36.59 | 36.82 | 36.87 | 36.99 | 36.71 |
| + RDumb | 33.76 | 36.66 | 36.67 | 34.30 | 37.01 | 37.26 | 33.51 | 36.69 | 36.76 | 34.44 | 36.16 | 37.19 | 34.38 | 36.55 | 36.52 | 35.05 | 35.77 | 36.98 | 36.34 | 33.80 | 35.79 |
| + ASR (Ours) | 33.62 | 36.95 | 37.89 | 38.37 | 39.54 | 39.49 | 40.42 | 40.22 | 40.63 | 40.83 | 40.77 | 41.23 | 41.24 | 41.48 | 41.60 | 41.66 | 41.99 | 41.82 | 41.51 | 41.68 | 40.15 |

Table 23: Results on IN-C with CDC for $\delta = 10.0$ across revisit steps.

## D.2 ROBUSTNESS TO TRULY SMALL BATCH SIZES

We evaluate our method on truly small batch sizes, specifically 2 and 4, using a single split (transition speed 1000; random seed 43) of CCC-Easy under the same setup as Fig. 11. As shown in Table 24, at batch size 4, our method achieves 25.58 compared to 17.85 for RDumb, indicating that its robustness extends to smaller batch sizes below 16. However, this advantage diminishes at extremely small sizes (e.g., 2), where only a marginal improvement is observed (Ours – 6.46 vs. RDumb – 5.87). It is important to note that online TTA and continual TTA are distinct. Online TTA represents an extreme scenario with a batch size of 1, where all continual TTA methods yield near-random performance ($\sim$0.1). A practical solution for continual TTA methods in this scenario is to stack samples over time and adapt when a sufficient number is obtained.

| Batch size | 2 | 4 |
|---|---|---|
| ROID | 0.13 | 16.91 |
| + RDumb | 5.87 | 17.85 |
| + ASR (Ours) | 6.46 | 25.58 |

Table 24: Accuracy (%) under truly small batch sizes on CCC-Easy.

## D.3 RESULTS FOR CIFAR10-C/100-C

We focus on more challenging and realistic evaluation settings, as addressed in Press et al. (2023). The standard CIFAR-C setting is relatively simple and less representative of real-world scenarios. Following Press et al. (2023); Hoang et al. (2024), we categorize corruption types into three groups (Easy / Medium / Hard) based on the source model's accuracy to ensure consistent difficulty within the group, as reported in Table 25 for CIFAR10-C and Table 26 for CIFAR100-C. We then repeat the corruptions cyclically. The experimental results on CIFAR10-C / CIFAR100-C are shown in Fig. 14 and 15.

| Level | Corruption Types |
|---|---|
| Easy | Motion_blur, Snow, Fog, Elastic, JPEG |
| Medium | Defocus_blur, Glass_blur, Zoom_blur, Frost, Contrast, Pixelate |
| Hard | Gaussian_noise, Shot_noise, Impulse_noise |

Table 25: Corruption types for each level of CIFAR10-C.

| Level | Corruption Types |
|---|---|
| Easy | Impulse_noise, Defocus_blur, Motion_blur, Zoom_blur, Snow, Brightness, Elastic |
| Medium | Glass_blur, Frost, Fog, Contrast, JPEG |
| Hard | Gaussian_noise, Shot_noise, Pixelate |

Table 26: Corruption types for each level of CIFAR100-C.

## D.4 FULL RESULTS ON RESNET

In Tables 28–34, we present the full results on ResNet-50, which extend those reported in Table 4.

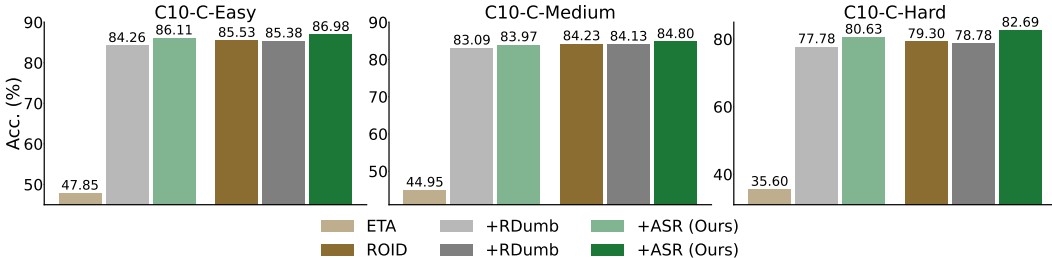

Figure 14: Comparison of ETA / ROID and their variants combined with RDumb or ASR across three levels of CIFAR10-C using accuracy (%), averaged over 1000 recurring visits of corruptions.

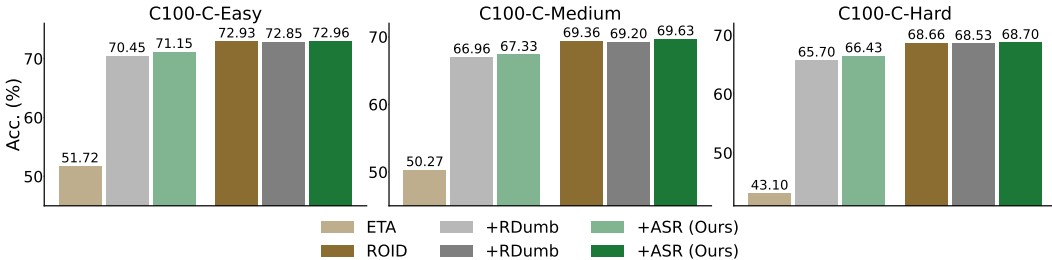

Figure 15: Comparison of ETA / ROID and their variants combined with RDumb or ASR across three levels of CIFAR100-C using accuracy (%), averaged over 1000 recurring visits of corruptions.

### D.5 FULL RESULTS ON VIT

In Tables 35–37, we present the full results on ViT-B-16, which extend those reported in Table 3.

### D.6 RESULTS ON VIT-TINY

We evaluate our method on a lightweight backbone (ViT-Tiny). As reported in Table 27, our method consistently improves over the baselines on CCC-Easy and CCC-Medium. However, due to the extremely limited backbone capacity, adapting to CCC-Hard is particularly challenging, as reflected in Table 27 where all ROID-based variants achieve only 0.10% accuracy. Even under such severe capacity constraints, our method maintains performance gains comparable to those reported in Table 3, demonstrating our robustness.

| ViT-Tiny | CCC-Hard | CCC-Medium | CCC-Easy |
|---|---|---|---|
| ETA | 2.29 | 34.20 | 47.09 |
| + RDumb | 4.45 | 32.51 | 45.51 |
| + ASR (Ours) | 5.30 | 36.48 | 47.23 |
| ROID | 0.10 | 32.14 | 45.29 |
| + RDumb | 0.10 | 31.61 | 44.92 |
| + ASR (Ours) | 0.10 | 34.47 | 45.64 |

Table 27: Accuracy (%) comparison on ViT-Tiny.

## E DISCUSSIONS

**Q: Does ASR rely on incremental or heuristic solutions for long-term TTA?**
**A:** ASR is not a collection of ad-hoc fixes. We reframe long-term TTA via a new perspective, treating collapse prevention as a *continuous decision-making task*. Prior work typically performs a reset at a fixed interval (Press et al., 2023) or after collapse (Niu et al., 2023), but ASR continuously estimates the risk of collapse. Although ASR is integrated with multiple components, these are built under the shared principle: *balancing catastrophic and graceful knowledge forgetting*.

**Q: Doesn't ASR fail to overcome the need for reset in long-term TTA?**
**A:** Reset is an essential and widely recognized mechanism for preventing collapse in long-term TTA. Neural networks often converge to sharp minima, which are tricky to escape using standard gradient

updates (Keskar et al., 2017), and collapse represents a more challenging situation, making recovery *nearly impossible* without a reset (Hoang et al., 2024). Despite its critical role, reset has been largely underexplored: existing methods typically apply full-parameter reset at fixed intervals, ignoring the model's current state. We directly address these fundamental limitations by proposing a scheme that dynamically adjusts both the timing and scope of reset.

***Q: Are the marginal gains worth the engineering effort, or would simpler variants suffice?***

*A:* ASR is specifically designed to address collapse in long-term TTA and proves highly effective in challenging realistic scenarios. CCC-Hard best exemplifies such conditions, where ASR achieves a substantial 44.12% improvement over the state of the art, demonstrating that it addresses hard tasks robustly. In contrast, easier benchmarks such as IN-C or IN-D109 naturally yield smaller gains that simpler methods could also achieve in these settings. This highlights that modest gains on easy tasks do not indicate that simpler variants would suffice for more challenging scenarios. As collapse-prone benchmarks become more available, we expect our advantages to be even more pronounced.

| Transition speed | 1000 | | | 2000 | | | 5000 | | | Acc. (%) |
|---|---|---|---|---|---|---|---|---|---|---|
| Corruption ordering | 43 | 44 | 45 | 43 | 44 | 45 | 43 | 44 | 45 | Mean |
| Source | 33.89 | 33.97 | 33.95 | 33.69 | 33.90 | 33.99 | 33.34 | 34.06 | 34.23 | 33.89±0.2 |
| RMT (CVPR'23) | 48.15 | 46.78 | 47.38 | 46.70 | 46.44 | 47.80 | 48.61 | 45.03 | 48.07 | 47.22±1.0 |
| RoTTA (CVPR'23) | 1.76 | 1.49 | 1.88 | 2.43 | 1.84 | 2.28 | 2.50 | 3.06 | 3.28 | 2.28±0.6 |
| SANTA (TMLR'23) | 47.33 | 47.47 | 47.49 | 47.87 | 47.68 | 47.77 | 48.32 | 47.11 | 48.10 | 47.68±0.4 |
| LAW (WACV'24) | 2.71 | 2.50 | 2.92 | 2.99 | 2.44 | 3.25 | 2.79 | 2.20 | 3.55 | 2.82±0.4 |
| ViDA (ICLR'24) | 13.52 | 12.28 | 12.74 | 13.68 | 12.32 | 11.89 | 13.81 | 12.29 | 11.61 | 12.68±0.8 |
| DPLOT (CVPR'24) | 36.68 | 35.71 | 35.84 | 36.18 | 34.02 | 35.79 | 33.55 | 32.94 | 34.61 | 35.04±1.2 |
| PALM (AAAI'25) | 1.55 | 1.34 | 1.66 | 1.67 | 1.29 | 1.70 | 1.84 | 1.23 | 1.73 | 1.56±0.2 |
| CoTTA (CVPR'22) | 17.01 | 15.98 | 16.24 | 18.05 | 17.02 | 17.13 | 19.33 | 18.10 | 18.60 | 17.50±1.0 |
| SAR (ICLR'23) | 36.65 | 36.24 | 36.47 | 39.21 | 37.55 | 38.75 | 39.92 | 37.84 | 38.83 | 37.94±1.2 |
| PETAL (CVPR'23) | 2.57 | 2.52 | 2.64 | 2.62 | 2.54 | 2.71 | 2.66 | 2.43 | 2.64 | 2.59±0.1 |
| CMF (ICLR'24) | 48.29 | 48.33 | 48.20 | 49.38 | 49.25 | 49.12 | 50.87 | 49.95 | 50.40 | 49.31±0.9 |
| + RDumb (NeurIPS'23) | 48.02 | 48.07 | 47.95 | 49.08 | 48.96 | 48.83 | 50.41 | 49.57 | 50.08 | 49.00±0.8 |
| + ASR (Ours) | 50.08 | 50.12 | 49.98 | 51.11 | 50.97 | 50.82 | 52.58 | 51.59 | 52.11 | 51.04±0.9 |
| DATTA (ECCV'24) | 9.87 | 18.26 | 23.48 | 28.49 | 24.46 | 20.79 | 25.26 | 29.36 | 23.65 | 22.62±5.5 |
| PeTTA (NeurIPS'24) | 34.61 | 34.43 | 34.56 | 36.45 | 36.26 | 36.40 | 40.43 | 38.90 | 40.01 | 36.89±2.2 |
| + RDumb (NeurIPS'23) | 34.53 | 34.36 | 34.50 | 36.36 | 36.18 | 36.33 | 40.30 | 38.81 | 39.92 | 36.81±2.2 |
| + ASR (Ours) | 36.96 | 36.92 | 37.11 | 38.72 | 38.59 | 38.79 | 43.14 | 41.52 | 42.53 | 39.36±2.3 |
| ETA (ICML'22) | 42.13 | 42.23 | 42.12 | 43.46 | 43.13 | 42.87 | 45.25 | 43.86 | 44.07 | 43.24±1.0 |
| + RDumb (NeurIPS'23) | 48.55 | 48.57 | 48.49 | 49.53 | 49.42 | 49.35 | 50.79 | 49.97 | 50.57 | 49.47±0.8 |
| + COME (ICLR'25) | 10.96 | 8.87 | 10.49 | 0.63 | 10.42 | 9.79 | 2.46 | 1.98 | 1.43 | 6.34±4.3 |
| + ReservoirTTA (NeurIPS'25) | 49.02 | 48.96 | 48.93 | 49.47 | 48.38 | 48.80 | 49.95 | 47.49 | 48.77 | 48.86±0.6 |
| + ASR (Ours) | 50.33 | 50.31 | 50.13 | 51.27 | 51.12 | 50.92 | 52.73 | 51.78 | 52.21 | 51.20±0.8 |
| EATA (ICML'22) | 48.53 | 48.65 | 48.48 | 49.52 | 49.47 | 49.35 | 51.00 | 50.07 | 50.64 | 49.52±0.9 |
| + RDumb (NeurIPS'23) | 49.06 | 49.09 | 49.03 | 50.02 | 49.89 | 49.85 | 51.20 | 50.41 | 51.02 | 49.95±0.8 |
| + COME (ICLR'25) | 46.99 | 47.04 | 47.00 | 37.50 | 47.72 | 47.63 | 49.11 | 48.26 | 48.80 | 46.67±3.3 |
| + ReservoirTTA (NeurIPS'25) | 51.07 | 51.19 | 51.18 | 51.72 | 51.45 | 51.49 | 52.39 | 51.30 | 51.76 | 51.51±0.4 |
| + ASR (Ours) | 49.40 | 49.40 | 49.32 | 50.43 | 50.31 | 50.17 | 52.00 | 51.06 | 51.53 | 50.40±0.9 |
| ROID (WACV'24) | 49.02 | 49.03 | 48.92 | 49.95 | 49.81 | 49.74 | 51.15 | 50.37 | 50.94 | 49.88±0.8 |
| + RDumb (NeurIPS'23) | 48.82 | 48.85 | 48.74 | 49.76 | 49.63 | 49.56 | 50.91 | 50.15 | 50.75 | 49.69±0.8 |
| + COME (ICLR'25) | 0.11 | 0.12 | 0.12 | 0.12 | 0.13 | 0.12 | 0.11 | 0.11 | 0.13 | 0.12±0.0 |
| + ReservoirTTA (NeurIPS'25) | 50.45 | 50.57 | 50.51 | 51.06 | 50.89 | 50.87 | 51.43 | 50.65 | 51.30 | 50.86±0.3 |
| + ASR (Ours) | 50.50 | 50.58 | 50.42 | 51.47 | 51.36 | 51.19 | 52.86 | 51.94 | 52.35 | 51.41±0.8 |

Table 28: Performance comparison with state-of-the-art methods on CCC-Easy across nine variants: three corruption transition speeds (1000 / 2000 / 5000) and three corruption orderings that are determined by random seeds (43 / 44 / 45).

| Transition speed | 1000 | | | 2000 | | | 5000 | | | Acc. (%) |
|---|---|---|---|---|---|---|---|---|---|---|
| Corruption ordering | 43 | 44 | 45 | 43 | 44 | 45 | 43 | 44 | 45 | Mean |
| Source | 16.95 | 16.78 | 16.95 | 16.59 | 16.87 | 16.97 | 16.57 | 16.91 | 17.20 | 16.87±0.2 |
| RMT (CVPR'23) | 35.48 | 35.38 | 35.60 | 36.07 | 35.65 | 34.08 | 35.41 | 31.42 | 36.09 | 35.02±1.4 |
| RoTTA (CVPR'23) | 1.23 | 1.00 | 1.36 | 1.84 | 1.31 | 1.70 | 2.76 | 2.08 | 2.54 | 1.76±0.6 |
| SANTA (TMLR'23) | 33.75 | 33.77 | 34.17 | 35.65 | 34.18 | 34.26 | 35.94 | 33.79 | 34.57 | 34.45±0.8 |
| LAW (WACV'24) | 1.56 | 1.09 | 1.50 | 1.66 | 0.64 | 1.57 | 1.38 | 0.76 | 1.57 | 1.30±0.4 |
| ViDA (ICLR'24) | 6.16 | 6.10 | 6.20 | 5.73 | 6.19 | 5.78 | 4.95 | 5.65 | 5.01 | 5.75±0.5 |
| DPLOT (CVPR'24) | 14.64 | 10.70 | 18.70 | 12.05 | 7.58 | 18.07 | 9.50 | 6.83 | 20.08 | 13.13±4.7 |
| PALM (AAAI'25) | 0.76 | 0.50 | 0.98 | 0.63 | 0.37 | 1.47 | 0.51 | 0.62 | 0.83 | 0.74±0.3 |
| CoTTA (CVPR'22) | 9.62 | 8.65 | 9.10 | 10.04 | 9.06 | 10.30 | 10.56 | 9.52 | 11.63 | 9.83±0.9 |
| SAR (ICLR'23) | 19.98 | 21.20 | 20.01 | 23.25 | 20.91 | 21.19 | 23.89 | 24.91 | 24.89 | 22.25±1.9 |
| PETAL (CVPR'23) | 2.14 | 1.92 | 2.18 | 2.06 | 1.84 | 2.18 | 2.02 | 1.69 | 1.98 | 2.00±0.2 |
| CMF (ICLR'24) | 38.77 | 38.41 | 38.79 | 41.32 | 40.28 | 40.28 | 42.84 | 41.93 | 42.85 | 40.61±1.6 |
| + RDumb (NeurIPS'23) | 38.13 | 37.86 | 38.20 | 40.66 | 39.59 | 39.61 | 41.96 | 40.99 | 41.95 | 39.88±1.5 |
| + ASR (Ours) | 40.23 | 40.01 | 40.33 | 42.69 | 41.70 | 41.66 | 44.22 | 43.25 | 44.15 | 42.03±1.6 |
| DATTA (ECCV'24) | 9.42 | 10.96 | 9.19 | 12.78 | 13.28 | 12.85 | 19.46 | 14.37 | 17.49 | 13.31±3.2 |
| PeTTA (NeurIPS'24) | 19.34 | 19.30 | 19.65 | 23.41 | 21.92 | 22.13 | 27.34 | 25.19 | 25.47 | 22.64±2.8 |
| + RDumb (NeurIPS'23) | 19.28 | 19.23 | 19.56 | 23.07 | 21.64 | 21.87 | 26.97 | 24.84 | 25.05 | 22.39±2.6 |
| + ASR (Ours) | 22.47 | 22.90 | 22.99 | 26.67 | 25.33 | 23.96 | 19.62 | 20.34 | 27.29 | 23.51±2.5 |
| ETA (ICML'22) | 22.28 | 13.05 | 18.40 | 25.36 | 17.55 | 22.01 | 20.87 | 3.37 | 28.41 | 19.03±6.9 |
| + RDumb (NeurIPS'23) | 37.72 | 37.45 | 37.83 | 40.21 | 39.05 | 39.15 | 41.55 | 40.46 | 41.34 | 39.42±1.5 |
| + COME (ICLR'25) | 0.77 | 0.53 | 0.55 | 0.68 | 0.47 | 0.57 | 0.68 | 0.31 | 0.44 | 0.56±0.1 |
| + ReservoirTTA (NeurIPS'25) | 41.22 | 40.97 | 41.40 | 42.97 | 41.56 | 41.87 | 43.30 | 41.72 | 43.29 | 42.03±0.9 |
| + ASR (Ours) | 40.10 | 39.78 | 40.04 | 42.34 | 41.47 | 41.49 | 44.13 | 43.35 | 44.25 | 41.88±1.6 |
| EATA (ICML'22) | 37.36 | 36.91 | 37.40 | 39.98 | 38.66 | 38.80 | 41.52 | 40.47 | 41.58 | 39.19±1.7 |
| + RDumb (NeurIPS'23) | 38.26 | 38.06 | 38.43 | 40.73 | 39.57 | 39.67 | 41.99 | 40.91 | 41.73 | 39.93±1.4 |
| + COME (ICLR'25) | 34.81 | 34.63 | 35.02 | 37.47 | 36.03 | 36.15 | 38.86 | 37.87 | 38.79 | 36.63±1.6 |
| + ReservoirTTA (NeurIPS'25) | 41.22 | 40.97 | 41.40 | 42.97 | 41.56 | 41.87 | 43.30 | 41.72 | 43.29 | 42.03±0.9 |
| + ASR (Ours) | 38.84 | 38.65 | 38.95 | 41.38 | 40.34 | 40.34 | 43.21 | 42.30 | 43.04 | 40.78±1.7 |
| ROID (WACV'24) | 38.79 | 38.64 | 38.91 | 41.24 | 40.16 | 40.19 | 42.44 | 41.44 | 42.41 | 40.47±1.4 |
| + RDumb (NeurIPS'23) | 38.42 | 38.26 | 38.56 | 40.85 | 39.75 | 39.77 | 42.00 | 40.94 | 41.90 | 40.05±1.4 |
| + COME (ICLR'25) | 0.15 | 0.12 | 0.15 | 0.11 | 0.10 | 0.18 | 0.11 | 0.11 | 0.13 | 0.13±0.0 |
| + ReservoirTTA (NeurIPS'25) | 39.99 | 39.87 | 40.15 | 41.85 | 40.60 | 40.67 | 42.07 | 40.87 | 41.98 | 40.89±0.8 |
| + ASR (Ours) | 41.13 | 40.91 | 41.20 | 43.40 | 42.49 | 42.42 | 44.77 | 43.98 | 44.91 | 42.80±1.5 |

Table 29: Performance comparison with state-of-the-art methods on CCC-Medium across nine variants: three corruption transition speeds (1000 / 2000 / 5000) and three corruption orderings that are determined by random seeds (43 / 44 / 45).

| Transition speed | 1000 | | | 2000 | | | 5000 | | | Acc. (%) |
|---|---|---|---|---|---|---|---|---|---|---|
| Corruption ordering | 43 | 44 | 45 | 43 | 44 | 45 | 43 | 44 | 45 | Mean |
| Source | 1.29 | 1.23 | 1.31 | 1.31 | 1.23 | 1.30 | 1.33 | 1.19 | 1.25 | 1.27±0.0 |
| RMT (CVPR'23) | 12.13 | 13.18 | 13.12 | 9.43 | 10.74 | 12.83 | 7.73 | 0.86 | 9.27 | 9.92±3.7 |
| RoTTA (CVPR'23) | 0.50 | 0.77 | 0.74 | 0.66 | 0.77 | 0.96 | 0.79 | 0.17 | 0.87 | 0.69±0.2 |
| SANTA (TMLR'23) | 9.28 | 9.93 | 9.16 | 9.08 | 9.89 | 9.14 | 8.96 | 9.77 | 9.97 | 9.46±0.4 |
| LAW (WACV'24) | 0.34 | 0.17 | 0.22 | 0.31 | 0.17 | 0.22 | 0.27 | 0.16 | 0.20 | 0.23±0.1 |
| ViDA (ICLR'24) | 0.44 | 0.39 | 0.45 | 0.45 | 0.40 | 0.44 | 0.48 | 0.38 | 0.38 | 0.42±0.0 |
| DPLOT (CVPR'24) | 0.53 | 0.88 | 0.77 | 0.64 | 0.24 | 0.31 | 1.22 | 0.12 | 0.36 | 0.56±0.3 |
| PALM (AAAI'25) | 0.14 | 0.12 | 0.10 | 0.14 | 0.11 | 0.17 | 0.13 | 0.12 | 0.16 | 0.13±0.0 |
| CoTTA (CVPR'22) | 1.73 | 1.98 | 1.95 | 1.43 | 1.71 | 2.09 | 1.44 | 0.20 | 1.19 | 1.52±0.5 |
| SAR (ICLR'23) | 1.54 | 1.64 | 1.52 | 1.61 | 2.29 | 1.67 | 2.90 | 2.50 | 2.56 | 2.03±0.5 |
| PETAL (CVPR'23) | 0.68 | 0.64 | 0.74 | 0.81 | 0.56 | 0.80 | 0.96 | 0.14 | 0.55 | 0.65±0.2 |
| CMF (ICLR'24) | 1.06 | 0.62 | 0.46 | 1.08 | 0.40 | 0.73 | 2.41 | 0.29 | 0.95 | 0.89±0.6 |
| + RDumb (NeurIPS'23) | 13.00 | 15.10 | 12.44 | 13.51 | 15.44 | 13.56 | 16.51 | 17.96 | 17.85 | 15.04±1.9 |
| + ASR (Ours) | 19.57 | 21.40 | 19.43 | 20.10 | 21.39 | 21.49 | 20.62 | 22.46 | 22.86 | 21.04±1.1 |
| DATTA (ECCV'24) | 3.00 | 2.48 | 2.57 | 1.49 | 1.51 | 1.56 | 1.61 | 1.70 | 1.53 | 1.94±0.5 |
| PeTTA (NeurIPS'24) | 4.93 | 5.44 | 4.70 | 5.88 | 6.53 | 5.71 | 6.58 | 7.12 | 7.15 | 6.00±0.8 |
| + RDumb (NeurIPS'23) | 5.04 | 5.48 | 4.72 | 5.69 | 6.18 | 5.82 | 6.14 | 6.63 | 6.62 | 5.81±0.6 |
| + ASR (Ours) | 7.42 | 7.67 | 7.03 | 7.62 | 8.42 | 7.53 | 6.31 | 6.86 | 6.55 | 7.27±0.6 |
| ETA (ICML'22) | 0.67 | 0.28 | 0.26 | 0.41 | 0.18 | 0.29 | 0.34 | 0.19 | 0.24 | 0.32±0.1 |
| + RDumb (NeurIPS'23) | 7.58 | 9.64 | 6.90 | 9.46 | 11.08 | 8.74 | 10.33 | 12.67 | 11.57 | 9.77±1.8 |
| + COME (ICLR'25) | 0.15 | 0.11 | 0.14 | 0.15 | 0.11 | 0.24 | 0.13 | 0.10 | 0.13 | 0.14±0.0 |
| + ReservoirTTA (NeurIPS'25) | 1.73 | 0.89 | 0.42 | 0.93 | 0.93 | 2.63 | 0.24 | 1.99 | 0.57 | 1.15±0.8 |
| + ASR (Ours) | 15.01 | 18.18 | 13.36 | 15.95 | 18.57 | 15.83 | 18.07 | 18.32 | 20.59 | 17.10±2.1 |
| EATA (ICML'22) | 1.25 | 0.80 | 0.57 | 1.11 | 0.49 | 0.64 | 1.67 | 0.32 | 0.51 | 0.82±0.4 |
| + RDumb (NeurIPS'23) | 9.14 | 10.97 | 8.07 | 10.52 | 12.48 | 9.94 | 11.67 | 13.88 | 13.16 | 11.09±1.8 |
| + COME (ICLR'25) | 0.74 | 0.96 | 0.79 | 1.07 | 0.29 | 0.85 | 1.51 | 0.21 | 0.79 | 0.80±0.4 |
| + ReservoirTTA (NeurIPS'25) | 11.34 | 10.53 | 11.47 | 9.36 | 4.21 | 10.86 | 6.42 | 2.82 | 1.20 | 7.58±3.8 |
| + ASR (Ours) | 15.01 | 17.19 | 14.03 | 15.63 | 17.49 | 15.38 | 17.42 | 15.36 | 18.44 | 16.22±1.4 |
| ROID (WACV'24) | 12.64 | 15.79 | 13.28 | 9.63 | 12.65 | 11.81 | 10.66 | 8.72 | 17.12 | 12.48±2.6 |
| + RDumb (NeurIPS'23) | 14.13 | 15.92 | 13.74 | 14.03 | 16.05 | 14.06 | 15.48 | 17.34 | 17.98 | 15.41±1.5 |
| + COME (ICLR'25) | 0.12 | 0.11 | 0.14 | 0.13 | 0.10 | 0.15 | 0.13 | 0.10 | 0.12 | 0.12±0.0 |
| + ReservoirTTA (NeurIPS'25) | 11.83 | 14.92 | 13.50 | 13.41 | 13.53 | 13.20 | 13.60 | 13.96 | 15.46 | 13.71±1.0 |
| + ASR (Ours) | 20.99 | 22.51 | 20.40 | 21.22 | 22.93 | 21.36 | 22.37 | 23.84 | 24.25 | 22.21±1.2 |

Table 30: Performance comparison with state-of-the-art methods on CCC-Hard across nine variants: three corruption transition speeds (1000 / 2000 / 5000) and three corruption orderings that are determined by random seeds (43 / 44 / 45).

| Method | 1 | 2 | 3 | 4 | 5 | 6 | 7 | 8 | 9 | 10 | Mean |
|---|---|---|---|---|---|---|---|---|---|---|---|
| Source | 18.01 | 18.01 | 18.01 | 18.01 | 18.01 | 18.01 | 18.01 | 18.01 | 18.01 | 18.01 | 18.01±0.0 |
| RMT (CVPR'23) | 47.68 | 45.48 | 44.09 | 43.87 | 46.48 | 45.70 | 44.68 | 44.86 | 43.61 | 43.64 | 45.01±1.3 |
| + Source-free | 42.33 | 39.13 | 33.49 | 36.63 | 40.32 | 38.21 | 37.75 | 34.24 | 33.02 | 33.63 | 36.88±3.1 |
| RoTTA (CVPR'23) | 27.21 | 31.85 | 27.23 | 24.99 | 28.56 | 30.99 | 30.55 | 29.61 | 30.70 | 28.85 | 29.05±2.0 |
| SANTA (TMLR'23) | 40.00 | 39.85 | 39.83 | 39.77 | 39.84 | 39.53 | 39.83 | 39.85 | 39.63 | 39.98 | 39.81±0.1 |
| LAW (WACV'24) | 22.91 | 17.65 | 1.14 | 14.63 | 24.72 | 17.70 | 10.91 | 11.27 | 11.66 | 2.06 | 13.47±7.4 |
| ViDA (ICLR'24) | 17.87 | 17.81 | 17.62 | 17.76 | 17.78 | 17.83 | 17.77 | 17.80 | 17.79 | 17.60 | 17.76±0.1 |
| DPLOT (CVPR'24) | 37.52 | 33.86 | 30.34 | 31.38 | 33.64 | 29.72 | 32.60 | 30.58 | 29.99 | 30.38 | 32.00±2.3 |
| PALM (AAAI'25) | 21.14 | 15.12 | 3.47 | 16.37 | 23.57 | 14.06 | 8.57 | 11.02 | 8.86 | 4.75 | 12.69±6.3 |
| EATA (ICML'22) | 48.03 | 47.60 | 47.85 | 47.42 | 48.18 | 47.87 | 47.75 | 47.78 | 48.01 | 47.62 | 47.81±0.2 |
| + COME (ICLR'25) | 44.34 | 43.66 | 44.13 | 43.59 | 44.48 | 44.24 | 44.04 | 44.16 | 44.62 | 44.13 | 44.14±0.3 |
| CoTTA (CVPR'22) | 39.59 | 36.76 | 31.44 | 35.60 | 38.70 | 36.71 | 36.41 | 34.66 | 33.18 | 32.03 | 35.51±2.6 |
| SAR (ICLR'23) | 41.62 | 41.13 | 40.77 | 40.04 | 41.61 | 40.71 | 41.48 | 40.63 | 40.44 | 35.11 | 40.35±1.8 |
| PETAL (CVPR'23) | 40.87 | 38.09 | 33.08 | 38.92 | 40.03 | 38.73 | 37.50 | 36.94 | 34.65 | 34.03 | 37.28±2.5 |
| CMF (ICLR'24) | 48.74 | 48.41 | 48.67 | 48.35 | 48.83 | 48.61 | 48.57 | 48.58 | 48.80 | 48.56 | 48.61±0.1 |
| DATTA (ECCV'24) | 35.97 | 37.58 | 33.45 | 37.68 | 35.69 | 32.80 | 31.88 | 36.86 | 28.88 | 34.81 | 34.56±2.7 |
| PeTTA (NeurIPS'24) | 31.57 | 31.57 | 31.44 | 31.59 | 31.56 | 31.36 | 31.60 | 31.40 | 31.65 | 31.76 | 31.55±0.1 |
| ETA (ICML'22) | 43.68 | 43.69 | 42.97 | 42.91 | 44.19 | 44.12 | 43.84 | 43.79 | 43.88 | 43.03 | 43.61±0.4 |
| + RDumb (NeurIPS'23) | 46.44 | 46.09 | 46.48 | 46.06 | 46.54 | 46.39 | 46.40 | 46.46 | 46.75 | 46.31 | 46.39±0.2 |
| + ASR (Ours) | 47.50 | 46.89 | 47.10 | 46.89 | 47.51 | 47.43 | 47.26 | 47.22 | 47.15 | 46.79 | 47.17±0.2 |
| ROID (WACV'24) | 48.66 | 48.53 | 48.57 | 48.47 | 48.66 | 48.56 | 48.56 | 48.53 | 48.66 | 48.56 | 48.58±0.1 |
| + RDumb (NeurIPS'23) | 48.01 | 47.92 | 48.07 | 47.90 | 48.02 | 47.96 | 48.04 | 48.02 | 48.10 | 48.00 | 48.00±0.1 |
| + COME (ICLR'25) | 0.17 | 0.32 | 0.39 | 0.33 | 0.31 | 0.42 | 0.29 | 0.85 | 0.75 | 0.28 | 0.41±0.2 |
| + ReservoirTTA (NeurIPS'25) | 47.40 | 47.42 | 47.23 | 47.30 | 47.38 | 47.33 | 47.47 | 47.30 | 47.28 | 47.28 | 47.34±0.1 |
| + ASR (Ours) | 49.76 | 49.31 | 49.40 | 49.20 | 49.78 | 49.63 | 49.42 | 49.60 | 49.54 | 49.32 | 49.50±0.2 |

Table 31: Accuracy (%) on CIN-C over ten random permutations of the corruption order.

| Method | 1 | 2 | 3 | 4 | 5 | 6 | 7 | 8 | 9 | 10 | Mean |
|---|---|---|---|---|---|---|---|---|---|---|---|
| Source | 18.01 | 18.01 | 18.01 | 18.01 | 18.01 | 18.01 | 18.01 | 18.01 | 18.01 | 18.01 | 18.01±0.0 |
| RMT (CVPR'23) | 46.53 | 44.99 | 42.80 | 44.17 | 45.95 | 44.01 | 43.88 | 43.59 | 42.99 | 42.92 | 44.18±1.2 |
| + Source-free | 42.07 | 38.68 | 32.79 | 36.20 | 40.16 | 37.47 | 37.30 | 34.04 | 32.34 | 33.08 | 36.41±3.2 |
| RoTTA (CVPR'23) | 29.33 | 32.30 | 27.21 | 27.16 | 29.09 | 32.17 | 30.24 | 30.19 | 30.41 | 28.96 | 29.71±1.7 |
| SANTA (TMLR'23) | 39.60 | 39.38 | 39.28 | 39.39 | 39.54 | 39.52 | 39.37 | 39.44 | 39.42 | 39.16 | 39.40±0.1 |
| LAW (WACV'24) | 21.82 | 15.81 | 1.47 | 15.34 | 24.20 | 16.41 | 9.00 | 13.86 | 13.06 | 2.74 | 13.37±6.9 |
| ViDA (ICLR'24) | 17.86 | 17.82 | 17.60 | 17.77 | 17.77 | 17.85 | 17.78 | 17.80 | 17.79 | 17.60 | 17.76±0.1 |
| DPLOT (CVPR'24) | 36.98 | 34.19 | 30.29 | 30.54 | 34.04 | 27.87 | 33.05 | 30.02 | 29.84 | 29.38 | 31.62±2.7 |
| PALM (AAAI'25) | 19.09 | 14.37 | 3.18 | 16.71 | 22.72 | 13.95 | 7.52 | 10.76 | 8.30 | 4.23 | 12.08±6.1 |
| EATA (ICML'22) | 47.70 | 47.29 | 47.63 | 47.12 | 47.89 | 47.63 | 47.51 | 47.52 | 47.71 | 47.41 | 47.54±0.2 |
| + COME (ICLR'25) | 44.26 | 43.69 | 44.11 | 43.62 | 44.41 | 44.11 | 43.86 | 44.24 | 44.55 | 44.05 | 44.09±0.3 |
| CoTTA (CVPR'22) | 39.10 | 36.39 | 31.57 | 35.61 | 38.40 | 36.41 | 36.15 | 34.40 | 32.14 | 32.73 | 35.29±2.4 |
| SAR (ICLR'23) | 40.75 | 40.57 | 40.08 | 39.04 | 40.89 | 39.68 | 40.30 | 39.52 | 39.44 | 40.40 | 40.07±0.6 |
| PETAL (CVPR'23) | 26.41 | 23.71 | 17.45 | 22.96 | 24.88 | 22.74 | 22.97 | 20.34 | 17.88 | 19.07 | 21.84±2.9 |
| CMF (ICLR'24) | 48.44 | 48.06 | 48.28 | 48.03 | 48.57 | 48.33 | 48.15 | 48.27 | 48.44 | 48.19 | 48.28±0.2 |
| DATTA (ECCV'24) | 7.94 | 3.30 | 2.42 | 1.81 | 3.75 | 2.46 | 1.66 | 2.07 | 2.72 | 2.37 | 3.25±1.7 |
| PeTTA (NeurIPS'24) | 31.49 | 31.62 | 31.60 | 31.55 | 31.57 | 31.69 | 31.66 | 31.47 | 31.69 | 31.74 | 31.61±0.1 |
| ETA (ICML'22) | 43.75 | 43.61 | 43.29 | 43.09 | 44.32 | 43.95 | 43.90 | 43.56 | 43.72 | 43.08 | 43.63±0.4 |
| + RDumb (NeurIPS'23) | 46.21 | 45.92 | 46.34 | 45.68 | 46.20 | 46.12 | 46.19 | 46.18 | 46.42 | 46.00 | 46.13±0.2 |
| + ASR (Ours) | 47.31 | 46.62 | 46.93 | 46.47 | 47.04 | 47.00 | 46.72 | 46.87 | 46.81 | 46.50 | 46.83±0.2 |
| ROID (WACV'24) | 48.46 | 48.32 | 48.25 | 48.11 | 48.28 | 48.27 | 48.17 | 48.24 | 48.24 | 48.16 | 48.25±0.1 |
| + RDumb (NeurIPS'23) | 47.64 | 47.60 | 47.77 | 47.60 | 47.64 | 47.72 | 47.58 | 47.72 | 47.75 | 47.66 | 47.67±0.1 |
| + COME (ICLR'25) | 0.15 | 0.38 | 0.29 | 0.29 | 0.37 | 0.31 | 0.31 | 0.37 | 0.37 | 0.28 | 0.31±0.1 |
| + ReservoirTTA (NeurIPS'25) | 46.93 | 47.08 | 46.90 | 46.91 | 46.96 | 46.99 | 47.04 | 46.88 | 46.98 | 46.88 | 46.96±0.1 |
| + ASR (Ours) | 49.37 | 48.98 | 49.07 | 48.84 | 49.45 | 49.27 | 49.04 | 49.27 | 49.16 | 48.99 | 49.14±0.2 |

Table 32: Accuracy (%) on *non-i.i.d.* CIN-C over ten random permutations of the corruption order.

| Method | Recurring visit 1 | 2 | 3 | 4 | 5 | 6 | 7 | 8 | 9 | 10 | 11 | 12 | 13 | 14 | 15 | 16 | 17 | 18 | 19 | 20 | Mean |
|---|---|---|---|---|---|---|---|---|---|---|---|---|---|---|---|---|---|---|---|---|---|
| Source | 3.08 | 3.08 | 3.08 | 3.08 | 3.08 | 3.08 | 3.08 | 3.08 | 3.08 | 3.08 | 3.08 | 3.08 | 3.08 | 3.08 | 3.08 | 3.08 | 3.08 | 3.08 | 3.08 | 3.08 | 3.08±0.0 |
| RMT | 27.63 | 33.91 | 37.28 | 39.08 | 39.99 | 40.78 | 41.18 | 41.36 | 41.72 | 41.76 | 41.92 | 42.01 | 42.02 | 41.96 | 42.05 | 42.10 | 42.08 | 42.08 | 42.08 | 42.09 | 40.24±3.5 |
| + Source-free | 27.63 | 34.15 | 37.14 | 38.11 | 38.70 | 38.87 | 39.22 | 39.43 | 39.40 | 39.60 | 39.60 | 39.60 | 39.71 | 39.80 | 39.76 | 39.68 | 39.74 | 39.79 | 39.76 | 39.76 | 38.47±2.8 |
| RoTTA | 12.45 | 17.22 | 19.19 | 20.77 | 19.92 | 21.29 | 21.88 | 21.23 | 19.71 | 19.25 | 18.70 | 18.00 | 17.33 | 16.91 | 16.20 | 15.58 | 14.97 | 14.44 | 13.97 | 12.96 | 17.60±2.8 |
| SANTA | 27.28 | 27.75 | 27.30 | 27.20 | 27.10 | 26.94 | 27.04 | 26.81 | 26.91 | 26.59 | 26.49 | 26.42 | 26.53 | 26.39 | 26.25 | 26.38 | 26.18 | 26.17 | 26.25 | 25.94 | 26.70±0.5 |
| LAW | 23.83 | 30.62 | 31.98 | 32.03 | 31.60 | 31.11 | 30.75 | 30.34 | 30.06 | 29.76 | 29.65 | 29.52 | 29.44 | 29.36 | 29.37 | 29.35 | 29.41 | 29.37 | 29.31 | 29.24 | 29.81±1.6 |
| ViDA | 3.09 | 3.09 | 3.08 | 3.08 | 3.07 | 3.05 | 3.02 | 3.03 | 3.02 | 3.02 | 3.02 | 2.99 | 2.97 | 2.95 | 2.92 | 2.91 | 2.89 | 2.89 | 2.88 | 2.84 | 2.99±0.1 |
| DPLOT | 30.16 | 33.83 | 35.76 | 36.61 | 36.94 | 37.07 | 37.18 | 37.14 | 37.28 | 37.41 | 37.35 | 37.33 | 37.36 | 37.38 | 37.35 | 37.34 | 37.35 | 37.36 | 37.37 | 37.39 | 36.65±1.7 |
| PALM | 24.66 | 31.70 | 32.18 | 31.71 | 31.29 | 30.79 | 30.71 | 30.74 | 30.76 | 30.76 | 30.81 | 30.78 | 30.78 | 30.86 | 30.82 | 30.91 | 30.93 | 30.92 | 30.96 | 30.98 | 30.70±1.4 |
| EATA | 31.31 | 36.38 | 36.70 | 36.90 | 36.98 | 36.67 | 36.56 | 36.60 | 36.73 | 36.79 | 36.52 | 36.56 | 36.47 | 36.40 | 36.46 | 36.52 | 36.54 | 36.45 | 36.48 | 36.35 | 36.32±1.2 |
| + COME | 30.20 | 34.59 | 34.90 | 34.52 | 34.17 | 33.88 | 33.82 | 33.44 | 33.25 | 32.99 | 32.97 | 32.82 | 32.57 | 32.40 | 32.48 | 32.52 | 32.35 | 32.10 | 32.06 | 32.06 | 33.02±1.1 |
| CoTTA | 18.78 | 24.90 | 29.02 | 31.39 | 33.47 | 34.66 | 35.47 | 35.96 | 36.28 | 36.55 | 36.70 | 36.94 | 37.05 | 37.17 | 37.24 | 37.25 | 37.20 | 37.25 | 37.24 | 37.22 | 34.39±4.8 |
| SAR | 24.38 | 31.54 | 33.42 | 34.06 | 34.40 | 34.52 | 34.70 | 34.85 | 35.00 | 35.08 | 35.11 | 35.02 | 35.02 | 35.03 | 34.94 | 34.98 | 34.93 | 34.99 | 34.97 | 34.93 | 34.09±2.4 |
| PETAL | 18.74 | 25.64 | 29.12 | 30.91 | 31.76 | 32.36 | 32.80 | 33.28 | 33.55 | 33.75 | 33.89 | 34.04 | 34.10 | 34.18 | 34.21 | 34.24 | 34.22 | 34.24 | 34.24 | 34.24 | 32.18±3.7 |
| CMF | 35.07 | 38.66 | 39.22 | 39.52 | 39.58 | 39.62 | 39.90 | 39.90 | 39.95 | 39.92 | 39.76 | 39.70 | 39.73 | 39.28 | 39.61 | 39.54 | 39.65 | 39.52 | 39.84 | 39.52 | 39.35±1.0 |
| DATTA | 20.11 | 19.67 | 19.67 | 19.67 | 19.67 | 19.67 | 19.67 | 19.67 | 19.67 | 19.67 | 19.67 | 19.67 | 19.67 | 19.67 | 19.67 | 19.67 | 19.67 | 19.67 | 19.67 | 19.67 | 19.69±0.1 |
| PeTTA | 11.91 | 12.63 | 12.61 | 12.83 | 12.65 | 13.16 | 12.96 | 12.74 | 12.72 | 12.74 | 12.77 | 12.72 | 12.74 | 12.60 | 12.48 | 12.60 | 12.88 | 12.33 | 12.73 | 12.40 | 12.65±0.3 |
| ETA | 30.64 | 35.80 | 36.56 | 36.67 | 36.76 | 36.58 | 36.45 | 36.47 | 36.28 | 36.16 | 36.08 | 36.00 | 36.06 | 36.01 | 35.96 | 35.91 | 35.86 | 35.76 | 35.82 | 35.80 | 35.88±1.2 |
| + RDumb | 30.71 | 35.95 | 36.80 | 36.73 | 35.66 | 36.30 | 31.97 | 35.98 | 36.67 | 32.71 | 34.87 | 36.66 | 34.06 | 33.88 | 36.06 | 35.83 | 33.07 | 36.51 | 36.92 | 30.94 | 34.66±2.2 |
| + ASR (Ours) | 28.68 | 33.09 | 34.65 | 33.52 | 33.00 | 34.86 | 36.24 | 37.32 | 38.02 | 38.60 | 38.79 | 38.86 | 38.89 | 38.86 | 38.97 | 39.23 | 39.07 | 39.16 | 39.07 | 39.10 | 36.90±2.9 |
| ROID | 35.32 | 37.74 | 38.21 | 37.96 | 38.00 | 38.16 | 38.02 | 38.02 | 38.10 | 38.08 | 38.43 | 38.51 | 37.95 | 38.20 | 38.15 | 38.16 | 37.98 | 38.16 | 37.97 | 38.02 | 37.96±0.6 |
| + RDumb | 35.60 | 38.28 | 38.34 | 35.08 | 38.02 | 38.34 | 35.21 | 37.61 | 37.72 | 35.16 | 37.72 | 38.48 | 36.00 | 37.49 | 38.25 | 37.16 | 37.32 | 38.34 | 37.70 | 35.75 | 37.18±1.2 |
| + COME | 4.98 | 0.08 | 0.08 | 0.08 | 0.08 | 0.08 | 0.08 | 0.08 | 0.08 | 0.08 | 0.08 | 0.08 | 0.08 | 0.08 | 0.08 | 0.08 | 0.08 | 0.08 | 0.08 | 0.08 | 0.33±1.1 |
| + ReservoirTTA | 31.54 | 36.58 | 36.97 | 36.94 | 36.92 | 36.90 | 36.88 | 36.96 | 37.14 | 37.33 | 37.23 | 37.24 | 37.52 | 37.62 | 37.36 | 37.24 | 37.12 | 37.05 | 37.11 | 37.09 | 36.84±1.2 |
| + ASR (Ours) | 35.66 | 39.42 | 39.64 | 40.42 | 41.03 | 41.40 | 41.74 | 41.83 | 41.87 | 42.20 | 42.46 | 42.48 | 42.76 | 42.12 | 42.06 | 42.60 | 42.67 | 42.86 | 43.08 | 42.96 | 41.56±1.7 |

Table 33: Accuracy (%) on IN-C across 20 recurring visits of the domain sequence.

| Method | Recurring visit 1 | 2 | 3 | 4 | 5 | 6 | 7 | 8 | 9 | 10 | 11 | 12 | 13 | 14 | 15 | 16 | 17 | 18 | 19 | 20 | Mean |
|---|---|---|---|---|---|---|---|---|---|---|---|---|---|---|---|---|---|---|---|---|---|
| Source | 32.52 | 32.52 | 32.52 | 32.52 | 32.52 | 32.52 | 32.52 | 32.52 | 32.52 | 32.52 | 32.52 | 32.52 | 32.52 | 32.52 | 32.52 | 32.52 | 32.52 | 32.52 | 32.52 | 32.52 | 32.52±0.0 |
| RMT | 43.16 | 45.54 | 46.26 | 46.51 | 46.74 | 46.78 | 46.81 | 46.83 | 46.88 | 46.88 | 46.87 | 46.88 | 46.90 | 46.89 | 46.90 | 46.90 | 46.89 | 46.88 | 46.86 | 46.87 | 46.56±0.8 |
| + Source-free | 42.24 | 43.73 | 43.82 | 43.92 | 43.94 | 43.94 | 43.81 | 43.86 | 43.87 | 43.88 | 43.81 | 43.81 | 43.80 | 43.76 | 43.75 | 43.71 | 43.68 | 43.68 | 43.65 | 43.65 | 43.72±0.4 |
| RoTTA | 39.89 | 43.06 | 44.03 | 44.36 | 44.34 | 43.94 | 43.66 | 43.26 | 42.71 | 42.07 | 41.41 | 40.63 | 40.04 | 39.38 | 38.66 | 37.85 | 37.04 | 36.20 | 35.23 | 34.34 | 40.61±3.1 |
| SANTA | 41.52 | 41.68 | 41.74 | 41.66 | 41.75 | 41.80 | 41.68 | 41.59 | 41.65 | 41.54 | 41.44 | 41.47 | 41.39 | 41.58 | 41.50 | 41.41 | 41.42 | 41.44 | 41.40 | 41.29 | 41.54±0.1 |
| LAW | 40.19 | 35.70 | 32.19 | 30.78 | 30.25 | 30.01 | 29.85 | 29.78 | 29.75 | 29.72 | 29.68 | 29.67 | 29.65 | 29.67 | 29.67 | 29.67 | 29.66 | 29.66 | 29.66 | 29.67 | 30.74±2.6 |
| ViDA | 0.01 | 0.01 | 0.01 | 0.01 | 0.01 | 0.01 | 0.01 | 0.01 | 0.01 | 0.01 | 0.01 | 0.01 | 0.01 | 0.01 | 0.01 | 0.01 | 0.01 | 0.01 | 0.01 | 0.01 | 0.01±0.0 |
| DPLOT | 42.09 | 42.46 | 42.35 | 42.26 | 42.24 | 42.12 | 42.17 | 42.16 | 42.14 | 42.12 | 42.12 | 42.12 | 42.12 | 42.10 | 42.10 | 42.11 | 42.11 | 42.10 | 42.09 | 42.10 | 42.16±0.1 |
| PALM | 13.86 | 1.74 | 1.42 | 1.42 | 1.42 | 1.42 | 1.42 | 1.42 | 1.42 | 1.42 | 1.42 | 1.42 | 1.42 | 1.42 | 1.42 | 1.42 | 1.42 | 1.42 | 1.42 | 1.42 | 2.06±2.7 |
| EATA | 41.62 | 42.42 | 42.21 | 41.77 | 41.99 | 41.96 | 41.96 | 41.58 | 41.57 | 41.34 | 41.30 | 41.46 | 41.50 | 41.54 | 41.10 | 41.23 | 41.37 | 41.57 | 41.43 | 41.32 | 41.61±0.3 |
| + COME | 42.94 | 45.13 | 45.46 | 45.73 | 45.46 | 45.30 | 45.24 | 45.02 | 45.30 | 45.11 | 45.48 | 45.25 | 45.16 | 45.04 | 45.25 | 44.89 | 44.87 | 44.78 | 44.91 | 44.91 | 45.11±0.6 |
| CoTTA | 41.76 | 45.70 | 46.79 | 46.90 | 46.72 | 46.30 | 45.85 | 45.42 | 44.99 | 44.62 | 44.30 | 43.91 | 43.43 | 42.87 | 42.49 | 41.92 | 41.54 | 41.22 | 40.82 | 40.55 | 43.91±2.1 |
| SAR | 40.86 | 42.94 | 43.26 | 43.15 | 42.93 | 42.50 | 42.06 | 41.53 | 40.87 | 40.17 | 39.47 | 38.76 | 38.01 | 37.23 | 36.49 | 35.70 | 34.97 | 34.22 | 33.59 | 33.11 | 39.09±3.4 |
| PETAL | 0.03 | 0.03 | 0.02 | 0.03 | 0.03 | 0.09 | 0.39 | 0.39 | 0.39 | 0.39 | 0.39 | 0.39 | 0.39 | 0.39 | 0.39 | 0.39 | 0.39 | 0.39 | 0.39 | 0.39 | 0.28±0.2 |
| CMF | 44.69 | 45.21 | 45.42 | 45.25 | 45.38 | 45.59 | 44.92 | 45.14 | 45.31 | 45.55 | 45.52 | 45.52 | 45.18 | 45.26 | 44.94 | 45.00 | 45.02 | 45.13 | 45.27 | 45.46 | 45.25±0.3 |
| DATTA | 33.75 | 3.50 | 1.33 | 0.88 | 0.85 | 0.84 | 0.84 | 0.84 | 0.84 | 0.84 | 0.84 | 0.84 | 0.84 | 0.84 | 0.84 | 0.84 | 0.84 | 0.84 | 0.84 | 0.84 | 2.65±7.2 |
| PeTTA | 39.56 | 42.05 | 42.84 | 43.02 | 43.12 | 43.27 | 43.26 | 43.23 | 43.12 | 43.08 | 42.94 | 42.95 | 42.91 | 42.93 | 42.98 | 42.96 | 42.81 | 42.75 | 42.70 | 42.69 | 42.76±0.8 |
| ETA | 41.24 | 40.92 | 40.30 | 39.86 | 39.07 | 38.49 | 38.26 | 37.64 | 37.28 | 36.96 | 36.51 | 36.14 | 36.28 | 35.94 | 35.67 | 35.47 | 35.09 | 34.74 | 34.37 | 34.21 | 37.22±2.1 |
| + RDumb | 40.93 | 41.36 | 42.11 | 41.66 | 41.18 | 41.46 | 41.28 | 41.84 | 41.31 | 40.78 | 41.90 | 42.09 | 41.70 | 40.85 | 41.60 | 41.44 | 41.59 | 41.52 | 40.75 | 41.59 | 41.45±0.4 |
| + ASR (Ours) | 40.61 | 41.46 | 41.49 | 41.74 | 41.79 | 42.04 | 41.82 | 41.70 | 41.84 | 41.88 | 41.74 | 41.76 | 41.60 | 41.38 | 41.27 | 41.36 | 41.16 | 41.28 | 41.36 | 41.32 | 41.53±0.3 |
| ROID | 46.02 | 46.22 | 46.03 | 46.33 | 46.22 | 46.29 | 46.14 | 45.94 | 46.32 | 46.04 | 46.12 | 46.26 | 46.03 | 46.13 | 46.16 | 46.20 | 46.23 | 46.22 | 46.19 | 46.17 | 46.16±0.1 |
| + RDumb | 46.07 | 46.34 | 46.32 | 46.25 | 46.24 | 46.23 | 46.16 | 46.04 | 46.07 | 46.14 | 45.86 | 45.86 | 45.94 | 45.79 | 45.75 | 45.80 | 45.81 | 45.68 | 45.62 | 45.62 | 45.99±0.2 |
| + COME | 1.02 | 0.39 | 0.39 | 0.39 | 0.39 | 0.39 | 0.39 | 0.39 | 0.39 | 0.39 | 0.39 | 0.39 | 0.39 | 0.39 | 0.39 | 0.39 | 0.39 | 0.39 | 0.39 | 0.39 | 0.42±0.1 |
| + ReservoirTTA | 44.71 | 45.12 | 45.16 | 45.40 | 45.82 | 45.76 | 45.85 | 45.89 | 45.84 | 45.82 | 45.76 | 45.58 | 45.60 | 45.61 | 45.67 | 45.77 | 45.60 | 45.72 | 45.73 | 45.76 | 45.61±0.3 |
| + ASR (Ours) | 46.13 | 46.50 | 46.53 | 46.63 | 46.52 | 46.50 | 46.49 | 46.44 | 46.70 | 46.55 | 46.56 | 46.63 | 46.53 | 46.54 | 46.48 | 46.39 | 46.48 | 46.33 | 46.40 | 46.32 | 46.49±0.1 |

Table 34: Accuracy (%) on IN-D109 across 20 recurring visits of the domain sequence.

| Transition speed | 1000 | | | 2000 | | | 5000 | | | Acc. (%) |
|---|---|---|---|---|---|---|---|---|---|---|
| Corruption ordering | 43 | 44 | 45 | 43 | 44 | 45 | 43 | 44 | 45 | Mean |
| Source | 54.74 | 55.18 | 54.47 | 54.97 | 55.03 | 54.77 | 55.09 | 54.88 | 55.12 | 54.92±0.2 |
| CMF (ICLR'24) | 60.68 | 60.96 | 60.59 | 61.51 | 61.74 | 61.51 | 62.66 | 62.52 | 61.52 | 61.52±0.7 |
| CMAE (CVPR'24) | 48.11 | 49.94 | 48.19 | 50.68 | 50.44 | 52.57 | 51.04 | 54.89 | 54.50 | 51.15±2.3 |
| REM (ICML'25) | 65.82 | 66.12 | 65.79 | 66.14 | 66.23 | 66.05 | 66.93 | 66.02 | 66.36 | 66.16±0.3 |
| ETA (ICML'22) | 47.95 | 47.68 | 47.47 | 48.71 | 15.74 | 48.44 | 49.92 | 49.78 | 49.93 | 45.07±10.4 |
| + RDumb (NeurIPS'23) | 59.25 | 59.56 | 59.13 | 59.74 | 60.07 | 59.74 | 60.76 | 60.75 | 60.91 | 59.99±0.6 |
| + ASR (Ours) | 59.62 | 59.99 | 59.59 | 60.34 | 60.69 | 60.46 | 61.57 | 61.33 | 61.62 | 60.58±0.7 |
| ROID (WACV'24) | 60.01 | 60.32 | 59.88 | 60.67 | 60.99 | 60.68 | 61.66 | 61.64 | 61.82 | 60.85±0.7 |
| + RDumb (NeurIPS'23) | 59.78 | 60.09 | 59.65 | 60.44 | 60.75 | 60.43 | 61.35 | 61.39 | 61.56 | 60.60±0.7 |
| + ASR (Ours) | 60.61 | 60.86 | 60.51 | 61.28 | 61.54 | 61.30 | 62.46 | 62.21 | 62.52 | 61.48±0.7 |

Table 35: Performance comparison with state-of-the-art methods on CCC-Easy across nine variants: three corruption transition speeds (1000 / 2000 / 5000) and three corruption orderings that are determined by random seeds (43 / 44 / 45).

| Transition speed | 1000 | | | 2000 | | | 5000 | | | Acc. (%) |
|---|---|---|---|---|---|---|---|---|---|---|
| Corruption ordering | 43 | 44 | 45 | 43 | 44 | 45 | 43 | 44 | 45 | Mean |
| Source | 41.76 | 41.47 | 40.49 | 42.32 | 41.10 | 41.49 | 42.15 | 42.39 | 42.51 | 41.74±0.6 |
| CMF (ICLR'24) | 52.17 | 51.63 | 51.87 | 34.21 | 53.49 | 53.59 | 55.41 | 55.33 | 55.82 | 51.50±6.3 |
| CMAE (CVPR'24) | 41.76 | 36.63 | 41.88 | 43.16 | 40.52 | 44.40 | 45.02 | 46.41 | 51.52 | 43.48±3.9 |
| REM (ICML'25) | 57.34 | 57.23 | 56.92 | 58.36 | 57.28 | 57.68 | 58.94 | 58.41 | 59.71 | 57.99±0.9 |
| ETA (ICML'22) | 34.62 | 23.74 | 28.04 | 34.04 | 34.54 | 36.05 | 35.36 | 38.07 | 38.97 | 33.71±4.6 |
| + RDumb (NeurIPS'23) | 49.02 | 48.88 | 48.57 | 50.77 | 50.16 | 50.28 | 51.97 | 52.25 | 52.58 | 50.50±1.4 |
| + ASR (Ours) | 49.86 | 49.69 | 49.57 | 51.96 | 51.39 | 51.46 | 53.31 | 53.55 | 53.92 | 51.63±1.6 |
| ROID (WACV'24) | 50.72 | 50.67 | 50.39 | 52.61 | 51.91 | 52.00 | 53.62 | 53.72 | 54.08 | 52.19±1.3 |
| + RDumb (NeurIPS'23) | 50.23 | 50.20 | 49.81 | 52.11 | 51.45 | 51.39 | 53.09 | 53.26 | 53.62 | 51.68±1.3 |
| + ASR (Ours) | 52.14 | 52.08 | 51.98 | 53.91 | 53.07 | 53.26 | 55.03 | 54.95 | 55.57 | 53.55±1.3 |

Table 36: Performance comparison with state-of-the-art methods on CCC-Medium across nine variants: three corruption transition speeds (1000 / 2000 / 5000) and three corruption orderings that are determined by random seeds (43 / 44 / 45).

| Transition speed | 1000 | | | 2000 | | | 5000 | | | Acc. (%) |
|---|---|---|---|---|---|---|---|---|---|---|
| Corruption ordering | 43 | 44 | 45 | 43 | 44 | 45 | 43 | 44 | 45 | Mean |
| Source | 14.40 | 15.44 | 15.40 | 14.16 | 15.38 | 14.10 | 13.90 | 15.31 | 15.40 | 14.83±0.6 |
| CMF (ICLR'24) | 1.22 | 0.30 | 2.74 | 2.17 | 0.14 | 0.85 | 3.23 | 0.13 | 5.34 | 1.79±1.7 |
| CMAE (CVPR'24) | 26.47 | 26.78 | 22.70 | 25.95 | 28.33 | 24.96 | 26.60 | 30.27 | 30.20 | 26.92±2.3 |
| REM (ICML'25) | 3.80 | 8.53 | 7.03 | 5.58 | 5.94 | 9.39 | 10.67 | 38.45 | 9.31 | 10.97±9.9 |
| ETA (ICML'22) | 1.34 | 0.33 | 1.66 | 0.99 | 1.34 | 1.02 | 1.97 | 1.58 | 0.79 | 1.22±0.5 |
| + RDumb (NeurIPS'23) | 22.41 | 24.43 | 22.01 | 23.52 | 25.54 | 23.39 | 22.16 | 23.52 | 22.42 | 23.27±1.1 |
| + ASR (Ours) | 24.67 | 25.88 | 24.14 | 23.93 | 25.21 | 24.26 | 22.76 | 23.96 | 25.20 | 24.45±0.9 |
| ROID (WACV'24) | 11.74 | 23.23 | 1.00 | 25.75 | 9.08 | 25.10 | 12.49 | 6.75 | 13.55 | 14.30±8.2 |
| + RDumb (NeurIPS'23) | 24.17 | 25.40 | 23.76 | 25.05 | 26.62 | 24.84 | 26.25 | 27.37 | 28.01 | 25.72±1.4 |
| + ASR (Ours) | 27.69 | 28.76 | 27.31 | 27.62 | 28.82 | 27.52 | 27.58 | 28.67 | 28.85 | 28.09±0.6 |

Table 37: Performance comparison with state-of-the-art methods on CCC-Hard across nine variants: three corruption transition speeds (1000 / 2000 / 5000) and three corruption orderings that are determined by random seeds (43 / 44 / 45).

