# OpenReview forum: "When and Where to Reset Matters for Long-Term Test-Time Adaptation"
_ICLR.cc/2026/Conference — ICLR 2026 Poster_

### Official Review · Reviewer_8GCC · 2025-10-22

**Soundness:** 1
**Presentation:** 3
**Contribution:** 2
**Rating:** 2
**Confidence:** 4

**Summary:**

This paper proposes the Adaptive and Selective Reset (ASR) scheme to address the problem of model collapse in long-term Test-Time Adaptation (TTA). The main contributions are: 1) The ASR mechanism dynamically determines when and which parts of the model to reset; 2) An importance-aware knowledge recovery regularizer based on Fisher information; 3) Dynamic adjustment of hyperparameters according to domain differences to enhance adaptability. Experiments show that ASR performs well in multiple benchmark tests and significantly improves the stability and adaptability of the model.

**Strengths:**

1. Clear Structure: The article is well-organized with distinct logical sections, progressing from introduction to methodology and then experiments, which allows readers to easily grasp the research content and contributions.

2. Professional Expression: Accurate terminology, rigorous mathematical formulas, and clear presentation of charts and experimental results enhance the article's scientific validity and readability.

3. Dynamic and Selective Reset Mechanism: The ASR scheme dynamically determines the timing and scope of resets, effectively preventing model collapse while minimizing knowledge loss.

4. Enhanced Comprehensive Adaptability: By integrating knowledge recovery and dynamic adjustment mechanisms, ASR improves the model's adaptability in complex environments and achieves outstanding performance.

**Weaknesses:**

1.The author aims to address the challenge of model collapse in long-term TTA. The proposed ASR module triggers resets based on the concentration of predicted classes. However, there is an issue here. In long-term TTA, model collapse can lead to increased concentration of predicted classes. Yet, in real-world scenarios, such increased concentration could stem from various factors. For instance, the distribution of test data is often non-stationary and imbalanced. Test data may temporarily concentrate on a few classes over time, which is not due to any issue with the model itself. The author needs to analyze this aspect.

2.In Section 3.2, the calculation of prediction concentration seems to indicate that a higher value corresponds to lower class prediction concentration. However, the reset criterion set by the author is that prediction concentration is higher than cumulative concentration. Is there an error here? The author is advised to double-check.

**Questions:**

It is suggested that the author, when addressing the issue of when to reset, should first rule out other non-model factors that may cause the predicted classes to concentrate on a few categories.

---

> ### Author Response · Authors · 2025-11-21
> **Response to Reviewer 8GCC**
>
> > [W1] The author aims to address the challenge of model collapse in long-term TTA. The proposed ASR module triggers resets based on the concentration of predicted classes. However, there is an issue here. In long-term TTA, model collapse can lead to increased concentration of predicted classes. Yet, in real-world scenarios, such increased concentration could stem from various factors. For instance, the distribution of test data is often non-stationary and imbalanced. Test data may temporarily concentrate on a few classes over time, which is not due to any issue with the model itself. The author needs to analyze this aspect.
>
> We addressed this concern in `L427-L429`, where we noted that TTA models can be biased and collapse under extreme label imbalance, even when their predictions are correct, because the update signal is dominated by a small set of labels.
>
> In this context, a reset triggered by temporary label imbalance, which we refer to as a *false-positive reset*, can have a beneficial effect. It interrupts the **biased update signal** and prevents the model from drifting toward collapse.
>
> To validate the potential benefits of false-positive resets, we ran a controlled experiment in a non-i.i.d. label scenario, specifically using the first split from `Table D.5` with ROID as the baseline. We prepared a batch with i.i.d. labels in advance. Whenever a reset was triggered, we computed the prediction concentration from this batch and applied a threshold (determined according to `L168`) to decide whether to perform the reset, ensuring that only *true-positive resets* occurred. Higher thresholds favor true-positive resets; e.g., -log(700) is more likely than -log(900). By adjusting the threshold, we compared the resulting accuracy with and without false-positive resets.
>
> We observed that allowing false-positive resets resulted in higher performance. This confirms that, as argued earlier, reducing **biased parameter updates** is crucial in long-term TTA, and false-positive resets support this by interrupting these signals, ultimately improving long-term adaptation.
>
> | Threshold | -log(700) | -log(800) | -log(900) | ASR (Ours) |
> | :-- | :-: | :-: | :-: | :-: |
> | Acc. (%) | 48.18 | 48.23 | 48.59 | 49.37 |
>
> As shown in `Fig. 7`, ASR remains stable even under severe label imbalance, demonstrating that such resets do not harm and instead support reliable long-term adaptation.
>
> ---
>
> > [W2] In Section 3.2, the calculation of prediction concentration seems to indicate that a higher value corresponds to lower class prediction concentration. However, the reset criterion set by the author is that prediction concentration is higher than cumulative concentration. Is there an error here? The author is advised to double-check.
>
> We believe the reviewer may have misinterpreted the sign of our concentration $\mathcal{C}_ {t}$ in `Eq. (1)`. The concentration $\mathcal{C}_ {t}$ is defined as the **negative entropy** of the averaged predictive distribution. Under this definition, $\mathcal{C}_ {t}$ increases as the prediction distribution becomes more concentrated.
>
> For example, with $C = 3$ classes, a uniform distribution $\hat{p}_ {t}=(1/3, 1/3, 1/3)$ gives $\mathcal{C}_ {t} \approx -1.10$, whereas a skewed distribution $\hat{p}_ {t}=(0.9, 0.05, 0.05)$ gives $\mathcal{C}_ {t} \approx -0.39$. Therefore, a larger $\mathcal{C}_ {t}$ clearly indicates higher prediction concentration, and our reset criterion $\mathcal{C}_ {t} > \bar{\mathcal{C}}_ {t-1}$ correctly triggers when predictions are highly concentrated.
>
> ---
>
> > [Q] It is suggested that the author, when addressing the issue of when to reset, should first rule out other non-model factors that may cause the predicted classes to concentrate on a few categories.
>
> While it could in principle be beneficial to rule out non-model factors before a reset, our analysis in response to `[W1]` and the results in `Fig. 7` together show that this is not necessary in practice, because resets triggered by such factors do not harm and may even help maintain stable adaptation.

---

### Official Review · Reviewer_wwgg · 2025-10-26

**Soundness:** 3
**Presentation:** 2
**Contribution:** 3
**Rating:** 4
**Confidence:** 4

**Summary:**

This paper addresses the problem of model collapse in long-term continual test-time adaptation (TTA) due to error accumulation. The authors propose an Adaptive and Selective Reset (ASR) scheme that dynamically determines when to reset by monitoring prediction concentration and where to reset by selectively resetting layers based on the estimated collapse risk. The method is supplemented by an importance-aware regularizer to recover lost knowledge and an on-the-fly mechanism to adjust adaptation based on domain discrepancy.

**Strengths:**

The primary strength is the novel ASR mechanism, which offers a more motivated and flexible alternative to naive periodic resets by dynamically linking the reset trigger and scope to a quantifiable measure of model collapse (prediction concentration). This adaptive approach is intuitive and addresses a clear limitation of prior work. The method is validated by extensive experiments on long-term TTA benchmarks, demonstrating significant performance gains, particularly on the challenging CCC-Hard dataset.

**Weaknesses:**

1. The definition of "prediction concentration" (Eq. 1), which is central to the reset trigger, is based on the entropy of average logits. This metric seems sensitive to factors not fully explored: it's unclear if it's robust to logits of different magnitudes, and its dependency on batch composition (batch size, class distribution) is a concern. Furthermore, the supporting correlation in Figure 3 lacks context, as the dataset and settings used to generate it are not specified.
Moreover, the paper's justification for the method's effectiveness (e.g., Line 474) relies on the assumption that TTA models "strengthen predictive confidence" for the imbalance label setting. This holds for entropy minimization methods but not necessarily for all TTA strategies. This raises a significant question about whether ASR can reliably detect collapse in models using other objectives, such as consistency regularization, potentially limiting its applicability.

2. While the "importance-aware knowledge recovery" component is intended to address catastrophic forgetting, the experiments lack direct evidence of its efficacy. In recurring domain scenarios (e.g., IN-C 20 visits), it is not shown whether the model effectively re-utilizes previously learned knowledge from a specific domain or simply re-adapts more successfully from a cleaner reset state.

3. The comparisons in Table 1 are difficult to interpret. As ASR is presented as an add-on component, it's unclear if it should be compared only against RDumb or also against other SOTA add-ons like COME (which is listed but not directly compared, e.g., COME+ETA vs. ASR+ETA). To validate its general utility, ASR should be evaluated on a broader range of base TTA methods, particularly those utilizing different objective functions beyond ETA and ROID.

4. The evaluation benchmarks (like CCC and CIN-C) primarily feature slow and predictable domain changes. I'm wondering if we reset the model every time the domains change (i.e., exactly at the domain boundary), is this performance an upper bound for ASR or RDumb? (e.g., in Figure B.1, which is CIN-C)  Furthermore, this raises questions about the method's robustness in more dynamic environments, where it's unclear if it would still reset appropriately when the domain changes faster, or if frequent resets would degrade adaptation performance. Evaluating on a more dynamic benchmark, such as the CDC setting from DPCore, would be a valuable test.

**Questions:**

1. In Figure 2, what do the numbers, icons, and the different colors in the top bar represent? A legend would be helpful.

2. The paper states hyperparameters were tuned on a small holdout from CCC-Hard (Line 310). Were these exact same hyperparameters then used for all other datasets and settings (CIN-C, IN-C, IN-D109, non-i.i.d.)?

3. A general question about resetting. How would a parameter-reset method like ASR be applied to TTA methods that do not primarily adapt the model parameters, such as prompt-based methods (e.g., FOA, DPCore)?

---

> ### Author Response · Authors · 2025-11-21
> **Response to Reviewer wwgg (1/4)**
>
> > [W1-1] The definition of "prediction concentration" (Eq. 1), which is central to the reset trigger, is based on the entropy of average logits. This metric seems sensitive to factors not fully explored: it's unclear if it's robust to logits of different magnitudes, and its dependency on batch composition (batch size, class distribution) is a concern.
>
> Entropy depends on the magnitude of logits, but our reset trigger is robust to this. Although prediction concentration is computed from averaged logits, we trigger resets by comparing it against the cumulative concentration. Since both change together when logits are scaled, the trigger remains insensitive to logit magnitude.
>
> To support this, we run a simple test where we multiply a scalar to the averaged logits while keeping everything else unchanged. We use ROID as the baseline and evaluate on 10% of a single split of CCC-Hard. Accuracy and the number of resets triggered are reported in the table below. As shown below, scaling the logits has minimal effect on either metric.
>
> | Logit scale | 0.1 | 1.0 (Ours) | 10.0 | RDumb |
> | :-- | :-: | :-: | :-: | :-: |
> | Acc. (%) | 16.33 | 16.46 | 16.27 | 10.93 |
> | Resets (#) | 52 | 57 | 63 | 10 |
>
> Batch size and class distribution also affect prediction concentration, but our trigger remains stable. The cumulative concentration is a moving average that gathers information from many batches, providing reliable context on batch composition. This makes the trigger robust to variations in batch size or label imbalance. We further validate this robustness in `Fig. 10` (varying batch size) and `Fig. 7` (label imbalance).
>
> ---
>
> > [W1-2] Furthermore, the supporting correlation in Figure 3 lacks context, as the dataset and settings used to generate it are not specified.
>
> We computed the correlation in `Fig. 3` using ETA on CCC-Hard, which clearly exhibits collapse and is therefore suitable for illustrating the link between collapse and prediction concentration. Moreover, prediction concentration rises as performance drops, reflecting the increasing risk of collapse. This observation supports the motivation for our `when to reset` mechanism.
>
> ---
>
> > [W1-3] Moreover, the paper's justification for the method's effectiveness (e.g., Line 474) relies on the assumption that TTA models "strengthen predictive confidence" for the imbalance label setting. This holds for entropy minimization methods but not necessarily for all TTA strategies. This raises a significant question about whether ASR can reliably detect collapse in models using other objectives, such as consistency regularization, potentially limiting its applicability.
>
> We acknowledge that the phrase *"TTA models typically strengthen predictive confidence"* mainly describes entropy-minimization methods and does not strictly apply to all TTA objectives. We clarify that **ASR does not depend on confidence sharpening, but on the concentration of predicted classes that emerges whenever noisy updates accumulate, regardless of the underlying objective**.
>
> Any online TTA algorithm updates using pseudo-labels derived from its own predictions, which are often noisy. Even when the objective does not explicitly increase confidence (e.g., consistency regularization), these noisy updates gradually accumulate and degrade performance over time. This leads to **growing prediction concentration**, the quantity monitored by ASR. Label imbalance further accelerates this by introducing additional bias into pseudo-label updates.
>
> As an example of consistency regularization-based methods, PeTTA [1] uses a teacher-student mechanism in which the teacher’s output serves as the pseudo-label. Under strong label imbalance, the teacher inevitably produces biased predictions for extended periods. These pseudo-labels guide student updates, and the updated student in turn affects the teacher via EMA. Even if confidence increases more slowly, the model still drifts toward a small set of labels. As this drift progresses, prediction concentration rises, enabling ASR to detect the emerging risk of collapse.
>
> In summary, ASR works independently of entropy minimization, relying instead on the inherent feedback loop of online self-training.

---

> > ### Comment · Reviewer_wwgg · 2025-11-24
> >
> > Regarding [W1-1], the scalar scaling experiment misses my point. I am concerned about magnitude variance *within* a batch, not global scaling.
> >
> > In Eq. (1) , you compute Softmax(Mean(Logits)). If a batch contains logits of different scales (e.g., some ~1, others ~1000), the large values will dominate the arithmetic mean, effectively ignoring the samples with smaller logits. Averaging probabilities (Mean(Softmax(Logits))) is standard because it normalizes sample contributions. Why average raw logits despite this risk?
> >
> > Regarding Figure 5, a minimum batch size of 16 is still effectively large for online TTA. Furthermore, the claim of robustness is unconvincing: ASR suffers a performance drop (\~8%) comparable to the baselines (\~9%) when reducing batch size. To properly validate this, please provide results for truly small batch sizes (e.g., 1) rather than stopping at 16.
> >
> > Regarding [W1-2], please clarify what each data point in Fig. 3 represents. Is it a single batch or an aggregation over a domain? Do these points follow a temporal sequence? Furthermore, could this pattern be an artifact of the logit averaging in Eq. (1)? If a collapsing model produces high-magnitude logits for the dominant class, those values would dominate the arithmetic mean, artificially driving the "concentration" metric.

---

> > ### Comment · Reviewer_wwgg · 2025-11-27
> >
> > Regarding [W1-3], the **new** claim that "the concentration of predicted classes" ignores the fact that successful adaptation also results in high prediction concentration.
> >
> > The method treats growing concentration as a proxy for collapse. However, how does ASR distinguish between "bad" concentration (collapse to a single class) and "good" concentration (confident, correct predictions)? For instance, methods employing distribution alignment (e.g., FOA) can achieve high concentration on the correct classes. By relying solely on the magnitude of concentration relative to a moving average, ASR risks triggering false-positive resets on models that are actually adapting successfully.

---

> ### Author Response · Authors · 2025-11-21
> **Response to Reviewer wwgg (2/4)**
>
> > [W2] While the "importance-aware knowledge recovery" component is intended to address catastrophic forgetting, the experiments lack direct evidence of its efficacy. In recurring domain scenarios (e.g., IN-C 20 visits), it is not shown whether the model effectively re-utilizes previously learned knowledge from a specific domain or simply re-adapts more successfully from a cleaner reset state.
>
> We verify whether our model effectively reuses previously learned knowledge by ablating the knowledge recovery module (`Sec. 3.4`). We conduct experiments in a domain-recurring setting on IN-C (20 visits), following the same protocol as `Table D.6`. As shown below, the ablated variant (`w/o knowledge recovery`) exhibits accuracy drops over revisits, while our method gradually improves or maintains performance, indicating effective reuse of previously learned knowledge rather than mere adaptation from a reset state.
>
> | Acc. (Revisit#) | 1 | ... | 10 | 15 | 20 | Mean |
> | :-- | :-: | :-: | :-: | :-: | :-: | :-: |
> | ETA | 30.64 | ... | 36.16 | 35.96 |  35.80 | 35.88 |
> | + ASR (Ours) | 28.68 | ... | 38.60 | 38.97 | 39.10 | 36.90 |
> | + w/o knowledge recovery | 28.64 | ... | 37.45 | 36.49 | 36.34 | 36.56 |
> | ROID | 35.32 | ... | 38.08 | 38.15 | 38.02 | 37.96 |
> | + ASR (Ours) | 35.66 | ... | 42.20 | 42.06 | 42.96 | 41.56 |
> | + w/o knowledge recovery | 35.35 | ... | 41.64 | 41.64 | 41.19 | 40.96 |
>
> To confirm that performance gains arise from previously learned knowledge, we further measure how much knowledge from previously seen domains is recovered after resets. It is measured as the gap between the current performance and the best performance achieved so far for each domain, which is then averaged across domains. Positive values indicate recovery, while negative values indicate forgetting. As shown in the table below, our method substantially reduces forgetting and improves knowledge recovery. For instance, at revisit `#10`, ETA + ASR achieves 0.58 compared to -1.94 without the recovery module, and ROID + ASR achieves 0.16 compared to -0.52 without recovery. This confirms that the knowledge recovery module effectively preserves and reuses previously learned information.
>
> | Recovery (Revisit#) | 1 | ... | 10 | 15 | 20 | Mean |
> | :-- | :-: | :-: | :-: | :-: | :-: | :-: |
> | ETA + ASR (Ours) | 0.0 | ... | 0.58 | 0.08 | 0.02 | 0.24 |
> | + w/o knowledge recovery | 0.0 | ... | -1.94 | -1.16 | -0.76 | -0.56 |
> | ROID + ASR (Ours) | 0.0 | ... | 0.16 | 0.24 | 0.01 | 0.12 |
> | + w/o knowledge recovery | 0.0 | ... | -0.52 | -0.42 | -0.1 | -0.14 |
>
> These results indicate that our model does not simply adapt from a reset state, but effectively re-utilizes previously learned knowledge in recurring domain scenarios.
>
> ---
>
> > [W3-1] The comparisons in Table 1 are difficult to interpret. As ASR is presented as an add-on component, it's unclear if it should be compared only against RDumb or also against other SOTA add-ons like COME (which is listed but not directly compared, e.g., COME+ETA vs. ASR+ETA).
>
> ASR is primarily compared with RDumb because both rely on a reset-based mechanism, whereas COME focuses on improving entropy minimization. Furthermore, We combined ASR with ETA because ASR yields stronger gains in that setting. In contrast, COME is evaluated only with EATA, so the two methods do not share a common baseline.
>
> Although they address different aspects of TTA, a fair comparison still requires using a consistent baseline. To ensure fairness, we also report matched comparisons such as COME + EATA vs. ASR + EATA and COME + ETA vs. ASR + ETA, as shown in the table below.
>
> We present results from a single split (transition speed 1000; seed 44) for each CCC level. We note that COME was not originally designed for long-term TTA, which likely explains its limited or even decreasing performance in these experiments.
>
> | Method | CCC-Hard | CCC-Medium | CCC-Easy |
> | :-- | :-: | :-: | :-: |
> | ETA | 0.67 | 22.28 | 42.13 |
> | + COME | 0.12 | 3.72 | 31.87 |
> | + ASR (Ours) | 15.01 | 40.10 | 50.33 |
> | EATA | 1.25 | 37.36 | 48.53 |
> | + COME | 0.74 | 34.81 | 46.99 |
> | + ASR (Ours) | 16.00 | 43.89 | 53.93 |

---

> > ### Comment · Reviewer_wwgg · 2025-11-27
> >
> > Regarding [W2], while I appreciate the ablation, the practical benefit of the knowledge recovery module appears negligible. For ETA, the mean accuracy gain is only 0.34% (36.56% $\to$ 36.90%), and for ROID it is only 0.60% (40.96% $\to$ 41.56%). While the internal "recovery" metric shifts from negative to positive, this does not translate into meaningful performance improvements. Given the computational cost of computing Fisher information, is this complexity justified for such marginal gains?
> >
> > Regarding [W3-1], the new comparisons using consistent baselines is good. I encourage the authors to incorporate these rows into the main paper (Table 1) to ensure a comprehensive comparison against other SOTA add-on methods.

---

> ### Author Response · Authors · 2025-11-21
> **Response to Reviewer wwgg (3/4)**
>
> > [W3-2] To validate its general utility, ASR should be evaluated on a broader range of base TTA methods, particularly those utilizing different objective functions beyond ETA and ROID.
>
> We evaluate ASR with two recent TTA methods that represent distinct objective families: i) CMF [2], which combines entropy minimization with self-consistency regularization, and ii) PeTTA [1], which uses a teacher-student consistency objective. Regardless of the objective, noisy pseudo-label updates can accumulate over time, gradually degrading performance and increasing prediction concentration. ASR counteracts this effect, making it broadly applicable across objectives. As shown in the table below, ASR consistently improves performance for both methods, confirming that its effectiveness is not limited to a specific adaptation objective.
>
> | Method | CCC-Hard | CCC-Medium | CCC-Easy |
> | :-- | :-: | :-: | :-: |
> | CMF | 1.06 | 38.77 | 48.29 |
> | + RDumb | 12.62 | 39.50 | 48.47 |
> | + ASR (Ours) | 19.57 | 40.36 | 50.49 |
> | PeTTA | 4.93 | 19.34 | 34.61 |
> | + RDumb | 6.40 | 23.73 | 37.60 |
> | + ASR (Ours) | 6.55 | 24.32 | 40.64 |
>
> ---
>
> > [W4-1] The evaluation benchmarks (like CCC and CIN-C) primarily feature slow and predictable domain changes. I'm wondering if we reset the model every time the domains change (i.e., exactly at the domain boundary), is this performance an upper bound for ASR or RDumb? (e.g., in Figure B.1, which is CIN-C)
>
> We tested whether resetting at domain boundaries represents an **upper bound** for RDumb or ASR. CIN-C is suitable for this experiment because the domains change in order. We used one split of CIN-C with the same experimental settings as `Table D.4`. In addition to accuracy, we report the average distance between each domain boundary and the nearest reset point just before the boundary.
>
> As shown in the table below, resetting at domain boundaries outperforms both RDumb (i.e., every N batches) and ASR. Since the domains are randomly ordered, knowledge from one domain rarely helps the next, which can cause negative transfer. Resetting at each domain change is therefore beneficial. The fact that a smaller distance between a reset and a domain boundary is linked to better performance further supports this finding. Overall, domain-boundary resets provide an upper-bound reference for performance. However, in practice, domain boundaries are unknown and can be difficult to define, especially when domains are interleaved, so this upper bound is not realistic.
>
> | Method | RDumb@100 | RDumb@500 | RDumb@1000 | ASR (Ours) | Domain boundaries |
> | :-- | :-: | :-: | :-: | :-: | :-: |
> | Acc. (%) | 40.46 | 45.70 | 46.44  | 47.50 | 48.00 |
> | Distance (Avg.) | 682 | 282 | 218 | 41 | 0 |
>
> ---
>
> > [W4-2] Furthermore, this raises questions about the method's robustness in more dynamic environments, where it's unclear if it would still reset appropriately when the domain changes faster, or if frequent resets would degrade adaptation performance. Evaluating on a more dynamic benchmark, such as the CDC setting from DPCore, would be a valuable test.
>
> We can address whether ASR would still reset appropriately under faster domain changes, or whether frequent resets might degrade performance, even without fully dynamic environments.
>
> We observe that ASR maintains strong adaptation on IN-C, where domains evolve roughly 10× faster than CIN-C (`L263-L264`), indicating robustness under rapid changes (`Table D.6`). Although ASR triggers resets more frequently than a fixed-interval baseline such as RDumb, performance remains superior, suggesting that the timing and selection of resets are more critical than their frequency. We also consider evaluating ASR on fully dynamic benchmarks, such as the CDC setting.

---

> ### Author Response · Authors · 2025-11-21
> **Response to Reviewer wwgg (4/4)**
>
> > [Q1] In Figure 2, what do the numbers, icons, and the different colors in the top bar represent? A legend would be helpful.
>
> In the top bar of `Fig. 2`, the icons inside the dashed boxes, labeled with numbers, indicate class label indices. White icons represent correct predictions, while black icons represent incorrect predictions. We will include this in the camera-ready version.
>
> ---
>
> > [Q2] The paper states hyperparameters were tuned on a small holdout from CCC-Hard (Line 310). Were these exact same hyperparameters then used for all other datasets and settings (CIN-C, IN-C, IN-D109, non-i.i.d.)?
>
> Yes, the hyperparameters, tuned using only 5% of a holdout set from CCC-Hard (transition speed 2000; random seed 45), were applied unchanged to all other datasets and settings.
>
> ---
>
> > [Q3] A general question about resetting. How would a parameter-reset method like ASR be applied to TTA methods that do not primarily adapt the model parameters, such as prompt-based methods (e.g., FOA, DPCore)?
>
> ASR relies solely on the concentration of the model's outputs, not on which parameters are adapted. It can therefore reset learnable prompts in the same way it resets batch norm or full model parameters. While it remains an open question which parts of the prompts should be reset, we believe ASR is compatible with prompt-based TTA.
>
> ---
>
> **Reference**
>
> [1] Hoang et al. Persistent test-time adaptation in recurring testing scenarios. In NeurIPS. 2024.
>
> [2] Lee et al. Continual momentum filtering on parameter space for online test-time adaptation. In ICLR. 2024.

---

> ### Comment · Reviewer_wwgg · 2025-11-27
>
> Regarding [W4-1], I refer the authors to Table 6 in [a], which demonstrates two critical points: 1. Simply resetting at the domain boundary is not necessarily the upper bound of performance. 2. Knowledge learned from one domain can positively transfer to others (e.g., a "Gaussian Noise-learned Prompt" effectively improved performance on other noise or even blur domains). Consequently, ASR appears to suffer from a performance bottleneck by resetting too aggressively and negating this potential for positive transfer.
>
> Regarding [W4-2], the defense relying on IN-C is unconvincing. The statement that "domains evolve roughly 10× faster than CIN-C" misinterprets the concern; comparing the normal length of IN-C to a version repeated 10 times does not constitute the "dynamic" fast change I referred to. In IN-C, the domain remains constant for 50k samples, which is a block-stationary setting, not a fast domain change. I agree with **Reviewer CQqn** on the necessity of testing on **"fast switching distributions, stochastic duration of different domains, etc."** and **"The generation of this dataset could e.g. follow the CCC methodology; but designing a benchmark setting that considers different temporal processes for switching domains and data distributions would be quite interesting to study."**  However, it seems that **this setting has already been proposed in [a]** (the CDC setting with varying domain length and faster domain change). At a minimum, the method should be evaluated in this more challenging scenario to prove robustness.
>
> [a] Zhang et al. DPCore: Dynamic prompt coreset for continual test-time adaptation, ICML 2025

---

> ### Author Response · Authors · 2025-12-02
> **Follow-up Response to Reviewer wwgg (1/4)**
>
> > [Q4] Regarding [W1-1], the scalar scaling experiment misses my point. I am concerned about magnitude variance within a batch, not global scaling.
>
> Here we verify that the variance of the magnitude of logits within a batch is not a significant concern in our experiments.
>
> To manipulate this magnitude variance, we modify the logits within each batch as follows. For each sample, we subtract its mean logit value to obtain deviations, scale these deviations by a factor (e.g., 0.5, 2.0, 10.0), and then add the mean back. This procedure preserves the within-batch mean while scaling only the variance of logit magnitudes. As a result, large-magnitude logits become further amplified and small-magnitude logits become further compressed, or vice versa, depending on the scaling factor.
>
> For this experiment, we use ROID on a single split (transition speed 1000; seed 43) from CCC-Easy and CCC-Hard (see `Tables D.1` and `D.3`).
>
> The results show that our method is highly stable across a wide range of within-batch magnitude variances. Even when we increase the variance by more than 15×, accuracy remains nearly unchanged (<0.3%p on CCC-Easy, <1%p on CCC-Hard).
>
> This demonstrates that ***our concentration metric remains reliable even when logits of substantially different magnitudes occur within a batch***.
>
> | Avg. Var(logit scales) | 67 | 136 (Ours) | 264 | 1236 |
> | :-- | :-: | :-: | :-: | :-: |
> | Acc. (%) in CCC-Easy | 50.48 | 50.50 | 50.52 | 50.26 |
>
> | Avg. Var(logit scales) | 87 | 162 (Ours) | 344 | 1557 |
> | :-- | :-: | :-: | :-: | :-: |
> | Acc. (%) in CCC-Hard | 20.97 | 20.99 | 21.02 | 20.11 |
>
> ---
>
> > [Q5] In Eq. (1) , you compute Softmax(Mean(Logits)). If a batch contains logits of different scales (e.g., some ~1, others ~1000), the large values will dominate the arithmetic mean, effectively ignoring the samples with smaller logits. Averaging probabilities (Mean(Softmax(Logits))) is standard because it normalizes sample contributions. Why average raw logits despite this risk?
>
> The purpose of using `Softmax(Mean(Logits))` in `Eq. (1)` is to **emphasize high-confidence predictions**. Since large-magnitude logits reflect high-confidence predictions [3], averaging *raw logits* naturally highlights these samples.
>
> Models tend to update predominantly based on high-confidence predictions. Model collapse is similarly dominated by high-confidence predictions, and its early signs emerge when these predictions begin to concentrate on a small subset of classes.
>
> In contrast, `Mean(Softmax(Logits))` normalizes each sample, suppressing scale differences and discarding confidence information. This makes it much less sensitive to how strongly high-confidence predictions drift toward specific classes.
>
> By averaging raw logits before applying Softmax, `Eq. (1)` preserves confidence information and focuses on the growing concentration of high-confidence predictions, enabling more reliable detection of collapse signals.
>
> ---
>
> > [Q6] Regarding Figure 5, a minimum batch size of 16 is still effectively large for online TTA. Furthermore, the claim of robustness is unconvincing: ASR suffers a performance drop (\~8%) comparable to the baselines (\~9%) when reducing batch size. To properly validate this, please provide results for truly small batch sizes (e.g., 1) rather than stopping at 16.
>
> We evaluate our method on *truly* small batch sizes (`2` and `4`) using ROID as the base model on a single split (transition speed 1000; seed 43) of CCC-Easy, following the setting of `Fig. 10`. As shown in the table below, ASR consistently outperforms the baselines. At batch size `4`, ASR achieves 25.58, compared to 17.85 for RDumb, demonstrating that its robustness extends to smaller batch sizes than 16. However, at batch size `2`, the gap between ASR and RDumb narrows, as our reset mechanism requires a minimum number of samples to function effectively.
>
> | Batch size | 2 | 4 |
> | :-- | :-: | :-: |
> | ROID | 0.13 | 16.91 |
> | + RDumb | 5.87 | 17.85 |
> | + ASR (Ours) | 6.46 | 25.58 |
>
> Please note that, online TTA and continual TTA are different settings, and our focus is on a variation of the latter one: long-term continual TTA. Online TTA is an extreme scenario with the batch size of `1`, and most TTA methods fail to work under such an extreme condition. For batch size `1`, all methods yield near-random performance (~0.1), which is expected since standard continual TTA methods cannot operate under such extreme conditions. We emphasize that **online TTA with single-sample inputs represents a fundamentally different problem from long-term continual TTA**. One practical approach in this scenario, as noted in `L520-L521`, is to temporarily store online samples and evaluate the reset criterion once enough samples have been collected.

---

> ### Author Response · Authors · 2025-12-02
> **Follow-up Response to Reviewer wwgg (2/4)**
>
> > [Q7] Regarding [W1-2], please clarify what each data point in Fig. 3 represents. Is it a single batch or an aggregation over a domain? Do these points follow a temporal sequence? Furthermore, could this pattern be an artifact of the logit averaging in Eq. (1)? If a collapsing model produces high-magnitude logits for the dominant class, those values would dominate the arithmetic mean, artificially driving the "concentration" metric.
>
> In `Fig. 3`, each point corresponds to a single batch. The figure does not include temporal information, so points toward the right do not represent later adaptation steps.
>
> Below we empirically test whether the pattern in `Fig. 3` is an artifact of logit averaging. To do this, we measure prediction concentration after excluding the largest-magnitude logit in each batch. We also measure it after excluding the top 10% of logits by magnitude. The Pearson correlations are shown in the table below. Although slightly lower than the original value of 0.88 (as shown in `Fig. 3`), the correlations of 0.85 and 0.77 remain meaningful. This indicates that the effect of extremely large-magnitude logits is minimal, and **the pattern observed in** `Fig. 3` **cannot be attributed entirely to an artifact**.
>
> | Excluded logits | Pearson correlation |
> | :-- | :-: |
> | None | 0.88 |
> | Top-1 | 0.85 |
> | Top-10% | 0.77 |
>
> As the model approaches collapse, most logits increasingly take larger values for the dominant classes, with their magnitudes also growing. Consequently, the pattern in `Fig. 3` reflects contributions from many logits, not just a few extreme ones.
>
> ---
>
> > [Q8] Regarding [W1-3], the new claim that "the concentration of predicted classes" ignores the fact that successful adaptation also results in high prediction concentration.
> >
> > The method treats growing concentration as a proxy for collapse. However, how does ASR distinguish between "bad" concentration (collapse to a single class) and "good" concentration (confident, correct predictions)? For instance, methods employing distribution alignment (e.g., FOA) can achieve high concentration on the correct classes. By relying solely on the magnitude of concentration relative to a moving average, ASR risks triggering false-positive resets on models that are actually adapting successfully.
>
> We argue that performing resets when prediction concentration (`Eq. (1)`) is high is beneficial, regardless of whether the model is adapting successfully.
>
> **High prediction concentration produces a biased update signal for the model, regardless of whether predictions are correct or incorrect, and can ultimately lead to model collapse** (as noted in `L427-L429`).
>
> To test whether *false-positive resets*, which are triggered by temporarily high concentration in correctly adapting models, are actually beneficial, we conduct a controlled experiment under a non-i.i.d. label scenario, where such resets are particularly common. We prepare a batch with i.i.d. labels to ensure that any triggered reset would be considered a *true-positive*. For this experiment, we use a single split of CIN-C (the first split in `Table D.5`) with ROID as the baseline.
>
> For each reset, we compute the prediction concentration on that i.i.d. batch and apply a threshold to determine whether the reset is truly necessary, with the threshold initialized as described in `L168`. A higher threshold suppresses false-positive resets, allowing primarily true-positive resets to occur. As shown in the table below, we define thresholds over a broad range, and the highest threshold, −log(700), yields the fewest false-positive resets.
>
> We find that allowing false-positive resets leads to improved performance. This confirms that **interrupting biased parameter updates, even when the model appears to be adapting correctly, helps maintain stable long-term adaptation**. Therefore, false-positive resets support robustness in long-term TTA by preventing the accumulation of detrimental updates.
>
> | Threshold | -log(700) | -log(800) | -log(900) | ASR (Ours) | ROID (baseline) |
> | :-: | :-: | :-: | :-: | :-: | :-: |
> | Acc. (%) | 48.18 | 48.23 | 48.59 | 49.37 | 48.46 |

---

> ### Author Response · Authors · 2025-12-02
> **Follow-up Response to Reviewer wwgg (3/4)**
>
> > [Q9] Regarding [W2], while I appreciate the ablation, the practical benefit of the knowledge recovery module appears negligible. For ETA, the mean accuracy gain is only 0.34% (36.56% → 36.90%), and for ROID it is only 0.60% (40.96% → 41.56%). While the internal "recovery" metric shifts from negative to positive, this does not translate into meaningful performance improvements. Given the computational cost of computing Fisher information, is this complexity justified for such marginal gains?
>
> The benefit of the knowledge recovery module appears negligible because the evaluation in `[W2]` is conducted under an *easy-to-adapt* setting. However, **its benefit is not negligible** in more challenging adaptation scenarios. Indeed, `[W2]` confirms that **the knowledge recovery module is functioning as intended**, but IN-C used in `[W2]` is not an appropriate benchmark for measuring its performance contribution.
>
> As described in `L228–L231`, in challenging adaptation scenarios, we increase the regularization coefficient to encourage the model to reuse prior-domain information, thereby enhancing the effect of the knowledge recovery module. However, IN-C is relatively easy to adapt to. `Table D.6` shows that baseline accuracies are very similar in IN-C, so the benefit of knowledge recovery does not manifest strongly in this setting.
>
> To illustrate the module’s contribution, we consider **CCC-Hard** (see `Table D.3`). In several splits (e.g., 4, 7, and 8), removing the knowledge recovery module leads to substantial drops, while the full model consistently maintains higher accuracy. These observations indicate that the module functions flexibly, providing effective support under challenging domain shifts.
>
> | Split | 1 | 2 | 3 | 4 | 5 | 6 | 7 | 8 | 9 |
> | :-- | :-: | :-: | :-: | :-: | :-: | :-: | :-: | :-: | :-: |
> | ROID | 12.64 | 15.79 | 13.28 | 9.63 | 12.65 | 11.81 | 10.66 | 8.72 | 17.12 |
> | + ASR (Ours) | 20.99 | 22.51 | 20.40 | 21.22 | 22.93 | 21.36 | 22.37 | 23.84 | 24.25 |
> | + w/o knowledge recovery | 20.95 | 22.50 | 20.35 | 18.28 | 22.69 | 20.18 | 9.70 | 15.67 | 23.57 |
>
> Regarding computational cost, Fisher information is computed once per batch (batch size 64), adding only **~0.001s** per batch (<0.5% of the total 0.2s per batch; see `Table C.3`). This overhead has a negligible impact on per-batch computation and introduces no meaningful runtime burden.
>
> ---
>
> > [Q10] Regarding [W3-1], the new comparisons using consistent baselines is good. I encourage the authors to incorporate these rows into the main paper (Table 1) to ensure a comprehensive comparison against other SOTA add-on methods.
>
> We appreciate the reviewer’s positive comment on the new comparisons using consistent baselines (e.g., EATA+COME vs. EATA+ASR (Ours)). We are currently running experiments across all CCC splits and consistently observe that our method outperforms COME. In addition to CCC benchmarks, we also plan to evaluate on CIN-C under both i.i.d. and non-i.i.d. label scenarios, as well as on IN-C and IN-D109. Due to time and resource constraints, we cannot update `Table 1` during this rebuttal period, but we will include the full comparison results in the camera-ready version to further strengthen our contribution.
>
> ---
>
> > [Q11] Regarding [W4-1], I refer the authors to Table 6 in [a], which demonstrates two critical points: 1. Simply resetting at the domain boundary is not necessarily the upper bound of performance. 2. Knowledge learned from one domain can positively transfer to others (e.g., a "Gaussian Noise-learned Prompt" effectively improved performance on other noise or even blur domains). Consequently, ASR appears to suffer from a performance bottleneck by resetting too aggressively and negating this potential for positive transfer.
>
> While positive transfer across domains can occur, it typically happens only in **limited conditions** (e.g., between different noise types within the same “Noise” domain) **under mild distribution shifts**, as noted in DPCore [a] (`Sec. 4.3`; `Motivations`). Although our reset method may negate some potential for positive transfer in these scenarios, **this would not be a major concern for long-term TTA** with potentially significant distribution shifts.
>
> However, we do not entirely ignore knowledge transfer. Our method selectively resets model parameters and employs knowledge recovery, allowing it to accumulate information from previous domains. Leveraging this accumulated knowledge, it achieves stable adaptation even in challenging environments where inter-domain transfer is limited, as demonstrated by its advantage in CCC-Hard.

---

> ### Author Response · Authors · 2025-12-02
> **Follow-up Response to Reviewer wwgg (4/4)**
>
> > [Q12] Regarding [W4-2], the defense relying on IN-C is unconvincing. The statement that "domains evolve roughly 10× faster than CIN-C" misinterprets the concern; comparing the normal length of IN-C to a version repeated 10 times does not constitute the "dynamic" fast change I referred to. In IN-C, the domain remains constant for 50k samples, which is a block-stationary setting, not a fast domain change. (...) However, it seems that this setting has already been proposed in [a] (the CDC setting with varying domain length and faster domain change). At a minimum, the method should be evaluated in this more challenging scenario to prove robustness.
>
> We demonstrate the robustness of our method under dynamic domain shifts by applying the Continual Dynamic Change (CDC) protocol to IN-C. Unlike the block-stationary setting, IN-C configured under CDC explicitly introduces **fast switching** between domains and **stochastic domain durations**, controlled via the Dirichlet parameter $\delta$. We evaluate our method under the standard CDC setting ($\delta = 1.0$) and, to further emphasize its robustness, additionally under a more dynamic setting ($\delta = 10.0$).
>
> For $\delta=1.0$, RDumb experiences repeated drops, e.g., accuracy falls from 36.20 to 30.88 at the 4th transition when combined with ETA due to a full reset. In contrast, ASR steadily improves over time, rising from 28.48 to 38.98 across 20 transitions and maintaining more stable performance than RDumb.
>
> Similarly, under the more dynamic setting $\delta=10.0$, RDumb again suffers repeated drops, whereas ASR gradually improves and remains stable, reaching 41.68 at the 20th transition when combined with ROID.
>
> These results demonstrate that **ASR reliably maintains high and stable performance, even under rapid and stochastic domain shifts in real-world dynamic settings**.
>
> These experimental results below are included in `Tables D.14-D.15`.
>
> |   $\delta=1.0$ (Revisit#)    | 1     | 2     | 3     | 4     | 5     | 6     | 7     | 8     | 9     | 10    | 11    | 12    | 13    | 14    | 15    | 16    | 17    | 18    | 19    | 20    | Mean   |
> |-------|-------|-------|-------|-------|-------|-------|-------|-------|-------|-------|-------|-------|-------|-------|-------|-------|-------|-------|-------|-------|-------|
> | ETA   | 30.62 | 35.77 | 36.19 | 36.21 | 36.14 | 35.99 | 35.84 | 35.76 | 35.73 | 35.63 | 35.42 | 35.43 | 35.31 | 35.33 | 35.30 | 35.22 | 35.25 | 35.20 | 35.17 | 35.08 | 35.33 |
> | + RDumb| 30.62 | 35.73 | 36.20 | 30.88 | 35.24 | 36.24 | 30.83 | 34.62 | 36.11 | 32.01 | 35.14 | 36.39 | 33.46 | 34.56 | 36.67 | 34.33 | 32.97 | 36.23 | 36.16 | 31.11 | 34.28 |
> | + ASR (Ours)  | 28.48 | 30.25 | 33.48 | 33.03 | 32.42 | 35.28 | 35.33 | 35.27 | 35.72 | 36.19 | 36.85 | 37.43 | 38.49 | 38.34 | 38.65 | 38.41 | 38.94 | 38.92 | 38.96 | 38.98 | 35.97 |
> | ROID  | 34.11 | 36.82 | 37.04 | 36.99 | 36.86 | 36.78 | 37.20 | 37.05 | 36.59 | 36.89 | 36.68 | 36.74 | 36.86 | 36.95 | 36.66 | 36.83 | 36.62 | 36.68 | 36.67 | 36.84 | 36.69 |
> | + RDumb| 34.28 | 36.93 | 36.91 | 34.00 | 36.74 | 36.85 | 34.27 | 37.04 | 37.14 | 34.02 | 36.41 | 36.76 | 34.37 | 36.37 | 36.80 | 34.67 | 35.94 | 36.63 | 36.43 | 34.23 | 35.84 |
> | + ASR (Ours)  | 35.08 | 38.58 | 38.72 | 39.20 | 39.30 | 39.71 | 39.94 | 40.45 | 40.32 | 40.65 | 40.41 | 40.40 | 41.15 | 40.75 | 41.07 | 41.33 | 41.05 | 40.80 | 41.10 | 41.34 | 40.07 |
>
> |  $\delta=10.0$ (Revisit#)    | 1     | 2     | 3     | 4     | 5     | 6     | 7     | 8     | 9     | 10    | 11    | 12    | 13    | 14    | 15    | 16    | 17    | 18    | 19    | 20    | Mean   |
> |-------|-------|-------|-------|-------|-------|-------|-------|-------|-------|-------|-------|-------|-------|-------|-------|-------|-------|-------|-------|-------|-------|
> | ETA   | 29.79 | 34.80 | 35.72 | 35.68 | 35.72 | 35.56 | 35.33 | 35.47 | 35.31 | 35.23 | 35.23 | 35.09 | 35.02 | 35.01 | 34.88 | 34.78 | 34.83 | 34.75 | 34.70 | 34.62 | 34.88 |
> | + RDumb | 29.53 | 35.62 | 36.09 | 31.11 | 35.54 | 36.10 | 30.86 | 35.04 | 36.16 | 32.19 | 34.86 | 36.30 | 32.64 | 34.03 | 36.51 | 34.39 | 33.35 | 36.66 | 36.86 | 29.56 | 34.17 |
> | + ASR (Ours)  | 29.40 | 35.06 | 35.55 | 36.01 | 36.63 | 36.71 | 36.92 | 37.25 | 37.31 | 37.30 | 37.41 | 37.53 | 37.79 | 37.79 | 37.92 | 37.96 | 38.08 | 38.14 | 38.19 | 38.11 | 36.85 |
> | ROID  | 33.60 | 36.70 | 36.51 | 37.06 | 36.74 | 36.99 | 37.11 | 36.82 | 36.79 | 37.04 | 36.71 | 37.23 | 36.86 | 37.06 | 36.99 | 36.77 | 36.59 | 36.82 | 36.87 | 36.99 | 36.71 |
> | + RDumb | 33.76 | 36.66 | 36.67 | 34.30 | 37.01 | 37.26 | 33.51 | 36.69 | 36.76 | 34.44 | 36.16 | 37.19 | 34.38 | 36.55 | 36.52 | 35.05 | 35.77 | 36.98 | 36.34 | 33.80 | 35.79 |
> | + ASR (Ours)  | 33.62 | 36.95 | 37.89 | 38.37 | 39.54 | 39.49 | 40.42 | 40.22 | 40.63 | 40.83 | 40.77 | 41.23 | 41.24 | 41.48 | 41.60 | 41.66 | 41.99 | 41.82 | 41.51 | 41.68 | 40.15 |
>
> ---
>
> **Reference**
>
> [3] Wei et al. Mitigating neural network overconfidence with logit normalization. In ICML. 2022.

---

### Official Review · Reviewer_UtEd · 2025-10-31

**Soundness:** 3
**Presentation:** 3
**Contribution:** 3
**Rating:** 6
**Confidence:** 4

**Summary:**

This paper tackles long-term continual TTA, where models suffer from error accumulation and eventual collapse (predicting only a few classes). The authors propose:  1. Adaptive and Selective Reset (ASR) — dynamically determines when and which layers to reset based on prediction concentration, mitigating both over- and under-resetting. 2. Importance-aware knowledge recovery — recovers lost information post-reset using Fisher-based regularization with a hybrid (CMA + EMA) accumulation scheme. 3. On-the-fly adaptation adjustment — adaptively adjusts according to prediction inconsistency between the source and the current model. Extensive experiments on CCC, CIN-C, IN-C, and IN-D109 show large gains over prior TTA methods (e.g., +44.12% on CCC-Hard).

**Strengths:**

The studied problem of when and where to reset is a very practical problem, and it is a key step to enhance the stability of TTA under long-term and large-scale real-world application settings.

The proposed methods (from fixed or heuristic resets to a data-driven and risk-aware reset strategy) are simple yet effective.

The combination of adaptive reset timing, layer-wise selective reset, and Fisher-based knowledge recovery makes the overall method cohesive and effective.

**Weaknesses:**

The proposed method involves many hyperparameters, making it potentially difficult to tune in real-world online testing scenarios. Could the authors clarify how these hyperparameters are determined and whether ASR is sensitive to them?

**Questions:**

How about the performance of more recent and advanced TTA methods, such as CMF, PeTTA, or ReCAP (Region Confidence Proxy for Wild Test-Time Adaptation, ICML 2025), when combined with ASR? Is ASR still effective in these cases?

How about the computational efficiency of the accumulated Fisher Matrix? More detailed analyses are preferred.

I am also curious about the performance of ASR on lightweight backbones, such as ViT-Tiny or Swin-Tiny.

It would be more informative to replace Tables 3 and 4 with a unified figure showing results for a wider range of reset intervals (e.g., 2000, 4000, 6000) and reset ratios (e.g., 10%, 20%, 30%), enabling a more thorough sensitivity ablation analysis.

---

> ### Author Response · Authors · 2025-11-21
> **Response to Reviewer UtEd**
>
> > [W] The proposed method involves many hyperparameters, making it potentially difficult to tune in real-world online testing scenarios. Could the authors clarify how these hyperparameters are determined and whether ASR is sensitive to them?
>
> We agree that our method involves a relatively large number of hyperparameters. In practice, tuning can be done with minimal effort if a small holdout of test data is available. Under this assumption, we tune all hyperparameters using only 5% of a holdout split (transition speed 2000; random seed 45) from CCC-Hard, chosen to represent a median case, and apply the resulting values across all datasets and settings.
>
> We further provide a hyperparameter sensitivity study in `Appendix E.5`, showing that ASR maintains stable performance across all CCC levels within a broad range of hyperparameter values.
>
> ---
>
> > [Q1] How about the performance of more recent and advanced TTA methods, such as CMF, PeTTA, or ReCAP (Region Confidence Proxy for Wild Test-Time Adaptation, ICML 2025), when combined with ASR? Is ASR still effective in these cases?
>
> We evaluate ASR with several recent TTA methods on CCC, including CMF and PeTTA, using one split from each CCC level for a preliminary evaluation (transition speed 1000; seed 43). `Tables D.1-3` contain all relevant results, enabling direct comparison with the results shown here.
>
> Despite substantial differences in adaptation schemes and training objectives, ASR consistently improves performance, as shown in the table below. This demonstrates that ASR is broadly compatible with recent TTA methods and remains effective under diverse algorithmic choices. We plan to include results for ReCAP and the full CCC evaluation in the camera-ready version.
>
> | Method | CCC-Hard | CCC-Medium | CCC-Easy |
> | :- | :-: | :-: | :-: |
> | CMF | 1.06 | 38.77 | 48.29 |
> | + RDumb | 12.62 | 39.50 | 48.47 |
> | + ASR (Ours) | 19.57 | 40.36 | 50.49 |
> | PeTTA | 4.93 | 19.34 | 34.61 |
> | + RDumb | 6.40 | 23.73 | 37.60 |
> | + ASR (Ours) | 6.55 | 24.32 | 40.64 |
>
> ---
>
> > [Q2] How about the computational efficiency of the accumulated Fisher Matrix? More detailed analyses are preferred.
>
> The computational efficiency of Fisher-related parameters is summarized in `Table C.3`, specifically in the row labeled `+ w/o recovery (Sec. 3.4)`.
>
> All of these parameters, including $\bar{\mathcal{F}}$, $\tilde{\mathcal{F}}$, $\bar{\theta}$, and $\tilde{\theta}$, together account for only 0.1M parameters, which is negligible compared to the total model size of 25.5M. Specifically, $\bar{\theta}$ and $\tilde{\theta}$ each have size $|\theta|$, and $\bar{\mathcal{F}}$ and $\tilde{\mathcal{F}}$ are also of size $|\theta|$ because they store only the diagonal of the Fisher matrix, following the standard practice in Elastic Weight Consolidation (EWC) [1]. This means that each of the four components occupies just 0.025M parameters (0.098% of the total). The computation and memory overhead of these parameters is minimal, making ASR highly efficient in practice.
>
> ---
>
> > [Q3] I am also curious about the performance of ASR on lightweight backbones, such as ViT-Tiny or Swin-Tiny.
>
> We evaluate ASR on a lightweight backbone (ViT-Tiny). As shown in the table below, ASR consistently improves over baselines on CCC-Medium and -Easy. Because the backbone capacity is extremely limited, adapting to CCC-Hard is particularly challenging, which is reflected in the table where all ROID variants achieve only 0.1% accuracy. Even with such low accuracy, ASR achieves improvements similar to those in `Table 2`, demonstrating its effectiveness despite severe capacity constraints.
>
> | ViT-Tiny | CCC-Hard | CCC-Medium | CCC-Easy |
> | :- | :-: | :-: | :-: |
> | ETA | 2.29 | 34.20 | 47.09 |
> | + RDumb | 4.45 | 32.51 | 45.51 |
> | + ASR (Ours) | 5.30 | 36.48 | 47.23 |
> | ROID | 0.10 | 32.14 | 45.29  |
> | + RDumb | 0.10 | 31.61 | 44.92 |
> | + ASR (Ours)  | 0.10 | 34.47 | 45.64 |
>
> ---
>
> > [Q4] It would be more informative to replace Tables 3 and 4 with a unified figure showing results for a wider range of reset intervals (e.g., 2000, 4000, 6000) and reset ratios (e.g., 10%, 20%, 30%), enabling a more thorough sensitivity ablation analysis.
>
> We thank the reviewer for the valuable suggestion. By combining `Tables E.1` and `E.2` and expanding the range of reset intervals and reset ratios, we will provide a more informative ablation study. We are currently running the experiments and will include the results in the camera-ready version.
>
> ---
>
> **Reference**
>
> [1] Kirkpatrick et al. Overcoming catastrophic forgetting in neural networks. PNAS. 2017.

---

### Official Review · Reviewer_CQqn · 2025-11-01

**Soundness:** 3
**Presentation:** 3
**Contribution:** 2
**Rating:** 4
**Confidence:** 3

**Summary:**

The authors investigate the problem of long-term continual test time adaptation. Previous algorithms were shown to collapse at some point during longer term adaptation, and simple resetting methods have been proposed as baselines in this problem setting. Full model reset like in RDumb naturally yields a substantial drop in downstream performance in the first steps after the reset, hence the authors propose an adaptive resetting scheme where the timing of these resets is adaptively computed, and instead of resetting the full model, only parts of the model parameters are reset to baseline. The authors show in multiple experiments vs. main baselines ROID and RDumb that their adaptive strategy yields performance improvements in the CCC and other continual adaptation benchmarks.

**Strengths:**

- empirical performance of the proposed algorithm outperforms the considered methods
- results are clearly presented, and the paper is easy to follow
- considered experiments are well presented, relevant ablations performed

**Weaknesses:**

The method is overall incremental, and the components of the method are heuristic fixes to the collapse problem. The scientific depth of the study is limited; I see limited value between what is already known in the field. The method fundamentally does not overcome the issue that reset is required to prevent collapse in longer-term test-time adaptation.

While there are certaintly gains over the state of the art, they seem marginal and the amount of engineering to get these 1%-point improvements makes it questionable if the simpler variants are also sufficient in practice.

**Questions:**

1. Which empirically observed behaviors motivated the development of the different adaptation of the reset procedure? Can you make statements about sources of the collapse (as you allude to with Figure 3)
2. Section 3.3:  "recover essential knowledge lost" and "While parameters and their Fisher matrices increasingly align with the current domain, their proximity to reset makes them more vulnerable (...)" -- is there empirical and theoretical evidence for these statements?
3. Could you make the code for the algorithm available, will it be released under an open source license?
4. Could you comment on additional hyperparameters that were introduced, and the robustness of the method under variations of these?
5. Were the algorithms in e.g. Table 1 and other tables re-run for this paper, or are any number in the paper copied from previous papers?

---

> ### Author Response · Authors · 2025-11-21
> **Response to Reviewer CQqn (1/3)**
>
> > [W-1] The method is overall incremental, and the components of the method are heuristic fixes to the collapse problem. The scientific depth of the study is limited; I see limited value between what is already known in the field.
>
> We understand the reviewer’s concern, but our method is not a collection of small fixes. We reframe long-term TTA through a reset-based view, in which preventing collapse is treated as a task of continuous decision-making rather than following a fixed schedule. Prior work typically applies resets at fixed intervals [1] or only after collapse occurs [2]. In contrast, our approach continuously estimates this risk and jointly adjusts both reset and recovery through a unified mechanism.
>
> ASR integrates three components (`Sec. 3.3-4`) under a single principle: *intervening before collapse while preserving previously acquired knowledge*. This unified framing has not been explored in prior TTA works, where reset and knowledge recovery have been studied separately.
>
> To demonstrate how these components work together, we focus on the knowledge recovery module (`Sec. 3.4`), which effectively restores information lost due to resets. We evaluate this in a domain-recurring setting (IN-C, 20 visits; `Table D.6`) to measure how much knowledge from previously seen domains is recovered after resets.
>
> Recovered knowledge is measured as the gap between the current performance and the best performance achieved so far for each domain, which is then averaged across domains. Positive values indicate recovery, while negative values indicate forgetting.
>
> As shown in the table, our method substantially reduces forgetting and improves knowledge recovery. For instance, at revisit `#10`, ETA + ASR achieves 0.58 compared to -1.94 without the recovery module, and ROID + ASR achieves 0.16 compared to -0.52 without recovery. This confirms that the knowledge recovery module effectively preserves and reuses previously learned information.
>
> | Recovery (Revisit#) | 1 | ... | 10 | 15 | 20 | Mean |
> | :-- | :-: | :-: | :-: | :-: | :-: | :-: |
> | ETA + ASR (Ours) | 0.0 | ... | 0.58 | 0.08 | 0.02 | 0.24 |
> | + w/o knowledge recovery | 0.0 | ... | -1.94 | -1.16 | -0.76 | -0.56 |
> | ROID + ASR (Ours) | 0.0 | ... | 0.16 | 0.24 | 0.01 | 0.12 |
> | + w/o knowledge recovery | 0.0 | ... | -0.52 | -0.42 | -0.1 | -0.14 |
>
> ---
>
> > [W-2] The method fundamentally does not overcome the issue that reset is required to prevent collapse in longer-term test-time adaptation.
>
> Reset is an essential and widely recognized mechanism to prevent collapse in long-term TTA. Neural networks typically converge to sharp minima, making it difficult to escape and find better solutions through standard gradient updates [3]. Collapse is an even more challenging state than a sharp minimum, making recovery *nearly impossible* without reset [4]. Despite its importance, reset has been largely unexplored: existing approaches simply apply resets at fixed intervals with full-parameter recovery. We tackle these fundamental limitations, effectively exploring the potential of reset and proposing a strategy that dynamically adjusts both its timing and extent based on the model’s state.
>
> ---
>
> > [W-3] While there are certaintly gains over the state of the art, they seem marginal and the amount of engineering to get these 1%-point improvements makes it questionable if the simpler variants are also sufficient in practice.
>
> Designed to tackle model collapse in long-term TTA, our method is highly effective in challenging and realistic scenarios. CCC-Hard best reflects such scenarios, where we achieve a substantial 44.12% improvement over the state of the art, demonstrating that our approach effectively handles difficult tasks. In contrast, other benchmarks, such as CCC-Easy or IN-D109, are easier, and the modest improvements are what any method could achieve in such simple settings. This shows that the smaller gains on easy tasks do not imply that simpler variants would be sufficient for the more challenging benchmarks. As more benchmarks prone to model collapse become available, we expect the benefits of our approach to become even clearer.

---

> ### Author Response · Authors · 2025-11-21
> **Response to Reviewer CQqn (2/3)**
>
> > [Q1-1] Which empirically observed behaviors motivated the development of the different adaptation of the reset procedure?
>
> Two empirical findings motivated our redesign of RDumb's reset procedure.
>
> **First**, we observed that RDumb's **fixed periodic reset** is only suitable for standard TTA benchmarks where domain shifts occur at regular, pre-defined intervals. In realistic continual settings, domain shifts do not follow a fixed schedule, and their timing can vary significantly. As a result, RDumb resets either too early or too late, misaligned with the model’s risk of collapse, leading to suboptimal or unstable adaptation.
>
> **Second**, as shown in `Fig. 1`, RDumb suffers from substantial performance drops immediately after each reset. We found that this is primarily due to its **full-parameter reset** strategy, which discards all adaptation knowledge accumulated so far, causing significant recovery delays. This behavior becomes increasingly harmful under long-term, continually shifting domains.
>
> These observations motivated ASR, which introduces **adaptive and selective resets** to mitigate knowledge erasure and trigger resets when the model is at risk.
>
> ---
>
> > [Q1-2] Can you make statements about sources of the collapse (as you allude to with Figure 3)
>
> As described in `Sec. 2` (`Continual test-time adaptation`), collapse arises from the gradual accumulation of errors during long-term adaptation. TTA algorithms generate pseudo-labels for self-training, but these labels are often noisy. Over time, these errors compound, causing the model to become increasingly confident in incorrect predictions, which ultimately leads to performance degradation.
>
> Although `Fig. 3` does not directly show the cause of collapse, it demonstrates a strong correlation between high prediction concentration and performance drops. This suggests that **growing prediction concentration** serves as an early warning signal and can be used to anticipate collapse before it occurs.
>
> ---
>
> > [Q2-1] Section 3.3: "recover essential knowledge lost" -- is there empirical and theoretical evidence for these statements?
>
> **Theoretically**, our importance-aware regularizer can be seen to recover essential knowledge lost due to resets. This relies on two key mechanisms. **First**, we accumulate the updated parameters using a combination of CMA and EMA, preserving adaptation information in a manner similar to Polyak averaging [5], which provides a reliable reference of previously acquired knowledge. **Second**, the Fisher-based regularization follows the principle of Elastic Weight Consolidation (EWC) [6], assigning stronger penalties to parameters that are important for prior domains. Together, these mechanisms encourage critical parameters to remain close to their pre-reset values, effectively restoring knowledge that would otherwise be lost.
>
> **Empirically**, we evaluate the knowledge recovery effect in a domain-recurring setting on IN-C, where the same domains reappear multiple times (see `Table D.6`). This setup allows us to directly test whether the model can regain previously learned knowledge after a reset. We compare our method with a variant without the knowledge recovery module (`w/o knowledge recovery`). As shown in the table below, the variant without the knowledge recovery module gradually declines in accuracy across revisits, while our method consistently recovers it, demonstrating that the module effectively mitigates forgetting and preserves previously learned knowledge.
>
> | Acc. (Revisit#) | 1 | ... | 10 | 15 | 20 | Mean |
> | :- | :-: | :-: | :-: | :-: | :-: | :-: |
> | ETA | 30.64 | ... | 36.16 | 35.96 |  35.80 | 35.88 |
> | + ASR (Ours) | 28.68 | ... | 38.60 | 38.97 | 39.10 | 36.90 |
> | + w/o knowledge recovery | 28.64 | ... | 37.45 | 36.49 | 36.34 | 36.56 |
> | ROID | 35.32 | ... | 38.08 | 38.15 | 38.02 | 37.96 |
> | + ASR (Ours) | 35.66 | ... | 42.20 | 42.06 | 42.96 | 41.56 |
> | + w/o knowledge recovery | 35.35 | ... | 41.64 | 41.64 | 41.19 | 40.96 |
>
> We note that the recovered knowledge values reported previously in `[W-1]` also provide additional empirical evidence for the effectiveness of the knowledge recovery module, showing how much performance is recovered relative to the best performance achieved so far for each domain.

---

> > ### Comment · Reviewer_CQqn · 2025-11-22
> >
> > Dear reviewers, thanks for the reply. Some quick clarification questions:
> >
> > > In realistic continual settings, domain shifts do not follow a fixed schedule, and their timing can vary significantly.
> >
> > Where is this shown in the paper? I agree with what you are saying, but I think none of the benchmarks exhibit such a varying schedule? Could you e.g. run such an experiment on a modified variant of CCC to show this point?
> >
> > > This behavior becomes increasingly harmful under long-term, continually shifting domains.
> >
> > Can you quantify this? It seems that the recovery or RDumb indeed needs some time, I am unsure if this is really a significant drop.
> >
> > >> [Q2-1] Section 3.3: "recover essential knowledge lost" -- is there empirical and theoretical evidence for these statements?
> > > Theoretically, our importance-aware regularizer can be seen to recover essential knowledge lost due to resets.
> >
> > Can you define the term "knowledge" in this context? Is this equivalent to task performance, some property of the weights, or something different?

---

> ### Author Response · Authors · 2025-11-21
> **Response to Reviewer CQqn (3/3)**
>
> > [Q2-2] Section 3.3: "While parameters and their Fisher matrices increasingly align with the current domain, their proximity to reset makes them more vulnerable (...)" -- is there empirical and theoretical evidence for these statements?
>
> We previously argued that parameters become increasingly vulnerable to corruption as they approach a reset ("proximity to reset"; `L212-L213`). This behavior is widely recognized in the continual TTA literature [4, 7]. Conceptually, proximity to reset corresponds to the point at which accumulated errors can significantly compromise parameter integrity and harm adaptation.
>
> To empirically verify this, we define a *delayed reset* variant. Normally, resets are triggered when $\mathcal{C}_ {t} > \bar{\mathcal{C}}_ {t-1}$ (`L172-L173`). In the delayed variant, we postpone the reset until $\mathcal{C}_ {t} - \bar{\mathcal{C}}_ {t-1} > \epsilon$, retaining parameters beyond the usual reset point. As shown in the table below, delaying resets leads to performance drops, even below ETA, confirming that parameters are particularly vulnerable after the reset point and that such corruption substantially impairs adaptation. Results across various $\epsilon$ settings will be included in the camera-ready version.
>
> | IN-C (Revisit#) | $\epsilon$ | 1 | 5 | 10 | 15 | 20 | Mean |
> | :- | :-: | :-: | :-: | :-: | :-: | :-: | :-: |
> | ETA | - | 30.64 | 36.76 | 36.16 | 35.96 | 35.80 | 35.88 |
> | + ASR (Ours) | 0.0 | 28.68 | 33.00 | 38.60 | 38.97 | 39.10 | 36.90 |
> | + w/ delay | 0.01 | 27.94 | 28.06 | 27.94 | 28.30 | 28.12 | 28.29 |
> | ROID | - | 35.32 | 38.00 | 38.08 | 38.15 | 38.02 | 37.96 |
> | + ASR (Ours) | 0.0 | 35.66 | 41.03 | 42.20 | 42.06 | 42.96 | 41.56 |
> | + w/ delay | 0.01 | 35.60 | 37.78 | 38.28 | 38.61 | 38.62 | 38.07 |
>
> ---
>
> > [Q3] Could you make the code for the algorithm available, will it be released under an open source license?
>
> Yes, we plan to publicly release our code upon acceptance to ensure reproducibility.
>
> ---
>
> > [Q4] Could you comment on additional hyperparameters that were introduced, and the robustness of the method under variations of these?
>
> Below is a revised `Table C.2` that summarizes the hyperparameters introduced in this work. We also briefly describe their roles.
>
> | Hyperparameter | Description | Reference | ResNet-50 | ViT-B-16 |
> | :-: | :-- | :-: | :-: | :-: |
> | $\alpha_0$ | Cumulative concentration initialization factor | `L168` | 0.5 | $5.0 \times 10^{-4}$ |
> | $\mu_{\mathcal{C}}$ | Cumulative concentration’s EMA update momentum | `Eq. (2)` | 0.995 | 0.995 |
> | $r_0$ | Minimum reset proportion | `Eq. (3)` | 0.5 | 0.5 |
> | $\lambda_r$ | Reset proportion (or collapse risk) scaling factor | `Eq. (3)` | 20.0 | 0.1 |
> | $\lambda_{\mathcal{F}}$ | Fisher regularization coefficient | `Eq. (4)` | 5.0 | 5.0 |
> | $\lambda_0$ | Regularization coefficient initialization factor | `Eq. (6)` | 5.0 | 5.0 |
> | $\mu_0$ | EMA update momentum initialization factor | `Eq. (7)` | 0.15 | $1.0 \times 10^{-3}$ |
>
> * $\alpha_0$ and $\mu_{\mathcal{C}}$ control the cumulative concentration $\mathcal{C}_t$ (`Sec. 3.3`).
> * $r_0$ and $\lambda_r$ determine the proportion of layers to reset (`Sec. 3.3`).
> * $\lambda_{\mathcal{F}}$ is used for Fisher-based regularization (`Sec. 3.4`).
> * $\lambda_0$ and $\mu_0$ are used for hyperparameter reparametrization (`Sec. 3.5`).
>
> We evaluate the robustness of our method under variations of these hyperparameters on all CCC levels (see `Appendix E.5`). The results demonstrate that our method maintains stable performance across a broad range of values.
>
> ---
>
> > [Q5] Were the algorithms in e.g. Table 1 and other tables re-run for this paper, or are any number in the paper copied from previous papers?
>
> All results reported in this paper were obtained by re-running all algorithms for a fair and consistent comparison.
>
> ---
>
> **Reference**
>
> [1] Press et al. RDumb: A simple approach that questions our progress in continual test-time adaptation. In NeurIPS. 2023.
>
> [2] Niu et al. Towards stable test-time adaptation in dynamic wild world. In ICLR. 2023.
>
> [3] Keskar et al. On large-batch training for deep learning: Generalization gap and sharp minima. In ICLR. 2017.
>
> [4] Hoang et al. Persistent test-time adaptation in recurring testing scenarios. In NeurIPS. 2024.
>
> [5] Polyak et al. Acceleration of stochastic approximation by averaging. SIAM J. Control Optim. 1992.
>
> [6] Kirkpatrick et al. Overcoming catastrophic forgetting in neural networks. PNAS. 2017.
>
> [7] Wang et al. Continual test-time domain adaptation. In CVPR. 2022.

---

> ### Comment · Reviewer_CQqn · 2025-11-22
>
> Dear authors,
>
> I wanted to post a quick follow up after reading your rebuttal. The rebuttal overall clarifies various (but not all, I will follow up separately) concerns I had, but one important one remains: This paper feels too incremental for a venue like ICLR. I understand the motiviation, the approach, and the idea of adaptive reset while retaining more performance of the network than in a naive reset.
>
> The opportunity I see here is to show that this technique is actually required if we consider data streams that we expect in the real world: e.g. biases in the label distribution, fast switching distributions, stochastic duration of different domains, etc.
>
> What might make this story and motivation more convincing is if you could extend your contribution beyond the method to the *evaluation setting*. Similar to how RDumb which you compare to prominently introduced CCC to make the point that really long-term continous adaptation is challenging for existing TTA methods, could you think of a data sampling procedure that underlines the point and claims you are trying to make; i.e., where the naiive reset would actually severely degrade the performance, and the adaptive reset brings a substantial advantage.
>
> The generation of this dataset could e.g. follow the CCC methodology; but designing a benchmark setting that considers different temporal processes for switching domains and data distributions would be quite interesting to study.

---

> ### Author Response · Authors · 2025-11-28
> **Follow-up Response to Reviewer CQqn (1/2)**
>
> Thank you for further engaging in the discussion.
>
> >> In realistic continual settings, domain shifts do not follow a fixed schedule, and their timing can vary significantly.
> >
> > [Q6] Where is this shown in the paper? I agree with what you are saying, but I think none of the benchmarks exhibit such a varying schedule? Could you e.g. run such an experiment on a modified variant of CCC to show this point?
>
> As the reviewer pointed out, our benchmarks do not include varying domain-shift intervals. To evaluate robustness under such conditions, we construct modified CCC variants where the duration of each corruption is **randomly sampled from 1,000, 2,000, or 5,000 batches**. Note that, in the original CCC setting, each corruption persists for a fixed duration (e.g., **always 2,000 batches for an experiment**). This modification introduces a stochastic corruption-evolution schedule that better reflects realistic adaptation scenarios.
>
> We evaluate robustness by comparing results on our modified CCC variants with the original CCC benchmarks. In CCC-Easy, where collapse is less severe, all methods show similar trends in both the original and modified settings. In contrast, CCC-Hard reveals a clear difference. ROID+RDumb exhibits degraded performance under the modified setting, and we conjecture that RDumb's fixed reset schedule cannot adapt when challenging corruptions evolve unpredictably. Meanwhile, our method consistently preserves its performance gains, demonstrating that it adapts effectively even when the corruption is difficult and evolves irregularly.
>
> | Easy | original CCC | gain (%) | modified CCC | gain (%) |
> | :-- | :-: | :-: | :-: | :-: |
> | ETA | 43.46 | - | 43.17 | - |
> | + RDumb | 49.53 | +13.9 | 47.36 | +9.7 |
> | + ASR (Ours) | 51.27 | +17.9 | 51.15 | +18.4 |
> | ROID | 49.95 | - | 49.54 | - |
> | + RDumb | 49.76 | -0.3 | 49.33 | -0.4 |
> | + ASR (Ours) | 51.47 | +3.0 | 51.46 | +3.8 |
>
> | Hard | original CCC | gain (%) | modified CCC | gain (%) |
> | :-- | :-: | :-: | :-: | :-: |
> | ETA | 0.41 | - | 1.83 | - |
> | + RDumb | 9.46 | +2207 | 11.88 | +549 |
> | + ASR (Ours) | 15.95 | +3790 | 17.61 | +862 |
> | ROID | 9.63 | - | 16.51 | - |
> | + RDumb | 14.03 | +45.6 | 15.99 | -3.1 |
> | + ASR (Ours) | 21.22 | +120 | 21.56 | +30.5 |
>
> ---
>
> >> (RDumb suffers from substantial performance drops ...) This behavior becomes increasingly harmful under long-term, continually shifting domains.
> >
> > [Q7] Can you quantify this? It seems that the recovery or RDumb indeed needs some time, I am unsure if this is really a significant drop.
>
> We quantify `substantial performance drops` and `significant recovery delays` under the same evaluation setup as `Fig. 1` (ETA; CCC-Hard).
>
> **Substantial performance drops**
>
> For each reset, we measure the change in average accuracy by comparing the 10 batches before and after the reset, and then average these values over all reset points. RDumb exhibits an average **1.26%p drop** per reset, which corresponds to roughly *12% of its overall average accuracy (9.77%)*. This confirms that RDumb’s degradation at each reset is non-trivial.
>
> Within the region displayed in `Fig. 1`, RDumb drops by 11%p, which is 53.6% relative to the mean accuracy (20.50%) in that region. Note that we show relatively easy corruptions throughout CCC-Hard in `Fig. 1`, to better display the performance drop in RDumb. For other test batches, the recovered performance is overall low while the dropped performance is similar to `Fig. 1`, so the average drop is smaller than that of `Fig. 1`.
>
>
> **Significant recovery delays**
>
> We measure how many batches RDumb requires after each reset to reach the highest accuracy observed in the preceding 20 batches. When full recovery does not occur before the next reset, we count all batches until that reset. On average, RDumb requires **330 batches** to recover, indicating that RDumb takes substantially long to regain its pre-reset performance and highlighting the inefficiency of its full-parameter reset mechanism.
>
> Since `Fig. 1` visualizes only the high-accuracy region to emphasize post-reset degradation, the recovery delays shown in the figure appear longer; within this region, RDumb requires 702 batches.
>
> For better clarity and consistency, we are considering updating `Fig. 1` so that it more faithfully reflects the most representative degree of behavior.
>
> ---
>
> > [Q8] Can you define the term "knowledge" in this context? Is this equivalent to task performance, some property of the weights, or something different?
>
> Knowledge refers to information encoded in the model weights accumulated during adaptation, which correspond to $\bar{\theta}^{i}$ in `Eq. (4)`. Essential knowledge is identified via Fisher information, which highlights weights that are more informative about previous domains. Note that a direct quantification of knowledge is challenging; therefore, we use task performance as a proxy to assess it.

---

> ### Author Response · Authors · 2025-11-28
> **Follow-up Response to Reviewer CQqn (2/2)**
>
> We appreciate the reviewer's suggestion.
>
> The reviewer asked *whether our method is truly necessary under real-world-like data streams*. As addressed in our response to `[Q6]`, we evaluate this by constructing a ***CCC variant*** in which corruption transitions occur stochastically, creating an irregular and unpredictable shift schedule.
> In this setting, the naive reset strategy (RDumb) suffers from performance degradation, whereas the adaptive reset strategy (ASR) maintains consistently high performance. This shows that under more realistic data-stream dynamics, **ASR is not only beneficial but also practically necessary for stable long-term adaptation**.
> We are currently incorporating the experiments in the revision, and it will be visible via the updated manuscript as soon as possible.
>
> If you have further suggestions, please let us know; we are happy to take account of them.

---

### Author Response · Authors · 2025-11-21
**General Response**

We deeply appreciate your time and effort in reviewing our paper. We are currently revising the manuscript and will submit the updated version as soon as possible.

Thank you for your patience.

---

### Author Response · Authors · 2025-12-03
**Follow-up General Response**

We would like to once again express our sincere gratitude to all reviewers (**`CQqn#1`**, **`UtEd#2`**, **`wwgg#3`**, **`8GCC#4`**) for their time and constructive feedback. Below, we summarize the main concerns and how we have addressed them in the revised manuscript.

**Summary of Key Concerns and Our Responses**
1. **Reliability of our concentration metric:** We clarified its background in `Sec. 4.4` and demonstrated its robustness under challenging scenarios in `Sec. 4.5` and `Appendix C.1`.
2. **Effectiveness under dynamic domain shifts:** We demonstrated effectiveness both on our modified benchmark in `Sec. 4.6` and under conventional dynamic settings in `Sec. 4.7`.
3. **Clarifications on our knowledge recovery mechanism:** We provided theoretical justification and empirical evidence supporting its effectiveness in `Appendix E.6` and clarified its computational efficiency in `Appendix C.4`.
4. **Motivation for our reset design:** We clarified the limitations of conventional resets in `Sec. 3.2` and supported these observations with empirical evidence in `Appendix F.1`.


**Reference**

1. **Reliability of our concentration metric:** **`wwgg#3: {[W1-1], [W1-3], [Q4], [Q5], [Q6], [Q7], [Q8]}`**, **`8GCC#4: {[W1]}`**
2. **Effectiveness under dynamic domain shifts:** **`CQqn#1: {[Q6]}`**, **`wwgg#3: {[W4-2], [Q12]}`**
3. **Clarifications on our knowledge recovery mechanism:** **`CQqn#1: {[Q2-1], [Q2-2]}`**, **`UtEd#2: {[Q2]}`**, **`wwgg#3: {[W2], [Q9]}`**
4. **Motivation for our reset design:** **`CQqn#1: {[Q1-1], [Q7]}`**

We hope that all concerns, including any not explicitly mentioned here, have been addressed. We sincerely appreciate the reviewers for giving us the opportunity to improve our work.

---

### Meta-Review · Area_Chair_NxoK · 2026-01-04

**Summary:**

This paper aims to address resetting mechanisms in continual adaptation to prevent model collapse. Four experts participated in the review process, and the paper received mixed ratings (one positive [UtEd] and three negative [CQqn, wwgg, 8GCC]). The main concerns can be summarized as follows: (i) the method is incremental with limited depth and does not fundamentally address the challenges of long-term test-time adaptation [CQqn]; (ii) marginal performance improvements [CQqn]; (iii) the need for broader experimental validation, including additional baselines, backbones, and sensitivity analyses [UtEd]; (iv) concerns regarding the rationale behind the proposed “prediction concentration” mechanism [wwgg]; (v) unclear efficacy of the proposed “importance-aware knowledge recovery” module [wwgg]; and (vi) the effects of rapid resetting and the corresponding upper bound [wwgg, 8GCC].

Reviewers [CQqn, wwgg] actively participated in the discussion phase with the authors. While most concerns raised by [CQqn] were addressed and clarified through additional experimental validation, some major issues remain. Although [CQqn] acknowledges the motivation, methodology, and the idea of adaptive resetting, [CQqn] notes that the method remains incremental and points out an opportunity for the authors to further strengthen the motivation by more clearly demonstrating why resetting is necessary. The AC carefully reviewed the paper and the author responses and believes that the method is intuitive, addresses clear limitations of prior works (also noted by [wwgg]), and provides a simple yet effective solution for continual learning (also recognized by [UtEd]). Continual learning and adaptation are active research directions, and many existing methods focus on sophisticated adaptation mechanisms while overlooking the orthogonal yet effective strategy of resetting. Each module in the proposed adaptive resetting framework is reasonably designed with clear observations and motivations. Overall, the method demonstrates effectiveness beyond marginal gains when integrated into various baselines with different backbones (addressing concern (iii) raised by [UtEd]). It also shows robustness across batch sizes, different data streaming characteristics (e.g., i.i.d. and non-i.i.d.), reduced sensitivity to hyperparameters, and rapid resetting scenarios.

Another major concern relates to the rationale behind the proposed “prediction concentration” mechanism and the efficacy of the “importance-aware knowledge recovery” module. The proposed metric relies on averaged entropy within a batch, raising questions about its robustness to magnitude variations and batch composition. Additional concerns were raised regarding the impact of rapid resetting. In response, the authors provided thorough experimental validations with controlled factors to support their claims.

Overall, the AC believes that the authors responded effectively during the rebuttal phase and addressed most of the major concerns. Therefore, the AC recommends acceptance of the paper.

**Reviewer Concerns:**

The authors performed well in the rebuttal. Most of the concerns raised by [CQqn] were addressed; however, issues related to method incrementality and marginal improvement remain. Nevertheless, the AC finds that the method is simple and demonstrates effective improvements supported by extensive experimental validation.

The concerns raised by [UtEd, wwgg] were adequately addressed through clarifications and additional experiments. The concerns from [8GCC] stemmed from a misunderstanding of the proposed concept, which was clarified by the authors.

**Reviewer Scores:**

Reviewer [CQqn] may increase their rating, as most of their concerns have been addressed. Reviewer [UtEd] is expected to remain positive. Reviewer [wwgg] raised additional concerns, which were properly addressed by the authors. Therefore, the AC believes that reviewers [wwgg, 8GCC] are likely to increase their ratings.

---

### Decision · Program_Chairs · 2026-01-26

Accept (Poster)